# Human-correlated genetic models identify precision therapy for liver cancer

Miryam Müller[1✉], Stephanie May[1,2,20], Holly Hall[1,20], Timothy J. Kendall[3], Lynn McGarry[1], Lauriane Blukacz[4], Sandro Nuciforo[4], Anastasia Georgakopoulou[1,2], Thomas Jamieson[1], Narisa Phinichkusolchit[1,2], Sandeep Dhayade[1], Toshiyasu Suzuki[1], Júlia Huguet-Pradell[5], Ian R. Powley[1], Leah Officer-Jones[1], Rachel L. Pennie[1], Roger Esteban-Fabró[5], Albert Gris-Oliver[5], Roser Pinyol[5], George L. Skalka[1], Jack Leslie[6,7], Matthew Hoare[8,9], Joep Sprangers[1], Gaurav Malviya[1], Agata Mackintosh[1], Emma Johnson[1], Misti McCain[7], John Halpin[1], Christos Kiourtis[1,2], Colin Nixon[1], Graeme Clark[1], William Clark[1], Robin Shaw[1], Ann Hedley[1], Thomas M. Drake[1,2,10], Ee Hong Tan[1], Matt Neilson[1], Daniel J. Murphy[1,2], David Y. Lewis[1,2], Helen L. Reeves[7,11,12], John Le Quesne[1,2,13], Derek A. Mann[6,7,14], Leo M. Carlin[1,2], Karen Blyth[1,2], Josep M. Llovet[5,15,16], Markus H. Heim[4,17], Owen J. Sansom[1,2,18,19], Crispin J. Miller[1,2] & Thomas G. Bird[1,2,3,18,19✉]

Hepatocellular carcinoma (HCC), the most common form of primary liver cancer, is a leading cause of cancer-related mortality worldwide[1,2]. HCC occurs typically from a background of chronic liver disease, caused by a spectrum of predisposing conditions. Tumour development is driven by the expansion of clones that accumulate progressive driver mutations[3], with hepatocytes the most likely cell of origin[2]. However, the landscape of driver mutations in HCC is broadly independent of the underlying aetiologies[4]. Despite an increasing range of systemic treatment options for advanced HCC, outcomes remain heterogeneous and typically poor. Emerging data suggest that drug efficacies depend on disease aetiology and genetic alterations[5,6]. Exploring subtypes in preclinical models with human relevance will therefore be essential to advance precision medicine in HCC[7]. Here we generated a suite of genetically driven immunocompetent in vivo and matched in vitro HCC models. Our models represent multiple features of human HCC, including clonal origin, histopathological appearance and metastasis. We integrated transcriptomic data from the mouse models with human HCC data and identified four common human–mouse subtype clusters. The subtype clusters had distinct transcriptomic characteristics that aligned with the human histopathology. In a proof-of-principle analysis, we verified response to standard-of-care treatment and used a linked in vitro–in vivo pipeline to identify a promising therapeutic candidate, cladribine, that has not previously been linked to HCC treatment. Cladribine acts in a highly effective subtype-specific manner in combination with standard-of-care therapy.

Precision medicine for patients with advanced HCC has lagged behind other cancers. This is not because HCC has no discernible subtypes, but because targeting these has proved challenging. Tyrosine kinase inhibitors (TKIs; such as sorafenib[8] and lenvatinib[9]) were the only first-line treatments for unresectable HCC until 2020. Thereafter, the IMbrave150 study (atezolizumab with bevacizumab)[10] highlighted the potential of combination approaches with immune checkpoint inhibition (ICI) therapy, with enhanced responses for some patients and improved overall survival. Alongside advances in treatment options came an increased appreciation that heterogeneous treatment responses in patients with HCC provide a potential for patient stratification[5,6]. The lack of necessity for clinical biopsies in advanced HCC has resulted in a lack of tissue from late-stage disease. This hinders advances in defining clinically relevant stratification biomarkers and mechanistic understanding within subtypes for these patients. Preclinical models offer a biological platform for disease interrogation but, currently, few models faithfully recapitulate the complexity of human disease or have been validated against transcriptomic and phenotypic human HCC profiles[11,12]. There is therefore currently a need for human-relevant preclinical models to investigate therapy efficacies, providing guidance on subtype-specific treatments for different patient populations.

## Development of a suite of HCC models

To address this need, we first set out to generate a broad range of mouse models guided by the most commonly found genetic drivers of human

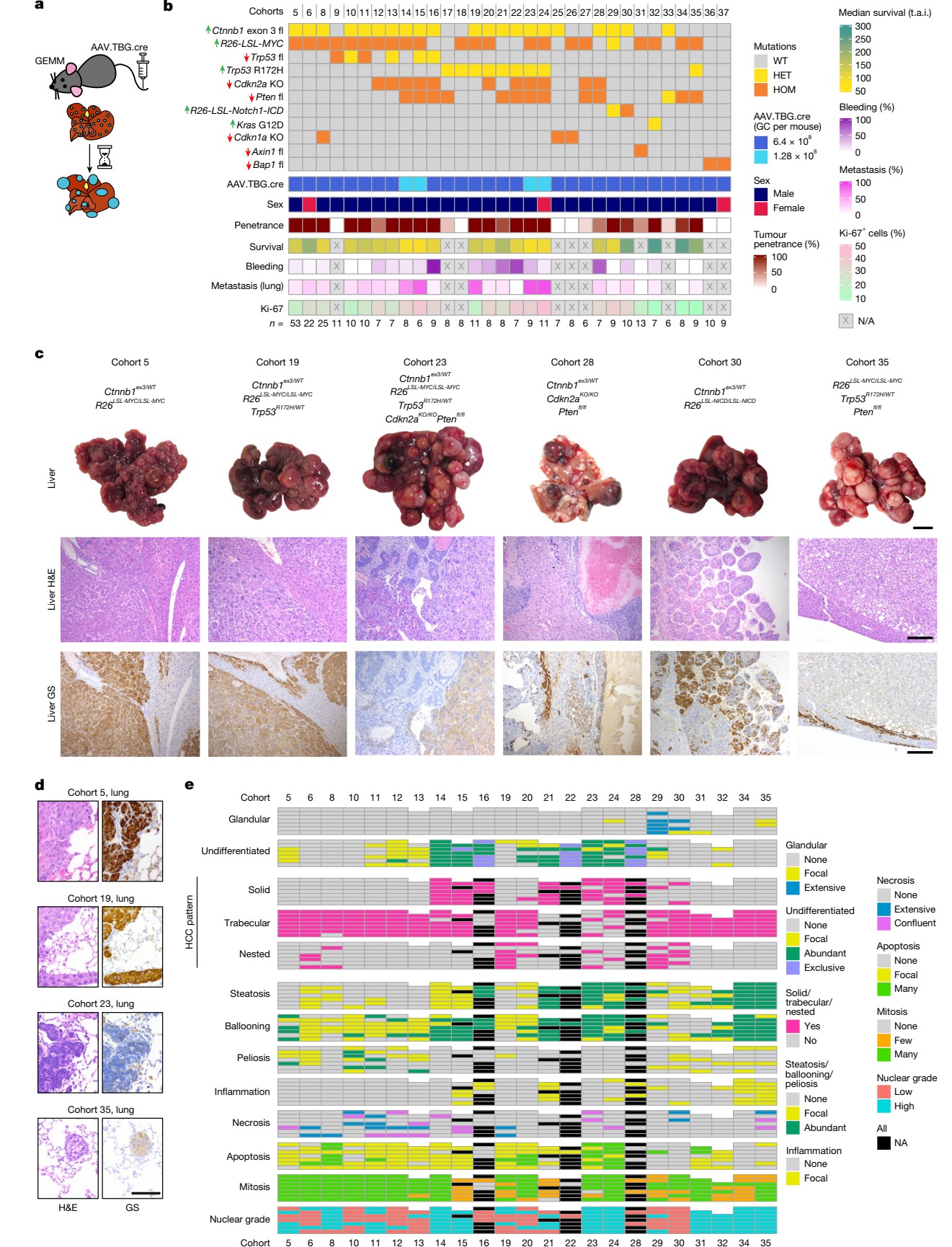

**Fig. 1 |** See next page for caption.

**Fig. 1 | Comprehensive characterization of the genetic HCC mouse models.**
**a**, Experimental scheme. Conditional genetically engineered mice induced with AAV.TBG.cre virus develop tumours after clonal recombination of genes classically associated with HCC in a TCGA study[4]. **b**, Specific combinations of mutations, but not numbers of mutations, drive model-specific features such as survival, tumour proliferation (Ki-67), bleeding from tumour and metastasis in mouse models of HCC. The up arrows represent gain of function (green) and the down arrows represent loss of function (red). T.a.i., time after induction (days); HET, heterozygous; HOM, homozygous. Exact values are provided in Supplementary Table 1. **c**, Representative images showing that variation in macroscopic and microscopic phenotype depends on combinations of mutations. Glutamine synthetase (GS) was used as an indicator of activated CTNNB1 signalling. Scale bars, 1 cm (macroscopy) and 200 μm (microscopy). Histology for the full range of HCC GEMMs is shown in Extended Data Fig. 3. **d**, Representative images show lung metastases resembling the primary tumour phenotype as demonstrated by haematoxylin and eosin (H&E) and GS staining. Scale bar, 100 μm. **e**, Mouse HCC models present common patterns and characteristics used for identification and classification of human HCC based on in-depth histopathological examination. *n* = 5–7 mice per cohort as indicated by bars.

HCC[4]. Human HCC is thought to evolve from a hepatocytic clonal origin under specific conditions promoting carcinogenesis, in contrast to recently described non-malignant clonal expansion[3,13–16]. We reproduced this aspect of cancer biology in our models by introducing the genetic alterations into adult mouse hepatocytes using conditional recombination technology and allowing the premalignant clones to evolve to HCC over time.

We intravenously injected adult mice with a viral vector encoding Cre recombinase with a hepatocyte tropism due to its thyroxine-binding globulin (TBG) promoter, AAV8.TBG.cre. This drove recombination of endogenous floxed alleles in individual hepatocytes in an immunocompetent environment (Fig. 1a). AAV8 was titrated to a dose ($6.4 \times 10^8$ genomic copies (GC) per mouse) that resulted in solitary hepatocyte targeting at low frequency (approximately 1%) and was highly hepatocyte specific (Extended Data Fig. 1a–d). Recombination occurred primarily in the first 5 days after injection, was observed across all three hepatocyte zones[17], but was significantly different between male and female mice (Extended Data Fig. 1e–h). This led to a lower tumour count and consequently extended survival in female mice after induction of HCC-related oncogenes (Extended Data Fig. 1i–k). Furthermore, varying the induction dose or mutational burden affected the tumour occurrence and the speed of progression to the end point (Extended Data Fig. 1i,l).

We next applied this strategy to a broad range of HCC-relevant oncogene/tumour suppressor genes using a standardized dose in male mice unless otherwise stated. We particularly focused on genes identified by a TCGA study[4] belonging to the WNT pathway, the cell cycle or the RTK–RAS–PI3K pathway growth. These genes were tested in multiple combinations with each other for their potency in tumour induction. (Fig. 1b). We included models with combinations that co-occur in early disease, such as *CTNNB1* + *MYC* or *PTEN* + *TP53*. However, we also included combinations that tend towards mutually exclusive in early disease but not in late-stage disease, such as *CTNNB1* + *TP53* (Extended Data Fig. 1m,n). We decreased the AAV induction titre in specific instances (cohorts 14, 15, 23 and 24, $1.28 \times 10^8$ GC per mouse) to reduce the clonal burden, facilitating progression of these more aggressive models to larger individual tumours. Genotyping of end-stage tumours confirmed a high fidelity of recombination in the alleles targeted by the AAV induction (97.4–100%) (Extended Data Fig. 1o). We monitored 35 genetically distinct models, including models with a whole-body knockout of *Cdkn1a* or *Cdkn2a*, for liver nodule growth for a minimum of 230 days after induction (Extended Data Figs. 1l and 2a).

The majority of our models (83%) developed end-stage tumours within the study timeframe and most (69%) showed a tumour penetrance of higher than 50%. Notably, some combinations, such as *MYC* overexpression + *Trp53* alteration, which induced HCC in some but not all previously published models[12,18], had very low to no tumour penetrance using our clonal evolution approach and did not reach end-stage tumours within the observed period. Reflective of human disease, we observed intratumoural haemorrhaging and/or rupture (bleeding) as well as metastatic spread to the lungs, one of the main metastatic sites in human HCC, together with bone and lymph nodes[2,19] (Fig. 1b–d and Supplementary Table 1). We observed a negative correlation between an increased number of driver mutations and survival, despite a reduced clonal induction with a lower AAV titre, and a positive correlation between an increased number of driver mutations and tumour proliferation, as well as between mutational burden and lung metastasis in our cohorts (Extended Data Fig. 2b). Tumour haemorrhage did not correlate significantly with mutational burden but occurred predominantly in cohorts with a mutational pattern showing activated *Ctnnb1* and *Pten* loss without *MYC* overexpression (Extended Data Fig. 2c). Macroscopic and microscopic appearances were consistent with human HCC and covered a wide range of histological subtype phenotypes microscopically. This included well-differentiated HCC (for example, cohorts 5 + 19), undifferentiated HCC (such as cohorts 23 + 28), pseudoglandular HCC (for example, cohort 30) and steatotic HCC (for example, cohort 35) (Fig. 1c and Extended Data Fig. 3). Lung metastatic lesions reflected primary tumour histopathology (Fig. 1d). Histopathological assessment of morphological parameters is currently the gold standard for differential diagnosis of liver cancer in patients[20]. They showed strong similarities to human HCC histopathology, including typically observed architectural patterns (trabecular, glandular, solid and nested) and cytological atypia. Different combinations of genetic alterations resulted in distinct morphologies (Fig. 1e).

In summary, we used combinatorial genetic alterations, relevant to human HCC, to drive the development of autochthonous tumours in 27 immunocompetent mouse models. Tumour growth happened progressively over several months with individual hepatocytes as the cell of origin. These models recreate key features characteristic of human HCC biology, including histopathological phenotypes and metastatic spread.

## Cross-species alignment and validation

To determine how well our models further represent human HCC, we performed unbiased transcriptional analysis. We included a range of well-established carcinogen-induced (TOX) and orthotopic transplant (OT) HCC mouse models with our genetically engineered mouse models (GEMMs) to make this comparison more comprehensive (Fig. 2a).

Using nonlinear dimensionality reduction (uniform manifold approximation and projection, UMAP[21]) we mapped mouse end-stage HCC data onto the human HCC data[4] (Fig. 2b). Individual models, both genetically modified and non-genetically modified, clustered within different regions in the UMAP plot (Extended Data Fig. 4a). However, mutational status is not always indicative of signalling status[22], and genomic profiling of human HCC previously showed that mutations are not exclusively prognostic of association with specific subtypes[4]. This is especially relevant for advanced disease stages with a relatively high mutational burden[23], where different genetic alterations can influence each other. We show that, for example, mutations in *CTNNB1*/*Ctnnb1* (human/mouse gene) do not always lead to upregulation of expression of downstream pathway targets (*GLUL*/*Glul*, *LGR5*/*Lgr5*, *LECT2*/*Lect2* or *NOTUM*/*Notum*) in human or mouse HCC (Extended Data Fig. 4b–f). Our mouse data also support the observation that mutational status by itself is not always predictive of the resemblance between cohorts (Extended Data Fig. 4a).

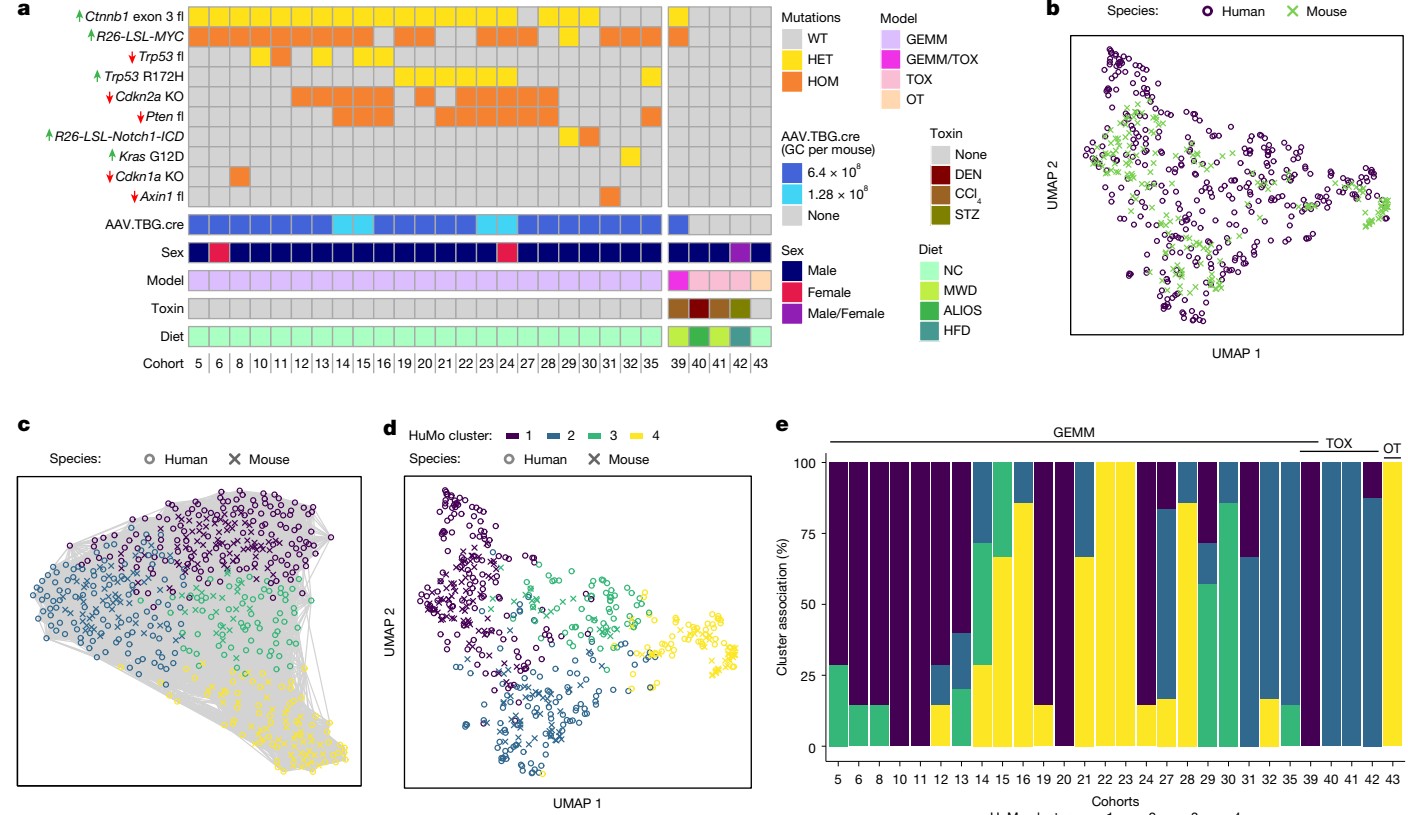

**Fig. 2 | Transcriptional alignment classifies four common human/ mouse (HuMo) clusters. a**, Summary overview of mouse models used for transcriptional analysis. In addition to the GEMMs, described in Fig. 1, TOX and OT models were included. These include mice that were treated with diethylnitrosamine (DEN), carbon tetrachloride (CCl₄) and streptozotocin (STZ), as well as multiple diets: modified western diet (MWD), American-lifestyle-induced obesity syndrome (ALIOS), high-fat diet (HFD) or normal chow (NC).

**b**, The UMAP visualization demonstrates overlap of mouse (GEMM, TOX and OT) and human (TCGA) HCC transcriptional datasets[4]. **c**, Unbiased clustering using a Louvain community detection algorithm identifies four groups within human and mouse (GEMM, TOX and OT) HCC data. **d**, The distribution of the subgroups identified in **c** with UMAP highlights shared HuMo clusters. **e**, All HuMo clusters are represented in the analysed GEMMs with varying heterogeneity within the individual cohorts.

We therefore went on to compare the human and mouse transcriptome data based on functionally and mechanistically relevant pathway enrichment. We used the Louvain method for community detection[24] to identify groups in our human/mouse HCC dataset (Fig. 2c). We detected four major human/mouse (HuMo) clusters (Fig. 2d). Genetic mouse models are represented in all four clusters with varying heterogeneity within cohorts, whereas the purely carcinogen-induced models are representative of only HuMo cluster 2 (Fig. 2e). Pathway enrichment analysis could establish cluster-specific characteristics. HuMo cluster 1 was enriched for pathways linked to metabolism and differentiation, but had negative enrichment for proliferation and inflammatory pathways. HuMo cluster 2 was related to cluster 1 but was distinct particularly through a higher enrichment in pro-inflammatory pathways. HuMo clusters 3 and 4 were both poorly differentiated and highly proliferative, with cluster 4 showing enrichment in epithelial-to-mesenchymal transition (Fig. 3a).

To assess whether the transcriptional clustering corresponded to similar histopathological features in mice and human HCC within the same cluster, we compared our mouse tumours to TCGA tissue[4]. We observed that mouse and human tissue belonging to the same HuMo cluster did indeed have analogous morphological characteristics (Extended Data Fig. 5a). Tissue from HuMo cluster 1 showed well-differentiated HCC (Extended Data Fig. 5b,c). HuMo cluster 2 tissue presented with inflammation, steatosis and steatohepatitis (Fig. 3b,c and Extended Data Fig. 5d). HuMo cluster 3 and 4 tissue displayed deposition of extracellular matrix and moderately (cluster 3) to poorly (cluster 4) differentiated HCC (Fig. 3d and Extended Data Fig. 5b,c). We

next validated our classification in a previously published, independent dataset of human HCC[25]. The patients could all be assigned a HuMo cluster with similar distribution dynamics to the TCGA dataset. Again, HuMo cluster 1 was enriched for immune-evasive signatures, including the immune-excluded subclass[25], and was de-enriched for ICI response signatures, including the IFNAP signature[26]. Conversely, HuMo cluster 2 had higher inflammatory signalling signatures and was enriched for immune-active tumours[25], but without WNT–β-catenin activation. HuMo clusters 3 and 4 featured a strong progenitor signature (CK19 mutation signature) consistent with the previously observed histological phenotype of these clusters. Only HuMo cluster 4 was significantly enriched for the inflamed HCC class with an immune-exhaustion signature and characterized by TGFβ and EMT signatures (Fig. 3e and Extended Data Fig. 6a,b).

When comparing survival across the species, there was general correlation between patients and the respective GEMMs across a range of molecular subtype classifications, including HuMo, Hoshida[27] and Chiang[28]. Importantly, HuMo offers a distinct patient classification. This clustering approach distinguished two patient populations within the Hoshida S3 molecular subclass, namely HuMo clusters 1 and 2. Hoshida et al. implied that S3 might consist of two subpopulations with *CTNNB1* as a dividing factor, but did not use this as a factor in their classification[27]. This distinction in our analysis resulted in differences in patient survival that were unappreciated when using the Hoshida classification; patients associated with HuMo cluster 2 had an improved survival probability relative to patients associated with the other HuMo clusters. Furthermore, this distinction separates the immune-excluded

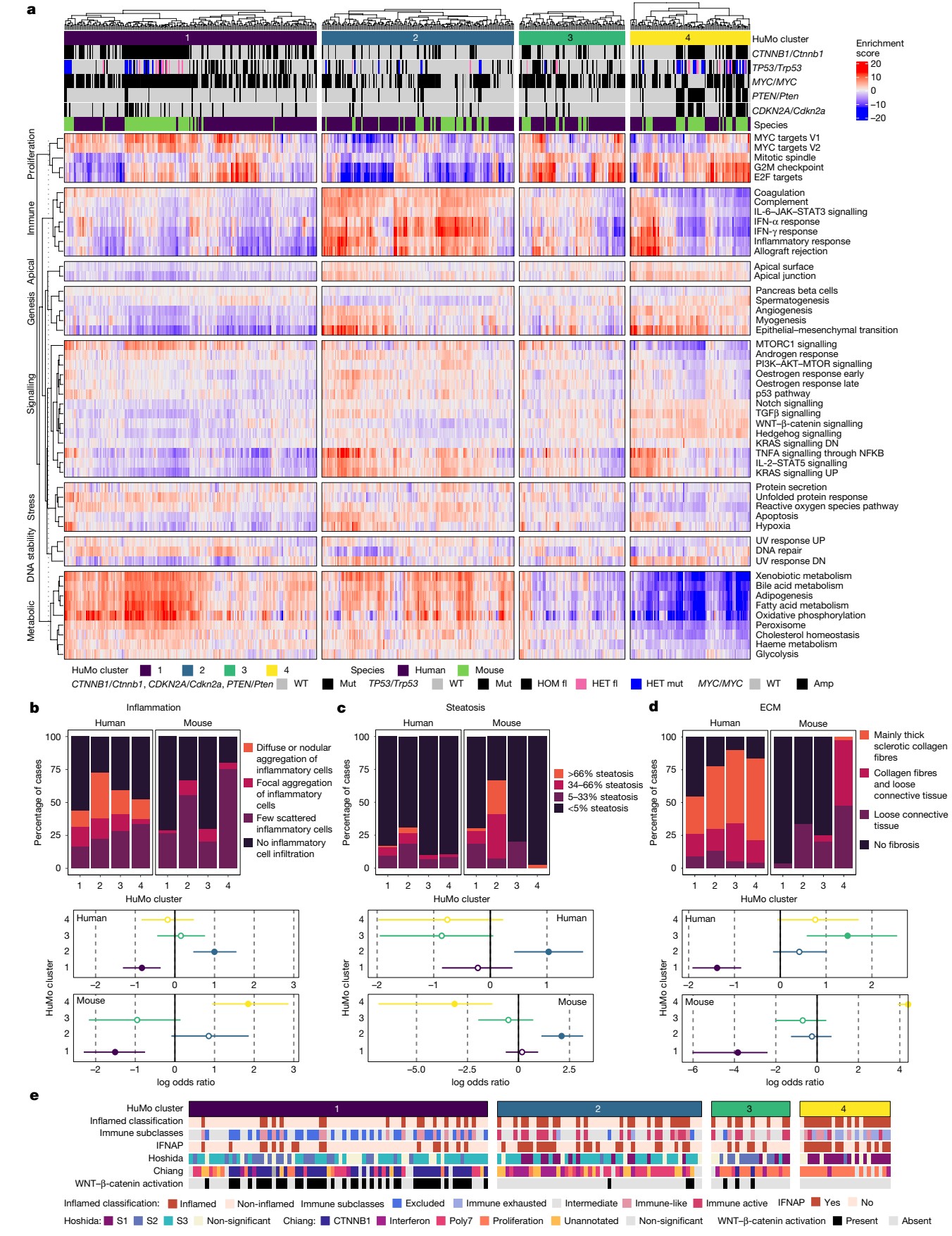

**Fig. 3 |** See next page for caption.

**Fig. 3 | Individual HuMo clusters have distinct transcriptional and histological features. a**, Pathway enrichment analysis across the GEMMs, non-GEMMs and human TCGA-HCC data[4], indicating distinct identifying characteristics, including metabolic activity/differentiation, *MYC/Myc* pathway activation, proliferation propensity or immune status for the four HuMo clusters. $n = 371$ (human) and 187 (mouse). DN, downregulated; UP, upregulated; mut, mutated; amp, amplified; fl, floxed. **b–d**, Transcriptional alignment correlates with histopathological similarities (inflammation (**b**), steatosis (**c**)

and extracellular matrix (ECM) (**d**)) between human ($n = 334$) and mouse ($n = 147$) liver samples from the same HuMo clusters. Data are the log odds ratio (dots) 95% confidence intervals (bars). Statistical analysis was performed using Fisher tests; $P > 0.05$ (open circles), $P \leq 0.05$ (closed circles). **e**, The distribution of HuMo clusters 1 to 4 and their alignment to previously reported molecular and immune HCC classifications and signatures[26–29] in a validation cohort of human HCC[25]. The full pathway heat map is shown in Extended Data Fig. 6 and associated statistical analysis is shown in Supplementary Table 2. $n = 171$ (human HCC).

(HuMo cluster 1) from the immune-active (HuMo cluster 2) subclasses. It also surpasses previous attempts of comparing mouse and human HCC data in scale and detail[11,12] (Extended Data Figs. 6a and 7 and Supplementary Table 2).

In brief, we identified four distinct clusters, common across human and mouse models, by integrating our mouse transcriptional data with human HCC transcriptional data. Our models recapitulate transcriptionally the full range of human HCC, including within individual clusters. This aligned with similar histopathological features and relative survival within clusters, with specific GEMMs representative of individual subtypes of human HCC. Moreover, our HuMo classification is able to discriminate between HCC with WNT–β-catenin activation (HuMo1) and those without WNT–β-catenin activation (Humo2) within non-proliferative tumours (Hoshida S3).

## Distinct responses to therapy by subtype

To examine the translational potential of our models, we investigated response to standard-of-care treatments. We focused on one model in a proof-of-principle set of experiments. Approximately 30% of patients with HCC have mutations leading to activation of the β-catenin signalling pathway[4]. HCC with activated β-catenin signalling has a low enrichment score for immune signatures and has been, in most cases, associated with immune exclusion[25,29]. Furthermore, active β-catenin pathway signalling has been linked to ICI resistance in a prospective HCC cohort study[5], suggesting a need for alternative treatment options for this patient subgroup. In the TCGA dataset, 65% (57 out of 88) of patients with mutations in *CTNNB1* were associated with HuMo 1 and made up 47% (57 out of 118) of patients in that cluster (Extended Data Fig. 8a,b). Moreover, humans and mice associated with HuMo cluster 1 had immune-cell paucity and a low immune score (Fig. 3a,b and Extended Data Fig. 8c–e). We therefore identified HuMo cluster 1 as the one most likely to correspond to the group of patients with activated β-catenin pathway signalling that would benefit from alternative treatment options. Cohort 5 mice (*Ctnnb1*<sup>ex3/WT</sup>*R26*<sup>LSL-MYC/LSL-MYC</sup>, hereafter BM mice) were used as a representative model and showed phenotypic resemblance to human *CTNNB1*-mutated HCC.

We aimed to mimic the treatment of established tumour lesions. We therefore first performed a time-course analysis for tumour onset in the BM mouse model (cohort 5) to determine an appropriate timepoint for the start of treatment. We observed clonal induction of hepatocytes, which evolved over time into microscopic lesions and then macroscopic tumour nodules, with glutamine synthetase (GS) as a marker of β-catenin driven tumour induction (Fig. 4a–c). Tumour evolution from single clones led to moderate intertumoural and intermurine transcriptional heterogeneity in end-stage tumours, including activation of pro-tumorigenic pathways such as proliferation or angiogenesis. However, while gene expression in tumours was markedly different to non-tumour tissue, it was also consistently different compared with livers with a global hepatocytic short-term expression of the same oncogenes (Extended Data Fig. 8f–i). This implied a consistent trajectory of clonal evolution occurring during tumour progression[3,13]. Relevant long-term models in which this evolution can take place are essential for studying HCC in preclinical models.

We started drug treatment at day 90, based on 100% of cohort 5 (BM) mice having macroscopic tumours and 96% of cohort 5 (BM) mice surviving past this timepoint (Fig. 4a–d and Extended Data Fig. 2a). Cohort 5 mice showed a significant increase in survival after treatment with the TKIs sorafenib and lenvatinib (Fig. 4e,f). However, treatment with the ICI agent anti-PD1 or treatment with ICI + VEGFRi (modelling atezolizumab + bevacizumab as first-line HCC systemic therapy[10]) did not impact the overall survival in this cohort (Fig. 4g and Extended Data Fig. 9a,b). These results are similar to the reported drug responses to TKIs and ICI in human patients with activated β-catenin signalling[5]. In mice from the immune-active HuMo cluster 2, ICI + VEGFRi resulted in an improved survival (Extended Data Fig. 9a,c). This is also consistent with the transcriptomic signatures predicting ICI response (Extended Data Fig. 6a).

Investigating disease progression after initial response to therapy, we observed changes in macroscopic and microscopic appearances in end-stage tumours of cohort 5 (BM) mice treated with lenvatinib. Tumours were different in colour and stiffer. Microscopic HCC patterns shifted from mostly well-differentiated to a poorer differentiated phenotype with a greater stromal presence (Extended Data Fig. 9d,e). Furthermore, more mice in this treatment arm presented with lung metastases compared with vehicle treatment or other treatments (Extended Data Fig. 9f). Monitoring of tumour growth using magnetic resonance imaging suggested a delayed and decreased tumour growth initially after lenvatinib treatment (Extended Data Fig. 9g). We also observed a higher metastatic burden in a second model (cohort 23, *Ctnnb1*<sup>ex3/WT</sup>*R26*<sup>LSL-MYC/LSL-MYC</sup>*Pten*<sup>fl/fl</sup>*Trp53*<sup>R172H/WT</sup>*Cdkn2a*<sup>KO/KO</sup>) with increased survival after lenvatinib treatment (Extended Data Fig. 9h–j). We hypothesized that the increased aggressiveness, manifested by morphological changes and greater metastatic burden, resulted from the extended survival coupled with an altered phenotype associated with acquired resistance to lenvatinib therapy. We therefore investigated livers of cohort 5 (BM) mice after 15 days and 30 days of lenvatinib treatment from day 90 after induction (Fig. 4d). We observed no differences in tumour morphology, but there was a decreased tumour burden through less proliferation, without increased cell death, at both the 15 and 30 day timepoints in lenvatinib-treated mice (Fig. 4h–j). There were no detectable metastases at either timepoint, supporting our hypothesis that the heightened aggressiveness in this model is a late-stage on-treatment event.

Overall, treatment responses in this specific GEMM were reminiscent of a distinct, common and difficult-to-treat subtype of HCC, characterized by a transient survival benefit observed in human phase 3 clinical studies[8,9].

## Screening-based therapy identification

After establishing the response to current standard-of-care treatments of mice representative of HuMo cluster 1 (cohort 5, BM), we concentrated on identifying therapeutic options for this difficult-to-treat subgroup. We performed an in vitro high-throughput screen based on GEMM-derived HCC organoids (HCCOs)[30], with subsequent in vivo validation in the respective GEMM (Extended Data Fig. 10a).

HCCOs recapitulate the transcriptomic profile, histological organization and tumorigenic potential of the primary tumour[31,32] and are

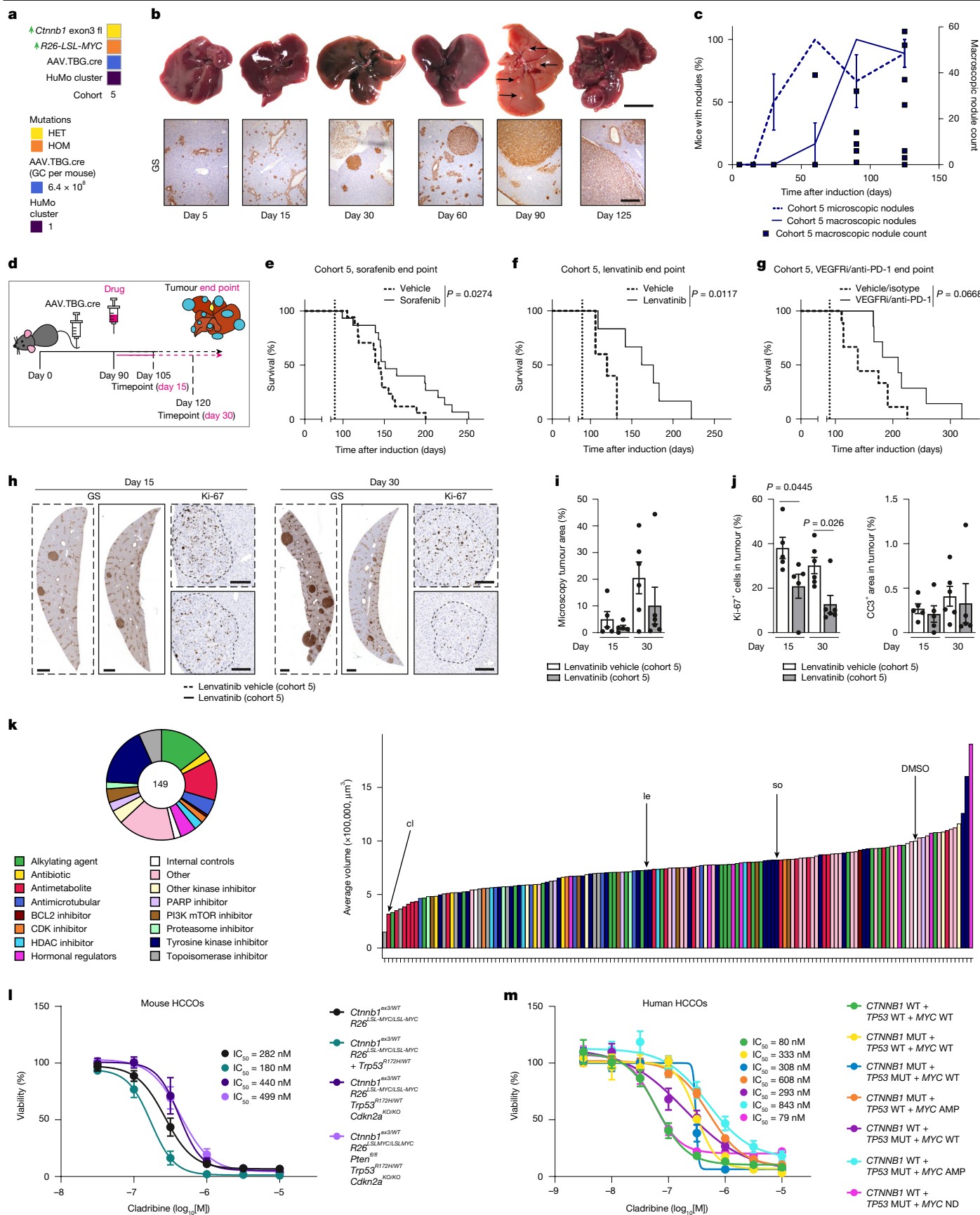

**Fig. 4 | See next page for caption.**

**Fig. 4 | Testing standard-of-care therapies and new therapeutic class identification in a representative mouse cohort of HuMo cluster 1. a**, The cohorts used in **b**–**k**. **b**, Temporal tracking of tumour development from a single clone to established HCC in male BM (cohort 5) mice using microscopic nodule detection through GS and macroscopic whole-liver assessment. The black arrows indicate macroscopic lesions at day 90. Scale bars, 200 μm (microscopic) and 1 cm (macroscopic). **c**, Quantification of microscopic and macroscopic nodules and macroscopic nodule count over time in male BM mice. *n* = 5, 6 and 9 mice for days 15, 30/60/90 and 125 respectively. Data are mean ± s.e.m. **d**, The treatment scheme for **e**–**j**. **e**,**f**, Treatment with the TKIs sorafenib (45 mg per kg, oral) (**e**) or lenvatinib (10 mg per kg, oral) (**f**) in male BM mice. The dotted vertical line indicates the treatment start. *n* = 17, 15, 5 and 6 mice for sorafenib vehicle, sorafenib, lenvatinib vehicle and lenvatinib, respectively. Statistical analysis was performed using log-rank tests. **g**, Combination treatment with VEGFRi (3 mg per kg, oral) and the immune-checkpoint inhibitor anti-PD1 (200 μg per mouse, intraperitoneal) in male BM mice. The dotted vertical line indicates the treatment start. *n* = 9 (vehicle + IgG isotype) and 8 (VEGFi + anti-PD1). Statistical analysis was performed using log-rank tests. **h**–**j**, Timepoint analysis (**h**) and quantification of GS (**i**) and Ki-67 and cleaved caspase 3 (CC3) (**j**) immunohistochemistry at day 15 and 30 after lenvatinib treatment in male BM mice. The dotted lines indicate the tumour borders. Scale bars, 1 mm (GS) and 200 μm (Ki-67). *n* = 5 and 6 mice at days 15 and 30, respectively (for vehicle and lenvatinib). Data are mean ± s.e.m. Statistical analysis was performed using a two-tailed unpaired *t*-test (day 15) and a Mann–Whitney *U*-test (day 30). **k**, High-throughput screening of 147 FDA-approved anti-cancer drugs plus internal controls, highlights antimetabolites (red) having an effect on growth of the HCCO tumouroids from BM mice; with cladribine (cl) having the greatest effect, while lenvatinib (le)/sorafenib (so) have only modest effect on the tumour cells. The full ranking is shown in Supplementary Table 3. **l**,**m**, In vitro validation of cladribine efficacy in mouse (**l**) and human (**m**) HCCOs. *n* = 3 different passages from 1–2 HCCO lines per mouse cohort, technical duplicates; 3 different passages from one to five human HCCO lines per driver combination, technical duplicates. Data are mean ± s.e.m.

therefore suited to investigate drug effects on tumour cells. They allow for rapid testing of a large range of drugs and for a side-by-side comparison between mouse-derived and human-derived tumour cells.

HCCOs derived from end-stage tumours of cohort 5 (BM) mice expressed β-catenin, its downstream target GS and retained MYC overexpression as well as markers of proliferation (Ki-67) and differentiation (HNF4a), features that are shared with the corresponding primary tumour (Extended Data Fig. 10b). Despite these similarities, the transcriptional phenotype of HCCOs differed from the original tumours. We propose that this is due to the simplified nature of HCCOs as an epithelial-cell-only model as well as adaptive response to the culture conditions. Overall, there was a convergence of HCCO phenotype arising from diverse GEMMs (Extended Data Fig. 10c). We tested a comprehensive drug library consisting of the 147 FDA-approved anti-cancer drugs available at the time (June 2019) plus internal controls and analysed their effect on HCCO growth (Fig. 4k and Supplementary Table 3). The most efficacious drugs were a group of antimetabolites—nucleobase analogues that interfere with DNA synthesis (Fig. 4k and Extended Data Fig. 11a). We validated the dose-dependent effect of cladribine, the most effective antimetabolite, in several distinct mouse and human HCCOs. This confirmed the results of the screen and demonstrated the nanomolar potency of cladribine (Fig. 4l,m). To establish whether this is a compound-specific effect, we tested a wide variety of antimetabolites and validated the high-throughput screen results. We demonstrated similar efficacy of clofarabine (a second-generation version of cladribine) and cladribine itself, suggesting a drug-specific on-target effect within this subclass of antimetabolites (Extended Data Fig. 11b,c). We also tested selected other drugs from our screen. Lenvatinib and sorafenib showed little tumour-epithelial efficacy in both the screen and separate validation, including in combination with cladribine (Extended Data Fig. 11b–f). Next, we treated cohort 5 (BM) mice, representing HuMo cluster 1, with either cladribine monotherapy or combination therapy of cladribine and lenvatinib, as a standard-of-care TKI (Fig. 5a,b). Cladribine monotherapy led to increased survival, but combination therapy extended survival further (Fig. 5c). Cladribine monotherapy reduced the number of tumours, but the remaining tumours still progressed to end-stage HCC. Combination therapy with lenvatinib showed a synergistic effect, almost completely eradicating all tumours (Fig. 5d,e).

Study progression to either clinical tumour end point or study end point (day 270 after induction) was limited in some animals (31% cladribine, 62% cladribine + lenvatinib) due to clinically substantial weight loss (<80%).

Treatment with either monotherapy or combination therapy showed a decrease in proliferation in end-stage tumours, but no alteration in apoptotic cell death. Notably, we observed an increase in CD3⁺ T cell infiltration into the tumour after combination therapy compared with vehicle treatment (Extended Data Fig. 12a–d). As the time of end point varied greatly between the different treatments, we analysed tumours at a defined timepoint of 30 days after the treatment start. Mice on monotherapy or combination therapy showed decreased tumour size and number, with a significant decrease in proliferation (Fig. 5f,g and Extended Data Fig. 12b,e). Both healthy and tumour tissue exhibited a greater extent of DNA damage (pH2AX), as expected after treatment with a nucleobase analogue, but this did not alter upregulation of another senescence marker (p53) nor apoptosis (Extended Data Fig. 12f–h). Again, we observed increased infiltration of CD3⁺ T cells into the tumour of mice that were treated with combination therapy (Fig. 5h). Given the lymphocyte infiltration observed after combination therapy, we tested a 'priming' approach with 1 week of combination therapy before ICI therapy (Extended Data Fig. 12i). This resulted in anti-tumour efficacy and immune infiltration and cytotoxicity (Fig. 5i–k and Extended Data Fig. 12j–l).

Finally, we tested whether cladribine, either as monotherapy or in combination with lenvatinib, is equally effective in mouse models representing other HuMo clusters. We treated cohort 23 (*Ctnnb1^{ex3/WT}* *R26^{LSL-MYC/LSL-MYC}Pten^{fl/fl}Trp53^{R172H/WT}Cdkn2a^{KO/KO}*) mice, representing HuMo cluster 4, and cohort 45 (*R26^{LSL-MYC/LSL-MYC}Kras^{G12D/WT}*) mice (induced with a higher titre of AAV.TBG.cre than cohort 32 to increase tumour burden to make survival time comparable to cohort 5), representing HuMo cluster 2 (Fig. 5b). Both monotherapy and combination therapy were effective in prolonging the survival of cohort 23 mice (Fig. 5l). However, cladribine did not extend the survival in cohort 45 mice (a wild-type (WT) *Ctnnb1* model with mutated *Kras*), either as a monotherapy or as a combination therapy with lenvatinib (Fig. 5m).

In this proof of concept, we demonstrated the potential of our GEMM platform to identify epithelial-targeting therapies that synergized effectively with standard-of-care treatments, the latter of which mainly targeting the tumour microenvironment. This combination of TKI and a repurposed FDA-approved anti-cancer compound led to highly effective subtype-specific treatment responses and a switch to a targetable immune phenotype.

## Discussion

Using a range of genetic alterations that are frequently associated with human HCC[4], we developed a suite of immunocompetent mouse models that closely resembles the development and progression of human HCC with hepatocytes as the cell of origin. Our models successfully recreate key molecular and pathophysiological events typical of human HCC, including tumour haemorrhaging and metastasis to the lungs[2,19]. They mimic various tumour microenvironments, such as immune-active and immune-desert or high/low stroma tumours. We demonstrated the clinical relevance of our models by integrating mouse data with publicly

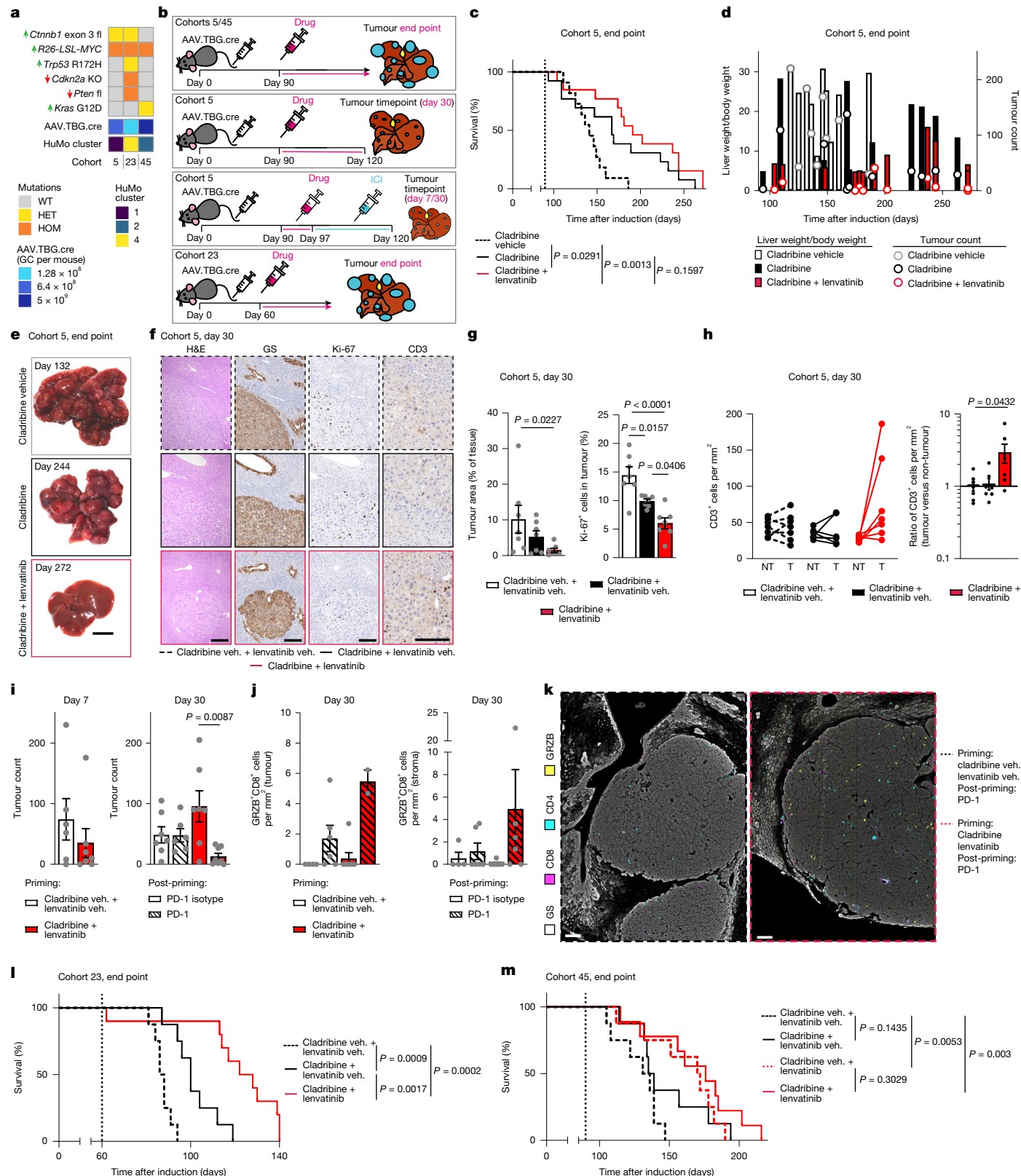

**Fig. 5 | See next page for caption.**

available human HCC datasets, defining shared subtypes and proving response to standard-of-care treatment. Furthermore, we showed that these models can be used as a preclinical platform, together with HCCOs, for investigating rapid drug repurposing, in addition to studying tumour evolution and mechanisms of drug resistance.

We appreciate that not all genetic alterations associated with HCC have been tested in this study. *TERT* promoter modifications, despite being frequent in human HCC (up to 60% of patients)[23], are difficult to model appropriately in mice due to biological differences between species. Mice have long telomeres and it would take several

**Fig. 5 | HuMo-cluster-specific treatment response to cladribine.**
**a**,**b**, Summary of the cohorts (**a**) and treatments (**b**) in **c**–**m**. **c**, Survival after cladribine treatment alone/or in combination with lenvatinib in male BM mice (cohort 5, HuMo 1). The dotted vertical line indicates the treatment start. $n = 11$ (vehicle), 13 (cladribine) and 13 (cladribine + lenvatinib). Statistical analysis was performed using log-rank tests. **d**, The liver-weight/body-weight ratio and tumour count over time. Data are shown as bars (liver weight/body weight) and symbols (tumour count) for individual mice. $n = 10$ and 9 (vehicle), 11 and 11 (cladribine), and 13 and 12 (cladribine + lenvatinib). **e**, Representative macroscopy images of male BM mice treated with vehicle, cladribine or cladribine + lenvatinib at the indicated days. Scale bar, 1 cm. **f**–**h**, BM mice treated with cladribine + lenvatinib were assessed for tumour proliferating cells (Ki-67) and CD3+ T cells. Representative images (**f**) and quantification of tumour area and Ki-67+ cells (**g**) and CD3+ cells (**h**). Non-tumour tissue (NT) and tumour tissue (T) are matched. Scale bars, 200 μm (H&E, GS and Ki-67) and 100 μm (CD3). $n = 7$ throughout. Data are mean ± s.e.m. Statistical analysis was performed using two-tailed Kruskal–Wallis tests with Dunn's correction (tumour area) and one-way analysis of variance (ANOVA) with Tukey's correction (Ki-67 and CD3 quantification). **i**–**k**, Priming of BM mice with cladribine + lenvatinib before treatment with anti-PD-1. **i**, Tumour counts at the day 7 and 30 timepoints. $n = 7$ (cladribine + lenvatinib, day 7; cladribine vehicle + lenvatinib vehicle +

anti-PD-1 isotype, day 30; and cladribine + lenvatinib + anti-PD-1 isotype, day 30), $n = 6$ (cladribine vehicle + lenvatinib vehicle, day 7; and cladribine vehicle + lenvatinib vehicle + anti-PD-1, day 30), $n = 8$ (cladribine + lenvatinib + anti-PD-1, day 30). Data are mean ± s.e.m. Statistical analysis was performed using two-tailed Mann–Whitney $U$-tests (day 7) and Kruskal–Wallis tests with Dunn's correction (day 30). **j**, The density of cytotoxic T cells in the tumour and stroma at day 30 after priming + anti-PD-1. $n = 4$ (cladribine vehicle + lenvatinib vehicle + anti-PD-1 isotype), $n = 6$ (cladribine vehicle + lenvatinib vehicle + anti-PD-1; and cladribine + lenvatinib + anti-PD-1 (note that four mice had no tumours to evaluate)) and $n = 7$ (cladribine + lenvatinib + anti-PD-1 isotype, day 30). Data are mean ± s.e.m. **k**, Representative images. Scale bars, 100 μm. **l**, Treatment with cladribine with or without lenvatinib in male cohort 23 mice ($Ctnnb1^{ex3/WT}$ $R26^{LSL-MYC/LSL-MYC}Pten^{fl/fl}Trp53^{R172H/WT}Cdkn2a^{KO/KO}$, HuMo 4). The dotted vertical line indicates the treatment start. $n = 9$ (cladribine vehicle + lenvatinib vehicle), 9 (cladribine + lenvatinib vehicle) and 10 (cladribine + lenvatinib). Statistical analysis was performed using log-rank tests. IC$_{50}$, half-maximal inhibitory concentration. **m**, Treatment with cladribine with or without lenvatinib in male cohort 45 ($R26^{LSL-MYC/LSL-MYC}Kras^{G12D/WT}$, HuMo 2). The dotted vertical line indicates the treatment start. $n = 8$ (cladribine vehicle + lenvatinib vehicle), 8 (cladribine + lenvatinib vehicle), 8 (cladribine vehicle + lenvatinib) and 9 (cladribine + lenvatinib). Statistical analysis was performed using log-rank tests.

generations of crossing mice with *Tert* deletions before detecting a noticeable effect of reactivating *Tert*[33]. This is an obstacle that will be difficult to overcome in mouse models of HCC and other means are needed to study *TERT* promoter mutations and their therapeutic targetability. However, as *TERT* promoter mutations are so omnipresent in HCC, they might be less relevant for subtyping and we did not identify a specific human *TERT* group that was separate from our GEMMs.

Furthermore, some combinations of genetic alterations showed low/no tumour penetrance in our GEMMs, for example, *Trp53* modifications in combination with *MYC* overexpression, while these showed high penetrance in HCC in previous models using hydrodynamic tail vein injections[12]. Administration of hydrodynamic tail vein injection has been shown to cause apoptosis in the liver[34], leading to higher inflammation and favourable conditions for tumour development. Moreover, levels of MYC might be a determining factor in a clone progressing to a tumour[35].

Although the majority of our studies were performed in male mice, we found no indication that the results are sex specific. Indeed, when we used the same genetic alterations in female mice, we observed a similar phenotype and cluster association (cohort 5/6, BM, male/female, respectively). However, AAV.TBG.cre induction seems to be less potent in female mice, which is particularly impactful in models with a lower mutational burden. Future experiments are needed to explore further genetic alterations or risk factors predominantly associated with female HCC in patient stratification, such as *Bap1* mutations or malignant transformation of hepatocellular adenomas[4,36].

In contrast to the GEMMs, human patients usually present with cirrhosis, which probably influences the course of disease establishment and progression[3] and impacts treatment options[2,20]. Future research incorporating multifaceted environmental factors in preclinical models, including advanced fibrosis, is needed to better understand HCC biology and potential differences between species. Our models can also be easily combined with environmental liver disease models, such as high-fat diets. Preliminary data from our transcriptomic analyses indicated that genetics dominate cluster association, with the addition of background fibrotic disease having little transcriptomic influence in mice (cohort 5 versus 37).

Our models strike a balance between allowing time for tumour evolution while still being time efficient. This enables future detailed investigation of tumour evolution and factors contributing to malignant transformation, especially as not all of the recombined clones expand into tumours.

Somatic mutations are poorly clinically actionable in HCC and remain difficult to target therapeutically. In the case of multiple genetic alterations, each individual contribution to tumorigenesis might be difficult to determine[37–39]. Our models with their increased complexity of multiple genetic alterations, similar to the mutational burden of late-stage HCC[23], enable the exploration of alternative targets and might contribute to understanding mutational dominance in different contexts. Moreover, by mimicking clonal evolution, they might help to identify the stage in tumour development—initiation, early nodule growth, malignant transformation—when a drug has an optimal effect.

We show that HCCOs are a tractable and rapid platform to identify treatments in combination with efficacy testing in vivo, and promote the principles of the 3Rs (replacement, reduction and refinement) for humane animal research. However, current cell culture conditions limit the translatability of HCCO-based drug response predictions and, therefore, validation in animal models remains essential. Future research in HCCOs needs to overcome the reduced complexity in cell culture, a general issue in organoid culture[40], and address options for co-culture with cells shaping the tumour microenvironment[41]. Modifying HCCOs with CRISPR technology may also provide useful insights to explore tumour biology and the mechanisms beyond drug vulnerabilities[42].

Importantly, we show that our GEMMs map transcriptionally and histologically to human HCC. Using a computational biology approach has enabled us not only to position our GEMMs, and select carcinogen-induced models, against human HCC, but also to identify four shared subclasses with defining characteristics. Notably, some of our models show a degree of heterogeneity often observed in human HCC[43], with tumours associated to several HuMo clusters. Our newly developed GEMMs represent all identified subtypes, whereas chemical-carcinogen-induced models included in this study only mapped to one HuMo cluster (cluster 2).

Our preclinical platform and classification system can be used as a resource for the HCC research community to streamline preclinical research and increase comparability of different mouse models. Furthermore, linking preclinical models with patient data can aid in stratifying patients to treatment, identifying new therapies and improving the likelihood of translational success. The HCCO screen enabled us to rapidly identify and test an FDA-approved anti-cancer drug, cladribine—not previously linked to HCC—in a clinically relevant model. We could show efficacy and improved survival in vivo together with standard-of-care treatment, which will allow for a swift translation into the clinic.

We believe that our approach of linking preclinical models to human data in a subtype-specific manner will also be applicable, cross-referable and advantageous in translational research of other solid cancers.

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

[1]Cancer Research UK Scotland Institute, Glasgow, UK. [2]School of Cancer Sciences, University of Glasgow, Glasgow, UK. [3]Centre for Inflammation Research, University of Edinburgh, Edinburgh, UK. [4]Department of Biomedicine, University Hospital and University of Basel, Basel, Switzerland. [5]Liver Cancer Translational Research Group, Institut d'Investigacions Biomèdiques August Pi i Sunyer (IDIBAPS), Hospital Clínic, Universitat de Barcelona, Barcelona, Spain. [6]Newcastle Fibrosis Research Group, Biosciences Institute, Faculty of Medical Sciences, Newcastle University, Newcastle upon Tyne, UK. [7]The Newcastle University Centre for Cancer, Newcastle University, Newcastle upon Tyne, UK. [8]Early Cancer Institute, University of Cambridge, Cambridge, UK. [9]Department of Medicine, University of Cambridge, Addenbrooke's Hospital, Cambridge, UK. [10]Centre for Medical Informatics, Usher Institute, University of Edinburgh, Edinburgh, UK. [11]Translational and Clinical Research Institute, Faculty of Medical Sciences, Newcastle University, Newcastle upon Tyne, UK. [12]Liver Group, Newcastle-upon-Tyne Hospitals NHS Foundation Trust, Newcastle upon Tyne, UK. [13]Department of Histopathology, Queen Elizabeth University Hospital, Glasgow, UK. [14]Department of Gastroenterology and Hepatology, School of Medicine, Koç University, Istanbul, Turkey. [15]Mount Sinai Liver Cancer Program, Division of Liver Diseases, Tisch Cancer Institute, Icahn School of Medicine at Mount Sinai, New York, NY, USA. [16]Institució Catalana de Recerca i Estudis Avançats, Barcelona, Spain. [17]University Digestive Health Care Center Basel—Clarunis, Basel, Switzerland. [18]Cancer Research UK Scotland Centre, Edinburgh, UK. [19]Cancer Research UK Scotland Centre, Glasgow, UK. [20]These authors contributed equally: Stephanie May, Holly Hall. ✉e-mail: miryam_mueller@yahoo.com; t.bird@crukscotlandinstitute.ac.uk

# Methods

## Mice, diets and treatments

All animal experiments were performed in accordance with UK Home Office licences (70/8891, PP0604995, 70/8646, 70/8468 and PP8854860) and in accordance with the UK Animal (Scientific Procedures) Act 1986 and EU direction 2010. They were subject to review by the animal welfare and ethical review board of the University of Glasgow and the University of Newcastle upon Tyne. To minimize pain, suffering and distress to the animals, we used single-use needles and non-adverse handling techniques. Mice were housed under controlled conditions (specific-pathogen free, 12 h–12 h light–dark cycle, 19–22 °C, 45–65% humidity) with access to food and water ad libitum. We added environmental enrichments, in the form of gnawing sticks, plastic tunnels and nesting material to all of the cages. Welfare of animals was defined by clinical symptoms, including visible masses, any degree of reduced mobility/distress, weight loss or evidence of haemorrhage; however no maximal tumour volume end points for intrahepatic tumours were mandated. No mouse exceeded the humane end points stipulated in the Home Office Licenses.

Unless otherwise specified, male mice on a mixed background were used. The following transgenic mice strains were used: $Gt(Rosa)26Sor^{tm14(CAG-tdTomato)Hze}$ ($R26^{LSL-Tom}$)[44], $Ctnnb1^{tm1Mmt}$ ($Ctnnb1^{ex3}$)[45], $Gt(Rosa)26Sor^{tm1(MYC)Djmy}$ ($R26^{LSL-MYC}$)[46], $Trp53^{tm1Brn}$ ($Trp53^{fl}$)[47], $Trp53^{tm2Tyj}$ ($Trp53^{R172H}$)[48], $Cdkn2a^{tm1.1Brn}$ ($Cdkn2a^{KO}$)[49], $Pten^{tm2Mak}$ ($Pten^{fl}$)[50], $Gt(Rosa)26Sor^{tm1(Notch1)Dam}$ ($R26^{LSL-NICD}$)[51], $Kras^{tm4Tyj}$ ($Kras^{G12D}$)[52], $Cdkn1a^{tm1Led}$ ($Cdkn1a^{KO}$)[53], $Axin1^{fl}$ (ref. 54), $Bap1^{tm2c(EUCOMM)Hmgu}$ (ref. 55). Genotyping was performed by Transnetyx using ear notches taken for identification purposes at weaning (3 weeks of age). Mice were induced between 8 and 12 weeks of age, unless otherwise indicated, with AAV8.TBG.PI.eGFP.WPRE.bGH (AAV8-TBG-GFP) (Addgene, 105535-AAV8), AAV8.TBG.PI.cre.rBG (AAV8-TBG-cre) (Addgene, 107787-AAV8) or AAV8.TBG.PI.Null.bGH (AAV8-TBG-Null) (Addgene, 105536-AAV8). Virus was diluted in 100 µl PBS to the desired concentration and injected through the tail vein. Unless otherwise specified, mice received a dose of $6.4 \times 10^8$ GC per mouse.

For the GEMM + MWD model, 6-week-old mice were kept on a modified western diet (Envigo, TD.120528) plus sugar water (23.1 g l$^{-1}$ fructose and 18.9 g l$^{-1}$ glucose) in combination with repeated CCl4 injections (intraperitoneal (i.p.), 0.2 µl g$^{-1}$ of body weight; vehicle, Cornoil) as previously described[56] and were induced with AAV.TBG.cre at 10 weeks of age.

For the DEN/ALIOS model, C57BL/6 WT mice, were injected with a single dose of DEN (80 mg per kg by i.p. injection) at 14 days of age. Mice were fed ALIOS diet (Envigo, TD.110201) and sugar water (23.1 g l$^{-1}$ fructose and 18.9 g l$^{-1}$ glucose) from 60 days of age. Mice were collected at day 284.

For the MWD + CCl4 model, the mice were kept on a modified western diet (Envigo, TD.120528) plus sugar water (23.1 g l$^{-1}$ fructose and 18.9 g l$^{-1}$ glucose) in combination with repeated CCl4 injections (i.p., 0.2 µl g$^{-1}$ body weight; vehicle, Cornoil) as previously described[56].

For the streptozotocin (STZ) model, male and female C57BL/6J WT mice were injected with a single dose of STZ (200 µg in 0.1 M citrate buffer, pH 4.0) subcutaneously at 2 days of age. Mice were fed a high-fat diet (TestDiet 58R3, 1810835) from 30 days of age. All STZ–HFD-treated livers showed pale yellow colour at 6 weeks, mild swelling at 8 weeks, granular surface at 12 weeks and tumour protrusion at 20 weeks of age[57]. Mice were collected between 17 and 35 weeks of age.

For the orthotopic model, Hep-53.4 cells (female C57BL/6J hepatoma cell line; Cytion, LOT-L230232R) were injected intrahepatically into the left lobe of male C57BL/6J mice. The procedure was performed under isoflurane general anaesthesia. Analgesia was given to the mice for pain management. Mice were collected at 28 days after implantation or left to reach an approved humane end point.

For therapeutic intervention in the BM model, drugs were given at 90 days after induction or in other models determined by mean cohort survival relative to the BM model survival as indicated in the figures. The following drugs were used: sorafenib (LC Laboratories S8502, daily, oral gavage, 45 mg per kg; vehicle, 50% chremophor/50% ethanol, then, before dosing, 3 parts H$_2$O added), lenvatinib (SelleckChem, S1164 (end-point studies); or Eisai (monotherapy timepoint studies); daily; oral gavage, 10 mg per kg, vehicle, 3 mM HCl), anti-PD1 (BioLegend, RMP1-14; twice per week; i.p., 200 µg; vehicle, PBS; control, IgG, BioLegend, RTK2758), cladribine (SelleckChem, S1199; daily; i.p., 20 mg per kg; vehicle, PBS), VEGFRi (AstraZeneca, AZD2171; daily; oral gavage, 3 mg per kg; vehicle, 0.5% (w/v) HPMC, 0.1% Tween-80, in H$_2$O). To help with drug-induced weight loss, mice on cladribine treatment received irradiated peanuts and sunflower seeds as diet supplements. If mice reached 83% of their weight at treatment start, cladribine treatment was withheld until they gained weight to at least 90% of weight at treatment start. Mice who dropped below 80% of weight at treatment start were sampled according to licence limitations. Confounding factors (for example, litter mates, induction date) were taken into consideration when allocating mice into groups but mice were not randomized using a specific method. Mice who presented with a visible tumour before treatment start were excluded from the experiments according to a priori established criteria. Animal technicians dosing the mice were blinded to the genotype of the mice. The number of biological replicates was ≥3 mice per cohort for all experiments. Further details are provided in the figure legends and Supplementary Table 1.

## Animal tissue collection

GEMMs were sampled at specific timepoints or at the end point. The end point was defined as the mouse having reached a liver weight/body weight ratio of >20% or having adverse side effects from the tumour, such as tumour haemorrhaging. Mice who died of tumour haemorrhaging were included in the survival analysis but not in any downstream analysis. Tumours were measured macroscopically using digital callipers, and visible tumours were counted. Images of whole livers were taken using a Canon PowerShot G9X camera with a ruler present in each picture. Tissue was either sampled in neutral buffered saline containing 10% formaldehyde or snap-frozen on dry ice.

## Histology and immunohistochemistry

Liver, tumour and lung tissues were fixed using neutral buffered saline containing 10% formaldehyde, dehydrated and embedded in paraffin, and cut into 4-µm-thick sections. The sections were dewaxed and stained with H&E or Sirius Red using standard protocols. Additional sections were stained immunohistochemically using the primary antibodies listed in Supplementary Table 5. Primary antibodies were detected by HRP-labelled secondary antibodies and subsequently stained using a peroxidase DAB kit; either Agilent (K3468) or Leica (DS92563) DAB for tissue processed in autostainer or Vector Laboratories (SK-4100) with haematoxylin as a counterstain (immunohistochemistry) or by fluorescent-labelled secondary antibodies (Invitrogen) with DAPI used as counterstain (SouthernBiotech, 0100-20) (immunofluorescence).

## Microscopy and quantitative analysis of immunohistochemistry

Images were obtained on the Zeiss Axiovert 200 microscope using the Zeiss Axiocam MRc camera. For image analysis, stained slides were scanned using the Leica Aperio AT2 slide scanner (Leica Microsystems) at ×20 magnification. Quantification of blinded, stained sections (GS, Ki-67, CC3, CD3, yH2AX, p53) was performed using the HALO image analysis software (v.3.1.1076.363, Indica Labs). Quantification of microscopy tumour area in BM mice was performed based on nodules, independent of GS status. Quantification of Ki-67$^+$ was by percentage/cell number and CC3$^+$ by percentage/tumour area.

Lungs were microscopically analysed for the presence of extrahepatic HCC spread by examining H&E and GS sections. Metastasis was scored in a binary manner as detected or not-detected but was not analysed in respect to individual metastasis burden per mouse.

Images for tissue comparison to HCCOs were taken on the Zeiss 710 confocal microscope.

## Tumour genotyping

After extraction from whole tumour, DNA was suspended in Transnetyx assay buffer and was analysed by Transnetyx using probes (p53Flox EX, Bap1-2 EX, PTEN-EX, LSL-EX-1, Tg-MYC, Axin1-1 EX, Ctnnb1-16 EX) and was additionally purified and concentrated using the Monarch Genomic DNA purification kit (New England Biolabs, T3010L) according to the manufacturer's instructions. Generation of amplicons indicating successful recombination of genetic loci was performed by PCR (Eppendorf Mastercycler x50a) using the OneTaq Quick-Load 2× Master Mix with Standard Buffer (M0486S); reactions were set up according to the manufacturer's instructions, amplification conditions (Supplementary Table 4) and primer sequences−β-catenin exon 3 (Supplementary Table 4) and $KRAS^{G12D}$ (Supplementary Table 4). The resulting PCR reactions were separated by electrophoresis on 1.5% agarose (Melford, A20090-500) gel, using the size marker Quick-Load Purple 1 kb Plus DNA Ladder (New England Biolabs, N0550S) and bands were visualized using SYBR Safe DNA gel stain (Invitrogen, S33102). The Gels were imaged on the Chemi-Doc Imaging System (Bio-Rad). Concordance between CTNNB1 recombination results between the two methods was 100%. Where possible, the samples used in histological comparison were also assessed genotypically. Where not possible, due to DNA contamination/low quality, tumours were replaced by other end-stage tumours from the same cohort to achieve $n \geq 6$ per cohort (total $n = 4$ additional samples).

## Quantitative analysis of fluorescence immunohistochemistry

Fluorescent tiled images were generated on the Opera Phenix High-Content Screening System (Perkin Elmer) at ×20 magnification. Fluorescence was detected using the same settings throughout. Consecutive, non-overlapping fields were analysed blindly using Columbus Image analysis software (v.2.8.0.138890, Perkin Elmer). Positivity gating thresholds were defined using negative controls. For representative images, processing adjustments were performed equally.

## Multiplex immunofluorescence immunohistochemical staining

Mouse liver samples (thickness, 4 µm) were sectioned and placed onto TOMO hydrophilic adhesive microscope slides (Matsunami, 0808228600). After antibody validation, semi-automated multiplex immunofluorescence staining was performed on the Ventana Discovery Ultra platform (Roche Tissue Diagnostics, RUO Discovery Universal v.21.00.0019). Fluorescence detection was performed using an Opal fluorophore tyramide-based signal amplification system (Akoya Biosciences). All primary antibodies were optimized using a pH 9 antigen retrieval solution (CC1, Roche Tissue Diagnostics, 06414575001) at 95 °C for 32 min. A denature step was applied using pH 6 antigen retrieval solution (CC2, Roche Tissue Diagnostics, 05279798001) for 24 min between each Opal detection and primary antibody application.

The primary−secondary−opal fluorophore combinations (CD45−HRP−Opal480; CD8−HRP−Opal690; CD4−HRP−Opal620; GranzymeB−HRP−Opal650; GS−HRP−Opal520; MYC−HRP−Opal570) are described in Supplementary Table 5. ImmPRESS rat and Opal 780 were manually applied in their specific sequences, and the remaining reagents were fully automated on the Ventana DISCOVERY ULTRA platform (Roche Tissue Diagnostics). Three drops of nuclear DAPI counterstain (Roche Tissue Diagnostics, 05268826001, RTU (Ready to Use), 24 min) were applied to each sample for nuclear detection.

## Multiplex immunofluorescence image acquisition and analysis

Whole-slide images were collected on the PhenoImager HT multispectral slide scanner (Akoya Biosciences, v.1.0) using a ×10 objective before the acquisition of each region of interest (ROI) using a ×20 objective. Each ROI was spectrally unmixed using InForm (Akoya Biosciences, v.2.6.0) using a project-specific spectral library created using single-channel dyes and an autofluorescence mouse liver control.

Visiopharm was used for all image analysis. For tissue segmentation, a bespoke, in-house-trained deep learning algorithm (v.2024.06.0.19093 ×64) was trained using the deep learning module with a U-Net backbone, to segment each lobe into tumour, stroma, non-tumour GS and background ROIs. Tumour regions that were smaller than 10,000 µm² were classified as a 'clone' region. Tumour and clonal regions were then dilated to generate 'peritumoural stroma' and 'periclonal' regions, respectively. Necrotic regions were manually segmented. Tumour regions were eroded to create 'tumour centre' and 'tumour periphery' regions. T cells were classified with a 'T cell' label within these regions using the threshold module (v.2024.07.1.16745 ×64) to threshold CD4, CD8 and CD45 fluorescence channels using the original image features. Each image was verified by a pathologist to confirm regional and T cell label segmentation. Post-processing steps were included to change T cell labels into their respective regional labels, that is, a T cell found within the tumour ROI would be changed to 'tumour T cell' and so on. Output variables were then generated and exported for downstream data analysis. Area of entire lobes were generated, and areas for each ROI as well as regional mean pixel intensities of MYC, mean pixel intensities of each marker in each T cell label and $x$–$y$ coordinates of each T cell label. The Phenoplex Guided workflow was used for T cell phenotyping, which generated a phenotype list that was exported for data analysis.

## Duplex immunofluorescence immunohistochemical staining

For duplex immunofluorescence immunohistochemical staining (Extended Data Fig. 12b (bottom)), 4-µm-thick mouse HCCOs and liver lobe samples were sectioned and placed onto TOMO hydrophilic adhesive microscope slides (Matsunami, 0808228600). After antibody validation, fully automated multiplex immunofluorescence staining was performed on the Ventana DISCOVERY ULTRA platform (Roche Tissue Diagnostics, RUO Discovery Universal v.21.00.0019). Fluorescence detection was performed using an Opal fluorophore tyramide-based signal amplification system (Akoya Biosciences). All primary antibodies were optimized using a pH 9 antigen retrieval solution (CC1, Roche Tissue Diagnostics, 06414575001) at 95 °C for 32 min. A denature step was applied using pH 6 antigen retrieval solution (CC2, Roche Tissue Diagnostics, 05279798001) for 24 min between each Opal detection and primary antibody application. The primary−secondary−opal fluorophore combinations (MYC−HRP−Opal570, GS−HRP−Opal520) are described in Supplementary Table 5. One drop of nuclear DAPI counterstain (Akoya, 232121) was applied to each sample for nuclear detection.

## Duplex immunofluorescence image acquisition and analysis

Whole-slide images were collected on the PhenoImager HT multispectral slide scanner (Akoya Biosciences, v.1.0) using a ×20 objective using Motif mode. Images were spectrally unmixed using Inform (Akoya, v.2.6.0) using an autofluorescence liver control slide to remove autofluorescence.

## Tumour scoring

H&E-stained sections and tumours were additionally assessed by a consultant liver histopathologist and UK liver pathology External Quality Assessment scheme member (T.J.K.) working at a national liver transplant centre. All assessment was undertaken blind to all other data, including genotype and sampling times. An initial screen of the first available 135 cases was made to identify prominent histological features in lesional and non-lesional liver that could be semi-quantitatively assessed.

Accepting the inherent limitations of semi-quantitative subjective histological assessment but using a single observer to remove inter-observer considerations, semi-quantitative/ordinal scoring systems were created for lesional and non-lesional features. Slides containing transections of whole lobes from each animal were assessed as a

whole, giving an overall score or impression rather than scoring on an individual-lesion basis.

Non-lesional liver was scored for steatosis (none, focal, abundant) and lobular inflammation (none, focal, abundant). A minority of slides included insufficient non-lesional liver for assessment.

For lesional assessment, the presence of glandular tumour, that is, meriting designation as adenocarcinoma (none, focal, extensive) and undifferentiated carcinoma (none, focal, abundant, exclusive) was assessed first. All hepatocellular neoplastic lesions had the morphological and cytological appearances of malignancy, that is, HCC. In all cases in which there was HCC, the following features were assessed using the categories in parentheses: lesional pattern (any from nested, trabecular, solid), lesional steatosis (none, focal, abundant), lesional cell ballooning (none, focal, abundant), intralesional inflammation (none, focal, abundant), lesional necrosis (none, focal, confluent, extensive), lesional cell apoptosis (none, focal, many), intralesional peliosis (none, focal, abundant), lesional nuclear grade (low, minimal/low pleomorphism; high, highly pleomorphic).

### Whole-tumour RNA-seq

Whole tumour and healthy tissue were snap-frozen and stored at −80 °C. To cover the breadth of our models, for each cohort, tissue from the shortest and longest surviving mouse as well as tissue from mice with survival closest to median cohort survival was chosen. Tissue was homogenized using the Precellys Evolution homogenizer and bulk RNA was isolated using a Trizol (Invitrogen) extraction protocol according to the manufacturer's instructions. RNA quality and quantity was analysed on the Nanodrop 2000 (Thermo Fisher Scientific) and Agilent 2200 TapeStation (D1000 screentape) systems. Only samples with RIN > 7 were used for library preparation. Libraries were prepared using a Lexogen QuantSeq FWD Kit (disease positioning) or the Illumina TruSeq stranded mRNA kit (tumour heterogeneity). Library quality and quantity were assessed using the 2200 TapeStation (Agilent) and Qubit (Thermo Fisher Scientific) systems. The libraries for the disease positioning were sequenced by Novogene Europe. The libraries for the tumour heterogeneity were run on the Illumina NextSeq 500 system using the high-output 75 cycle kit (2 × 36 cycle paired-end reads).

### Mapping of RNA-seq expression data

Quality checks and trimming on the raw RNA-seq data files were done using FastQC v.0.11.9 (https://www.bioinformatics.babraham.ac.uk/projects/fastqc/), FastP (v.0.20.1)[58], MultiQC (v.1.9)[59] and FastQ Screen (v. 0.14.0)[60]. RNA-seq single-end reads were mapped to the GRCm39.103 version of the *Mus musculus* genome and annotated[61] using STAR (v.2.7.8a)[62]. Expression levels were determined by FeatureCounts from the Subread package (v.2.0.1)[63].

### Computational disease positioning based on human TCGA data

TCGA data were downloaded using the GenomicDataCommons R package (v.1.12.0; https://bioconductor.org/packages/GenomicDataCommons)[64], TCGA 'HTSeq–counts' and corresponding clinical annotations. TCGA mutational data were downloaded using maftools (v.2.4.2)[65]. Both human and mouse RNA-seq counts were normalized using VST from the DESeq2 (v.1.28.1 and v.1.44.0) package[66] and then centred within a sample. Genes were reduced to those with direct one-to-one gene mapping between human and mouse genomes established by Ensembl, as retrieved from the biomaRt (v.2.56.1) package[67,68]. Singular-value decomposition (SVD) of the human data was performed followed by matrix factorization of both the human and mouse data into a 100-rank human space. UMAP of the combined dataset was executed using R package uwot (v.0.1.11; https://CRAN.R-project.org/package=uwot). An adjacency matrix was constructed from a nearest-neighbours search (RANN package v.2.6.1, https://CRAN.R-project.org/package=RANN) of the human and mouse SVD objects for clustering analysis. R package

igraph (v.1.2.11 and v.2.0.3)[69] was used to construct a graph object and the community structure was determined using Louvain •clustering.

Single-sample gene set enrichment analysis (ssGSEA) analysis was performed using the R package corto (v.1.2.4)[70] with the Hallmark gene set[71,72] downloaded using msigdbr (v.7.4.1; https://CRAN.R-project.org/package=msigdbr). Hoshida[27] and Chiang[28] (also downloaded using msigdbr) subclass classification was determined by the highest enriched subclass. Tumour immune cell estimation was performed using ConsensusTME[73].

Visualization of data by a combination of the ComplexHeatmap (v.2.4.3 and v.2.14.0)[74], ggplot2 (v.3.3.6 and v.3.5.1)[75], cowplot (v.1.1.1; https://CRAN.R-project.org/package=cowplot) and viridis[76] packages.

Human H&E-stained tissue sections were obtained from the TCGA collection (https://portal.gdc.cancer.gov/).

### Validation of HuMo clusters in an independent HCC cohort

HuMo clusters were validated with the bulk RNA-seq data of an independent cohort of 171 HCC samples from patients undergoing resection collected in the setting of the HCC Genomic Consortium[25] (European Genome-Phenome Archive: EGAS00001005364). Fastq files were aligned using STAR[62] (v.2.5.1b) to the hg19 reference genome with gencode annotation v19 and were quantified using featureCounts[63] (v.1.5.2). Raw counts were preprocessed and cluster attribution was performed as described above with the TCGA and mouse data. In the analysis of the transcriptomic data, positivity for previously reported gene signatures was evaluated using the Nearest Template Prediction[77] module from GenePattern (v.3.9)[78]. The ssGSEA projection[71] was performed using previously reported gene signatures as well as the Hallmark gene set downloaded using msigdbr (v.7.4.1). The mutational profile of 144 HCC samples was obtained by whole-exome sequencing[25]. Clinicopathological data (such as vascular invasion, AFP levels (≥400 ng ml$^{-1}$)) were originally reported previously[25].

### Differential expression analysis for intertumoural heterogeneity

Genes were restricted to those with significance in all comparisons (with significance defined as adjusted $P < 0.05$ and $\log_2[FC] > 1$). Data were scaled and visualized using the ComplexHeatmap[74] package. Gene Ontology over-representation analysis was performed using the clusterProfiler[79] package (v.3.16.1).

### Human sample ethical approval

The use of consenting patients' tissues surplus to diagnostic requirements for research purposes was approved by the Newcastle and North Tyneside Regional ethics committee, the Newcastle Academic Health Partners Bioresource (NAHPB) and the Newcastle upon Tyne NHS Foundation Trust Research and Development (R&D) department, in accordance with Health Research Authority guidelines. (10/H0906/41; NAHPB Project 48; REC 12/NE/0395; R&D 6579; Human Tissue Act licence 12534).

### MRI

Magnetic resonance imaging (MRI) scans were performed on liver-tumour-bearing mice using the nanoScan imaging system (Mediso Medical Imaging Systems). The mice were anaesthetized and maintained under inhaled isoflurane anaesthesia (induction, 4–5% (v/v); maintenance, 1.5–2.0% (v/v)) in 95% oxygen during the entire imaging procedure. Whole-body T1-weighted gradient echo (GRE) 3D coronal/sagittal MRI sequences (echo time (TE), 3.8 ms; repetition time (TR), 20 ms; flip angle, 30 degrees; and slice thickness, 0.50 mm) were used to obtain MRI images. For quantification of scans, volumes of interest were manually drawn around the liver region on MRI scans by visual inspection using VivoQuant software (v.4.0, InviCRO). For each scan, separate volumes of interest were prepared to adjust for the position and angle of each mouse on the MRI scanner and their tumour size.

## Mouse HCCO culture, drug screening and imaging

HCCOs were extracted and cultured as previously described[31,80], with the exception that HCCOs from mice with activated β-catenin signalling were cultured in the absence of WNT and RSPO1. All mouse HCCO cultures were regularly tested for mycoplasma.

For the high-throughput screen cohort 5 (BM) HCCOs were dissociated with TrypLE and plated at a density of $1 \times 10^3$ cells in 10 µl BME in prewarmed 384-well plates (Greiner BioOne, 781091) 5 days before adding the drugs. On day 0, a panel of 147 FDA-approved oncology drugs (AOD IX, acquired June 2019, https://dtp.cancer.gov/organization/dscb/obtaining/available_plates.htm) was added at a final concentration of 10 µM. Staurosporin was used as an internal positive control; DMSO and untreated cells were used as an internal negative control. The medium was changed on day 4 and the compounds were freshly added. Incucyte NucLight Rapid Red (Sartorius, 4717) was added on day 6 and cells were imaged using the Opera Phenix High-Content Screening System (Perkin Elmer) on day 9. Volumes were determined using Icy BioImage software (v.2.0.0.0; https://icy.bioimageanalysis.org)[81]. The experiment was performed twice (using different passages from one HCCO line) in technical quadruplicates.

For the drug dose–response curve screen, HCCOs (1–2 lines per cohort) were dissociated with TrypLE and plated at a density of $1 \times 10^3$ cells in 10 µl Matrigel (Corning, 356231) in prewarmed 96-well plates (Greiner BioOne, 655098). The treatment schedule was the same as for the HTP screen, except the medium was changed and fresh drugs were added on days 3 and 7. Drugs and concentrations are shown in the figures. Drugs were purchased from Selleckchem, dissolved in DMSO to 10 mM, aliquoted and stored at −20 °C. Cell viability was measured on day 9 using CellTitre-Glo 3D reagent (Promega, G9682) according to the manufacturer's instructions. Luminescence was measured on the Spark Microplate Reader (Tecan). The results were normalized to the vehicle. Curve fitting and $IC_{50}$ calculation were performed using a nonlinear regression equation. All of the experiments were performed in duplicate and at least three times using different passages from one to two HCCO lines per cohort.

Images of HCCOs were taken on an Olympus CKX41 using the Qimaging Retiga Exi Fast 1394 camera.

For immunofluorescence analysis, HCCOs were washed with ice-cold PBS, fixed with 4% PFA and permeabilized with 0.2% Triton X-100. A list of the antibodies used is provided in Supplementary Table 5. Images were taken using the Zeiss 710 confocal microscope.

Tumour-derived mouse HCCOs (available from all GEMMs) will be shared on reasonable request.

## Human HCCO culture and drug screening

Human HCCOs were derived from liver cancer needle biopsies or liver resections as described before[32]. The following human HCCO lines were used: D386-O and D953-O (*CTNNB1* WT, *TP53* WT, *MYC* WT); D455-O (*CTNNB1* MUT, *TP53* WT, *MYC* AMP); C948-O and C949-O (*CTNNB1* MUT, *TP53* WT, *MYC* AMP); C655-O (*CTNNB1* WT, *TP53* MUT, *MYC* ND); D045-O, D046-O, D803-O and R035-O (*CTNNB1* WT, *TP53* MUT, *MYC* WT); C798-O, C975-O, D324-O, D804-O and D876-O (*CTNNB1* WT, *TP53* MUT, *MYC* AMP); and D359-O (*CTNNB1* MUT, *TP53* MUT, *MYC* WT).

For expansion, the human HCCOs were seeded into reduced growth factor BME2 (R&D Systems, 3533-005-02) and cultured in expansion medium (EM): advanced DMEM/F-12 (Gibco, 12634010) supplemented with 1× B-27 (Gibco, 17504001), 1× N-2 (Gibco, 17502001), 10 mM nicotinamide (Sigma-Aldrich, N0636), 1.25 mM *N*-acetyl-L-cysteine (Sigma-Aldrich, A9165), 10 nM [Leu15]-gastrin (Sigma-Aldrich, G9145), 10 µM forskolin (Tocris, 1099), 5 µM A83-01 (Tocris, 2939), 50 ng ml⁻¹ EGF (Peprotech, AF-100-15), 100 ng ml⁻¹ FGF10 (Peprotech, 100-26), 25 ng ml⁻¹ HGF (Peprotech, 100-39), 10% RSPO1-conditioned medium (v/v, homemade). HCCOs were passaged after dissociation with 0.25% trypsin-EDTA (Gibco). All human HCCOs were regularly tested for

mycoplasma contamination using the MycoAlert mycoplasma detection kit (Lonza, LT07-118).

Drugs were purchased from ApexBio and Selleckchem, dissolved in DMSO to 10 mM, aliquoted and stored at −20 °C. For the screening, human HCCOs were dissociated with 0.25% trypsin-EDTA (Gibco) to single cells and $1 \times 10^3$ cells per well were plated in a 384-well plate (Greiner BioOne, 781986) on a layer of BME2 (R&D Systems, 3533-005-02) previously diluted with EM (50:50, v/v). Cells were cultured for 3 days without treatment to allow for organoid formation. At day 3, an eight-point half-log dilution series of each compound (ranging from 10 µM to 0.00316 µM) was added using a Tecan D300e. Cell viability was measured after 5 days of treatment using the CellTiter-Glo 3D reagent (Promega, G9682). Luminescence was measured on the Synergy H1 multi-mode reader (BioTek Instruments). Results were normalized to the vehicle (DMSO). The maximal DMSO concentration was 0.2%. Curve fitting was performed using Prism (GraphPad v.9 GraphPad Software) software and the nonlinear regression equation. Results are shown as mean ± s.e.m.

## Fluorescent activated cell sorting

After mincing into small pieces, 100 mg of healthy liver or liver tumour was digested on the gentleMACS Octo dissociator with heaters using the mouse tumour dissociation Kit (Miltenyi Biotec, 130-096-730). Dissociated cells were resuspended in 0.5% BSA in PBS, filtered through a 70 µm strainer and centrifuged at 400 g for 5 min. Cells were then resuspended in 5 ml RBC lysis buffer (Thermo Fisher Scientific, 00-4300-54) and incubated for 5 min at room temperature and washed with 0.5% BSA in PBS before being resuspended in 0.5% BSA in PBS. Cell suspensions were added to 96-well V-bottom plates (maximum density, $0.5 \times 10^6$ cells per well). Cells were stimulated for 3 h with complete IMDM medium containing 8% FCS, 50 µM β-mercaptoethanol, 1× penicillin–streptomycin with 1× cell activation cocktail with brefeldin A (BioLegend, 423304) at 37 °C as previously described previously[82]. After stimulation, cells were centrifuged at 800 g for 2 min. Cells were stained in Brilliant stain buffer (BD Biosciences, 566349) containing antibodies for surface antigens for 30 min at 4 °C in the dark. Cells were then washed with PBS/0.5% BSA, centrifuged at 800 g for 2 min, followed by ice-cold PBS and incubated with Zombie NIR Fixable Viability dye (BioLegend, 423106) to stain dead cells for 20 min at 4 °C. After further washing the cells with PBS/0.5% BSA, cells were fixed and permeabilized in FOXP3 transcription factor fixation/permeabilization solution (Thermo Fisher Scientific) for 20 min at 4 °C, according to the manufacturer's instructions. Intracellular antibodies were prepared in permeabilization buffer and incubated with cells for 30 min at 4 °C before cells were washed with permeabilization buffer, followed by PBS/0.5% BSA and finally resuspended in PBS/0.5% BSA. All of the experiments were performed using a five-laser BD LSRFortessa flow cytometer with DIVA software (BD Biosciences v.8.0.1). Compensation was determined automatically using Ultracomp eBeads (01-2222-42; Thermo Fisher Scientific). Data were analysed using FlowJo Software v.9.9.6.

## Statistics and reproducibility

Statistical analyses were performed using GraphPad Prism, software (v9 GraphPad Software) and R (v.4.0.2 and higher) with statistical tests as indicated in the figure legends. Data were tested for normal distribution. All performed *t*-tests were two-tailed. *P* values are displayed in figures. No statistical methods were used to predetermine sample sizes but our sample sizes are similar to those reported in previous publications[6,31,83–85]. For animal experiments, biological replicate sizes were chosen taking into account the variability observed in pilot and previous studies in respective cohorts. For all experiments, animal/sample assignment was matched for age-matched control, and assigned based on randomly assigned mouse identification markings. Batched staining and analyses alongside controls were used throughout. The investigators were not blinded for the in vivo experiments. Technical staff

administering therapy were blinded to the mouse genotypes. All subsequent tissue handling and analysis were blinded and/or performed using standardized automated analyses where possible. Quantitative image analysis was performed blinded to the genotype and treatment. The data distribution for normality testing and testing of equal variances was assessed using Prism 9 Software (GraphPad Software). No data were excluded, unless mentioned otherwise, except the following, which were excluded before analysis: one biological replicate failed quality control after transcriptomic sequencing—all other biological replicates from this and other cohorts successfully passed quality control and were included in downstream analysis; two drugs were excluded from the HCCO HTP screen due to microbiological contamination and drug precipitation in multiple replicates, respectively. One sample was excluded from the RFP expression analysis during analysis (total $n = 4$ biological replicates): testing AAV-mediated recombination of RFP alleles (Extended Data Fig. 1b,c), one sample was a notable outlier (4.9% versus 25.7%, 25.1% and 25.8%) which on re-review was caused by inconsistent RFP staining of the section—this outlier was removed from final analysis; details are provided in the figure legend also. Where the tumour number could not be quantified due to tumour rupture, no tumour number is reported (Fig. 5d).

Figures were assembled using Scribus v.1.4.8 (https://www.scribus.net/). Images were processed using Gimp v.2.10.14 (https://www.gimp.org/).

### Reporting summary

Further information on research design is available in the Nature Portfolio Reporting Summary linked to this article.

## Data availability

Data files for transcriptomic analyses are available at the Gene Expression Omnibus (GEO) under accession numbers GSE275444 and GSE273806 (mouse models) and GSE275443 (organoids). Our transcriptomic data are freely available to browse through a user-friendly interactive browser online, enabling HuMo classification of external transcriptomic datasets (http://shinyapps.crukscotlandinstitute.ac.uk/humo_app/). Immunohistochemical and H&E staining of the GEMMs is publicly available at the BioImage Archive under accession number S-BIAD1365. Montironi cohort data were provided on request to the original authors[25] and the TCGA data were accessed from publicly accessible databases[4]. The mouse genome (GRCm39.103) was accessed from Ensembl (https://www.ensembl.org). All data generated and/or analysed during the current study are also available from the corresponding authors on reasonable request. Source data are provided with this paper.

## Code availability

Scripts used for disease positioning is available at Code Ocean (https://codeocean.com/capsule/9804119/tree/v1).

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

**Acknowledgements** We thank the staff at the CRUK Scotland Institute's Core Facilities and, in particular, the Biological Services, Histology, Molecular Technology Services, Beatson Advanced Imaging Resource, Bioinformatics and Central Services, for their help; H. McKinnon for help with drug dosage studies; D. Athineos for support with animal work; C. Ficken for help with multiplex immunohistochemistry staining and imaging; V. M.-Y. Wang and C. Braconi for discussions and comments on the manuscript; C. Winchester for advice with Research Integrity and manuscript editing; and J.-C. Nault and J. Zucman-Rossi for reanalysis of pre-existing data. The results shown here are in part based on data generated by the TCGA Research Network (https://www.cancer.gov/tcga). AZD2171 was provided by S. Barry. M. Müller and T.G.B. were funded by the Wellcome Trust (grant number WT107492Z); T.G.B., D.A.M., H.L.R., J.M.L., M.H., M. McCain, A.G. and E.H.T. by an Accelerator Award from CRUK and Fondazione per la Ricerca sul Cancro (AIRC) and Fundación Científica de la Asociación Española Contra el Cáncer (FAECC) (grant number A26813); A.G. by a CRUK HUNTER Studentship (grant number A28221); S.D. and K.B. by CRUK (grant number, A29799); H.L.R., D.A.M. and M. McCain by CRUK centre

grant (grant numbers C9380/A18084) and programme grant (grant numbers C18342/A23390); S.M., T.J., N.P., S.D., G.L.S., C.K., C.N., G.C., W.C., K.B., O.J.S. and T.G.B. by CRUK core funding (grant numbers A17196, A31287 and CTRQQR-2021\100006); D.Y.L. by Cancer Research UK (grant number A25006); L.M.C. by Cancer Research UK (grant number A23983); T.G.B., H.L.R., D.A.M., L.M.C., J.L. and O.J.S. by CRUK (grant number 23390); D.A.M., L.M.C., O.J.S. and T.G.B. by Cancer research UK (grant number DRCRPG-Nov22/100007); I.R.P., L.O.-J., R.L.P. and J.L.Q. by the Mazumdar-Shaw Chair endowment; O.J.S. by Cancer Research UK (grant number A21139); H.H., C.J.M., K.B. and O.J.S. by the UKRI Mouse Genetics Network (grant numbers 21048 and 21039); H.H. and C.J.M. by Cancer Research UK (grant number A29801); H.L.R. by the NIHR (grant number 570556) and Cancer research UK (grant number A23390); J.M.L. by grants from the European Commission (Horizon Europe-Mission Cancer, THRIVE, 101136622), by the NIH (RO1-CA273932-01, RO1DK56621 and RO1DK128289), the Samuel Waxman Cancer Research Foundation, the Spanish National Health Institute (project PID2022-139365OB-I00, funded by MICIU/AEI/10.13039/501100011033 and FEDER), the Asociación Española Contra el Cáncer (Proyectos Generales, PRYGN223117LLOV; Reto AECC 70% Supervivencia, RETOS245779LLOV; Programa de Excelencia, EPAEC246711CLIN), the Generalitat de Catalunya (AGAUR, 2021-SGR 01347) and from "la Caixa" Foundation (agreement LCF/PR/SP23/52950009); and R.P. by the Fundació de Recerca Clínic Barcelona—IDIBAPS and a grant from the Spanish National Health Institute (MICINN, PID2022-139365OB-I00, funded by MICIU/AEI/10.13039/501100011033 and FEDER).

**Author contributions** M. Müller contributed to the conceptualization of the project, designed and performed experiments, supervised experiments, analysed data, created the figures and wrote the manuscript (original draft and subsequent editing). S.M. designed and performed in vivo experiments and analysed data and performed tumour genotyping. H.H. designed and performed the computational analysis of the disease positioning and created figures. T.J.K. provided histopathological analysis and advice. L.M. contributed to the design of mouse HCCO HTP experiments and performed and analysed mouse HCCO HTP experiments under L.M.C. supervision. L.B. and S.N. designed, performed and analysed experiments for human HCCO experiments under M.H.H. supervision. T.J. designed and performed in vivo DEN/ALIOS and MWD experiments under O.J.S. supervision. N.P. and S.D. designed and performed in vivo STZ experiments under K.B. supervision. T.S. performed immunophenotyping. J.H.-P., R.E.-F., A.G.-O. and R.P. performed the analyses on the independent patient cohort under J.M.L. supervision, and contributed discussions and manuscript writing. I.R.P., L.O.-J. and R.L.P. performed multiplex immunophenotyping under J.L.Q. supervision. G.L.S. designed and performed tumour genotyping and collated scanned images. J.L. designed and performed in vivo OT experiments under D.A.M. supervision. M.H. provided patient tissue and data. J.S. performed experiments and analysed data under T.G.B. supervision. G.M., A.M. and E.J. performed in vivo imaging under the supervision of D.L. who also contributed advice and resources. M. McCain and H.L.R. provided resources and discussions. J.H. analysed mouse HCCO data. C.K. and A.G. performed in vivo experiments under M. Müller and T.G.B. supervision. C.N., G.C. and W.C. performed experiments. R.S., M.N. and A.H. performed computational analysis. T.M.D. and E.H.T. assisted with in vivo experiments under M. Müller and T.G.B. supervision. D.J.M., D.A.M., K.B., M.H.H., L.M.C. and O.J.S. contributed discussion and provided resources. C.J.M. supervised computation analysis, contributed discussion and provided resources. T.G.B. contributed to the conceptualization of the project, designed and assisted with experiments, supervised, assisted in manuscript writing, edited the manuscript and acquired funding. All of the authors approved the final version.

**Competing interests** Material for the lenvatinib day 15 and day 30 timepoint experiments (Fig. 4h–j) was provided by Eisai. AZD2171 was provided by AstraZeneca. D.A.M. is a director, shareholder and employee of FibroFind. L.M.C. has consulted for Ono Pharmaceuticals UK on unrelated work. J.M.L. is receiving research support from Eisai Inc and Bayer Pharmaceuticals; is consulting and performing sponsored lectures for Eisai Inc., Merck, Roche, Genentech, AstraZeneca, Bayer Pharmaceuticals, AbbVie, Sanofi, Moderna, Glycotest and Exelixis; and participates in the Data Safety Monitoring Board for Bristol Myers Squibb. The other authors declare no competing interests.

**Additional information**
**Correspondence and requests for materials** should be addressed to Miryam Müller or Thomas G. Bird.

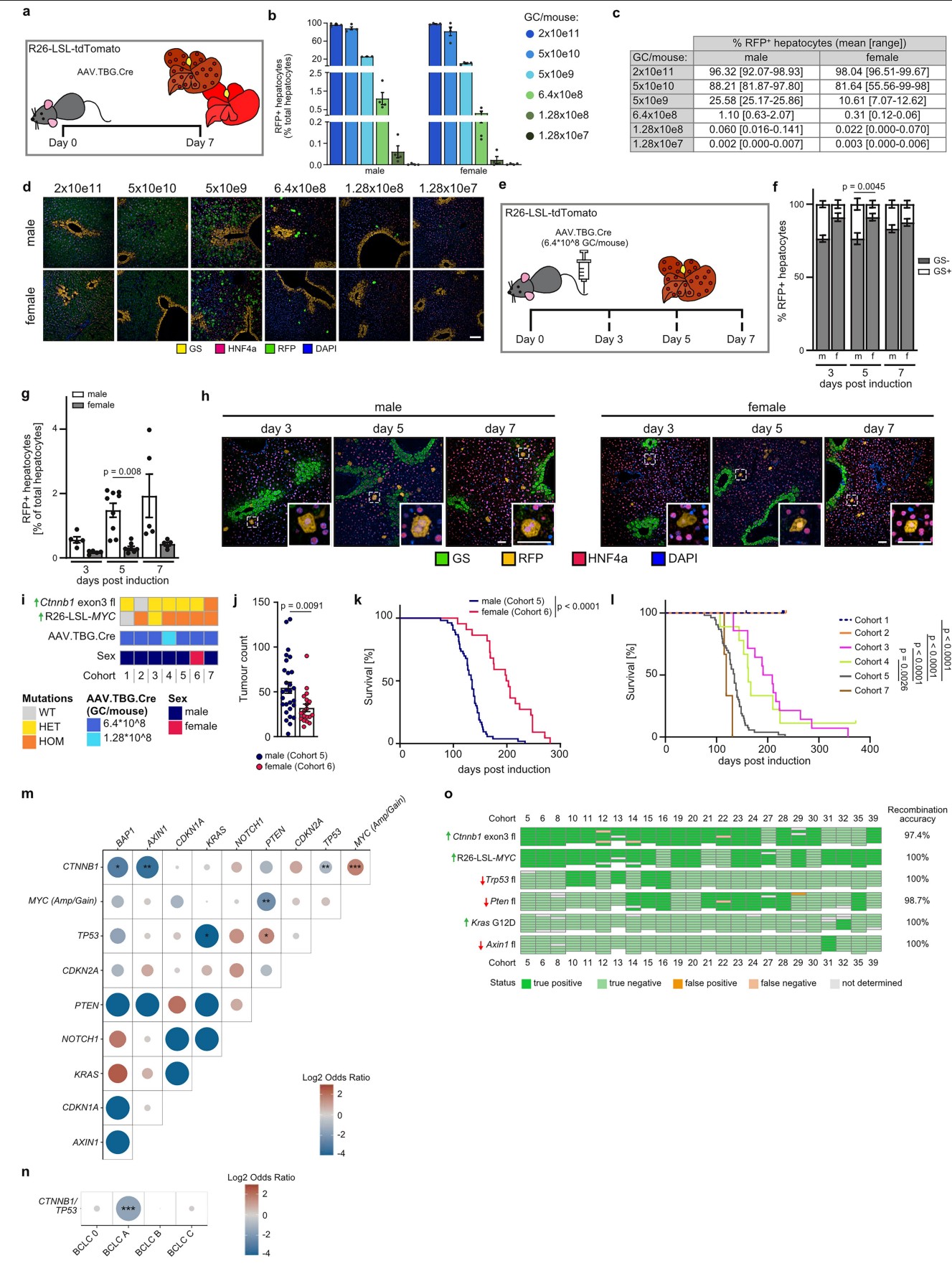

**Extended Data Fig. 1** | See next page for caption.

**Extended Data Fig. 1 | Justification for and validation of the genetics and genetic induction approach employed in the suite of GEMMs.** (**a**) Dose finding for clonal induction using AAV.TBG.Cre in a R26-LSL-tdTomato reporter mouse model. Experimental scheme for **b-d**. (**b-d**) Decreasing doses of AAV.TBG.Cre lead to decreased recombination of the LSL and thus less RFP+ hepatocytes (**b-c**) Quantification of RFP+ hepatocytes by sex and dose. n = 4 mice throughout except n = 3 for $5 \times 10^9$ female (single sample excluded due to inconsistent RFP staining). Data shown as mean ± s.e.m.. GC = genomic copies. (**d**) Representative images of immunofluorescent staining demonstrating exclusive and dose-dependent targeting of hepatocytes by AAV.TBG.Cre. Individual Channels for the zone 3 marker glutamine synthetase (GS, yellow), HNF4a (magenta), RFP (green), DAPI (blue). Scale bar equals 100 µm. (**e**) AAV.TBG.Cre sex-dependent clonal induction variation over time. Experimental scheme for **f-i**. (**f**) Quantification of RFP⁺ hepatocytes using GS shows clonal induction within zone 3 and outside zone 3 but no significant zonal expansion over time using a dose of $6.4 \times 10^8$ GC/mouse. n = 5 (male d3 + d7, female d3 + d7), 8 (female d5), 9 (male d5). Data shown as mean ± s.e.m. Two-way ANOVA with Tukey correction (**g**) Male mice recombine at a higher rate than female mice after induction with $6.4 \times 10^8$ GC/mouse with no additional residual recombination from 5 to 7 days post induction. n = 5 (male d3 + d7, female d3 + d7), 8 (female d5), 9 (male d5). Data shown as mean ± s.e.m., Kruskal-Wallis test with Dunn's correction (**h**) Representative images of Cre-driven recombination rates in males and females on d3, d5, and d7 post induction; GS (green), HNF4a (magenta), RFP (yellow) and DAPI (blue). Scale bar equals 50 µm. (**i**) Summary of mouse cohorts used in **j-l**. (**j**) A lower induction rate in females leads to a lower tumour burden compared to males with the same mutational background. n = 28 (Cohort 5) and 19 (Cohort 6) mice. Data shown as mean ± s.e.m. Unpaired two tailed t-test. (**k**) Lower tumour burden due to a lower induction rate causes a prolonged survival in female mice compared to males with the same mutational background. n = 53 (male, Cohort 5) and 22 (female, Cohort 6) mice. Log rank test. (**l**) Mutational burden and induction dose influence tumour penetrance and survival outcomes. n = 11 (Cohort 1), 8 (Cohort 2), 14 (Cohort 3), 9 (Cohort 4), 53 (Cohort 5 – same data as k), 3 (Cohort 7). Log rank test. All panels: GC = genomic copies. Please note that individual cohort survival data shown for Cohort 5 and 6 are also shown in Extended Data Fig. 2a to allow direct comparison with data in that figure. (**m**) Analysis of the TCGA PanCancer dataset shows odds ratio for co-occurrence and mutual exclusivity of modelled HCC driver genes. n = 353. One-sided Fisher Exact Test, */**/*** denote p ≤ 0.05/0.01/0.001 respectively. (**n**) Mutual exclusivity of drivers is fluid and can change depending on tumour stage as shown for *CTNNB1* and *TP53;* reanalysis of data from Nault et al.[86]. n = 73 (BCLC 0), 404 (BCLC A), 157 (BCLC B), 101 (BCLC C). One-sided Fisher Exact Test. (**o**) Genotyping of bulk end stage tumour in representative cohorts. Genotyping by cohort and allelic recombination within individual mice (bars) – representative mice are aligned between alleles within cohorts. Comparison is made to known colony genotype testing presence/absence of allelic recombination to prediction displayed as false/true positives/negatives. Recombination accuracy denotes true positive + true negative/positive + negative. *Cdkn1a* and *Cdkn2a* were constitutive knockouts in the relevant colonies.

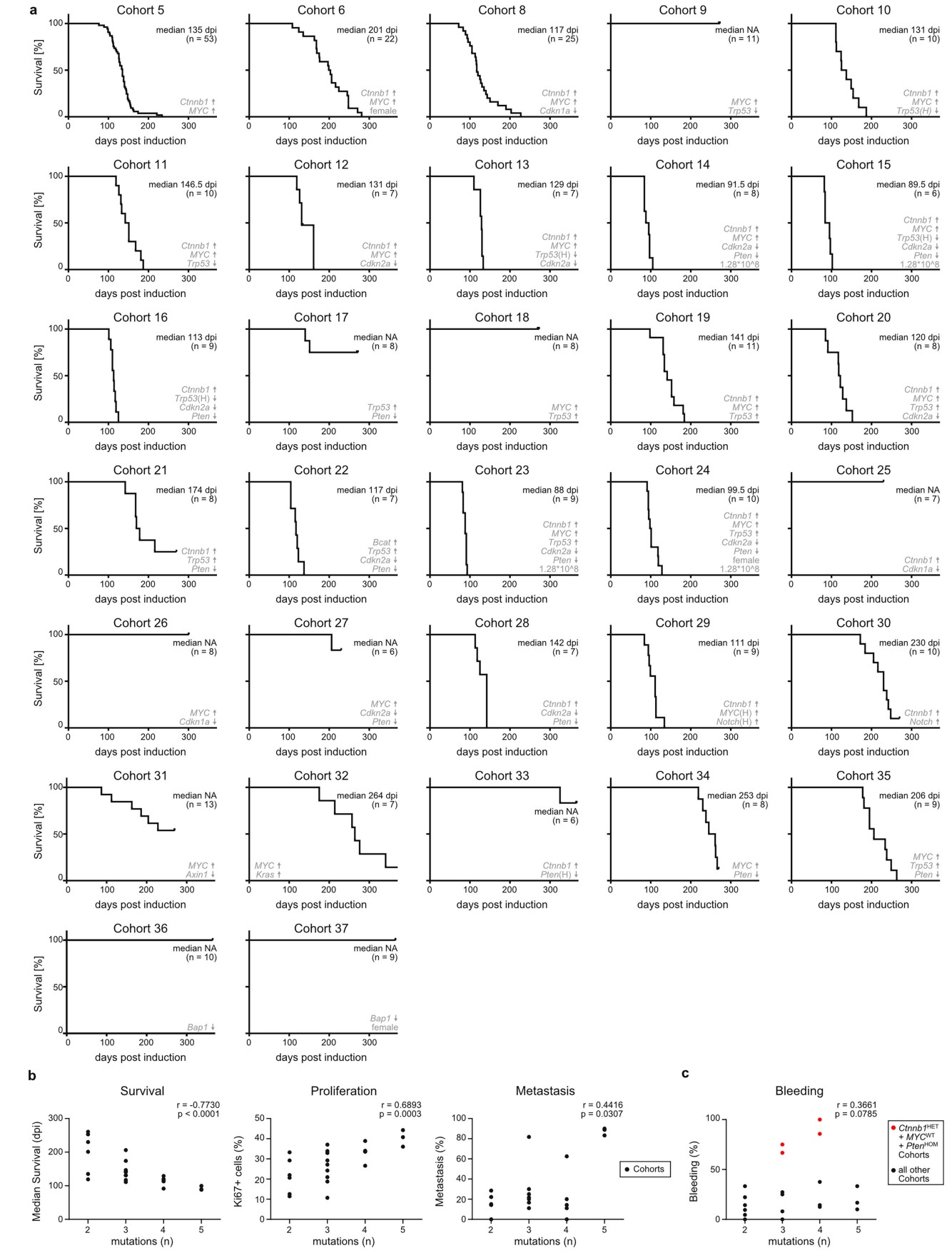

**Extended Data Fig. 2 |** See next page for caption.

**Extended Data Fig. 2 | Endpoint survival and tumour penetrance varies depending on co-occurrence of mutations.** (**a**) Detailed survival data for summary data shown in Fig. 1b. Median survival reported as time after AAV. TBG.Cre induction in days (dpi). Number of mice used per cohort is shown in the Figure. All cohorts except Cohort 6, 24, and 37 are male mice. Unless otherwise specified mice were induced with $6.4*10^8$ GC/mouse. (H) indicates heterozygosity of an otherwise homozygous allele. (**b**) Correlation analysis of mutational burden with survival, tumour proliferation, and metastasis. n = 7 (2 Mutations), 9 (3 Mutations), 5 (4 Mutations), 3 (5 Mutations). Spearman Rank Test (two-tailed). (**c**) Correlation analysis of mutational burden and bleeding. n = 7 (2 Mutations), 9 (3 Mutations), 5 (4 Mutations), 3 (5 Mutations). Spearman Rank Test (two-tailed). Please note that survival data shown for Cohort 5 and 6 are also shown in Extended Data Fig. 1l to allow direct comparison with data in that figure.

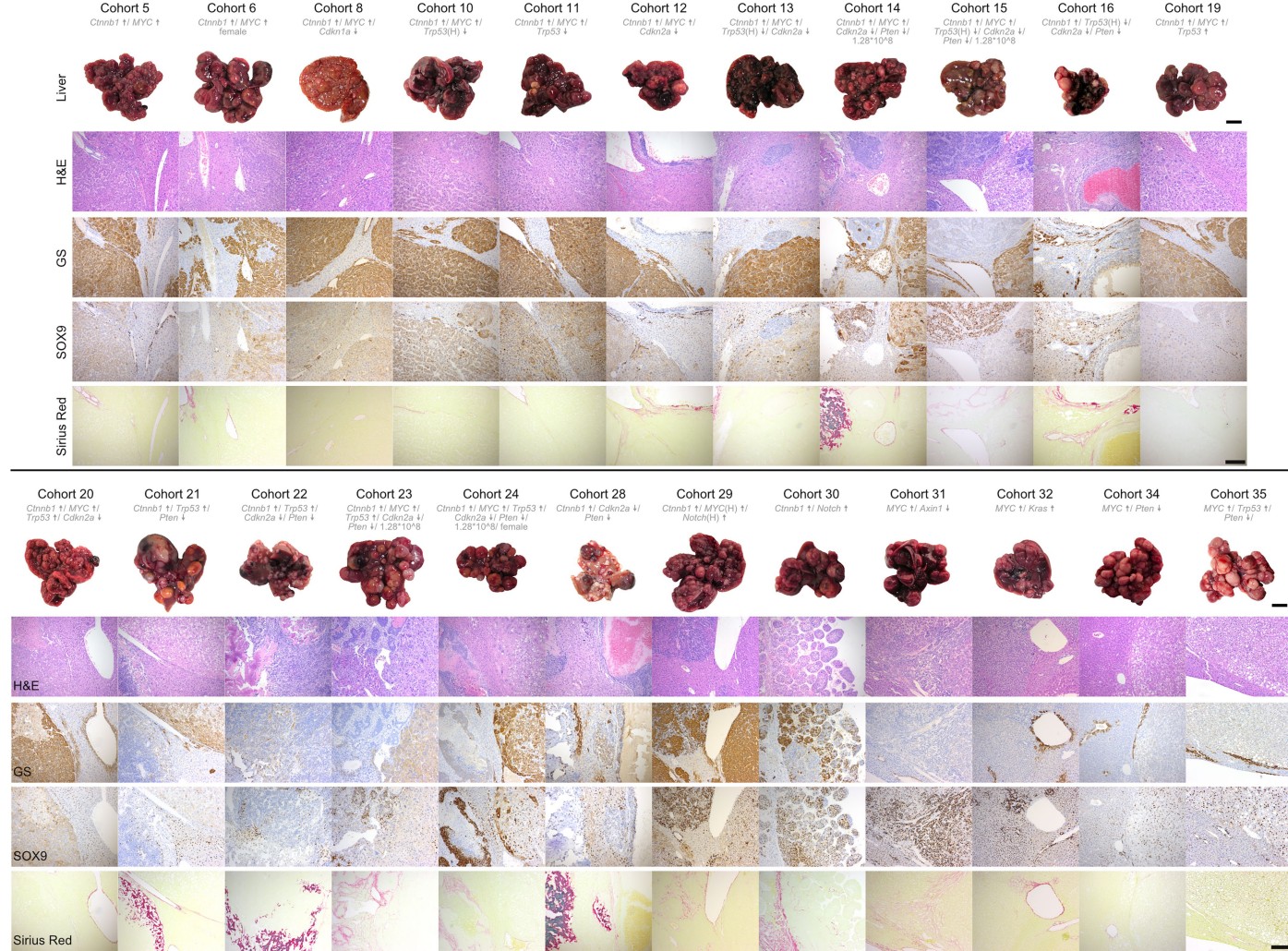

**Extended Data Fig. 3 | Macroscopic and microscopic tumour nodule phenotype reflect the heterogeneity of human HCC.** Representative images for each Cohort with staining for general morphology (H&E), glutamine synthetase (GS) for activated beta-catenin signalling, SOX9 as a progenitor marker, and Sirius Red as an indicator for extracellular matrix content in the tumours. All cohorts except Cohort 6 and 24 are male mice. Unless otherwise specified mice were induced with $6.4*10^8$ GC/mouse. (H) indicates heterozygosity of an otherwise homozygous allele. Scale bar equals 1 cm (macroscopic) or 200 µm (microscopic). Macroscopic images and microscopic of H&E and GS for Cohorts 5, 19, 23, 28, 30, and 35 are the same as in Fig. 1c and are shown here to allow direct comparison with data in this figure. Scanned whole liver lobes across biological replicates from each cohort are available via BioImage Archive (https://www.ebi.ac.uk/) via accession number S-BIAD1365.

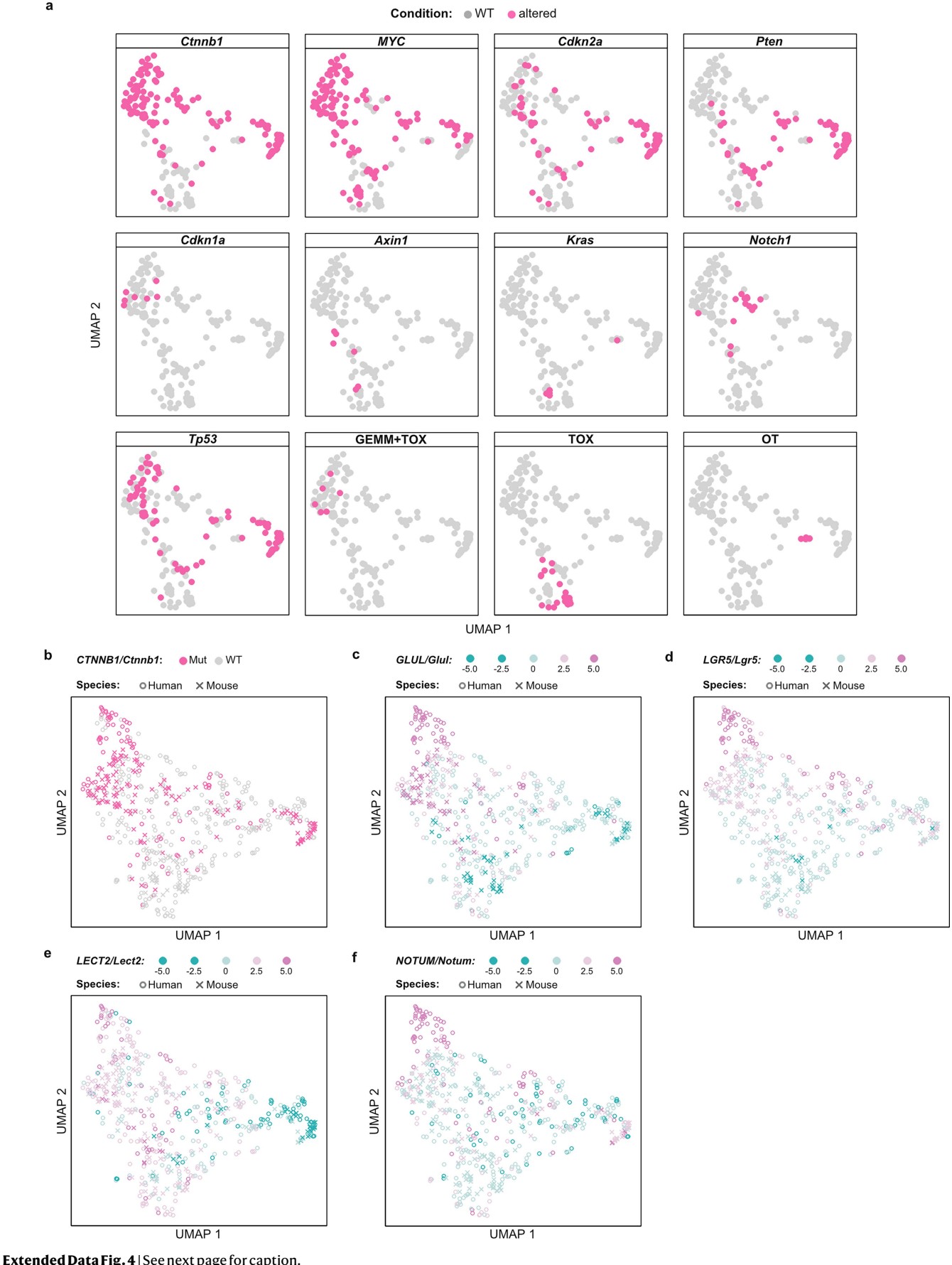

**Extended Data Fig. 4** | See next page for caption.

**Extended Data Fig. 4 | Mutational status alone does not explain cluster association.** (**a**) UMAP plots showing distribution of mouse cohorts by specific genetic alterations, carcinogen treatment (TOX), or orthotopic transplant (OT). (**b-f**) Samples with mutated *CTNNB1/Ctnnb1* (**b**) are spread over the whole UMAP spectrum, whereas samples with expression of beta-catenin pathway downstream targets *GLUL/Glul* (**c**), *LGR5/Lgr5* (**d**), *LECT2/Lect2* (**e**), and *NOTUM/Notum* (**f**) are confined to the upper left quadrant.

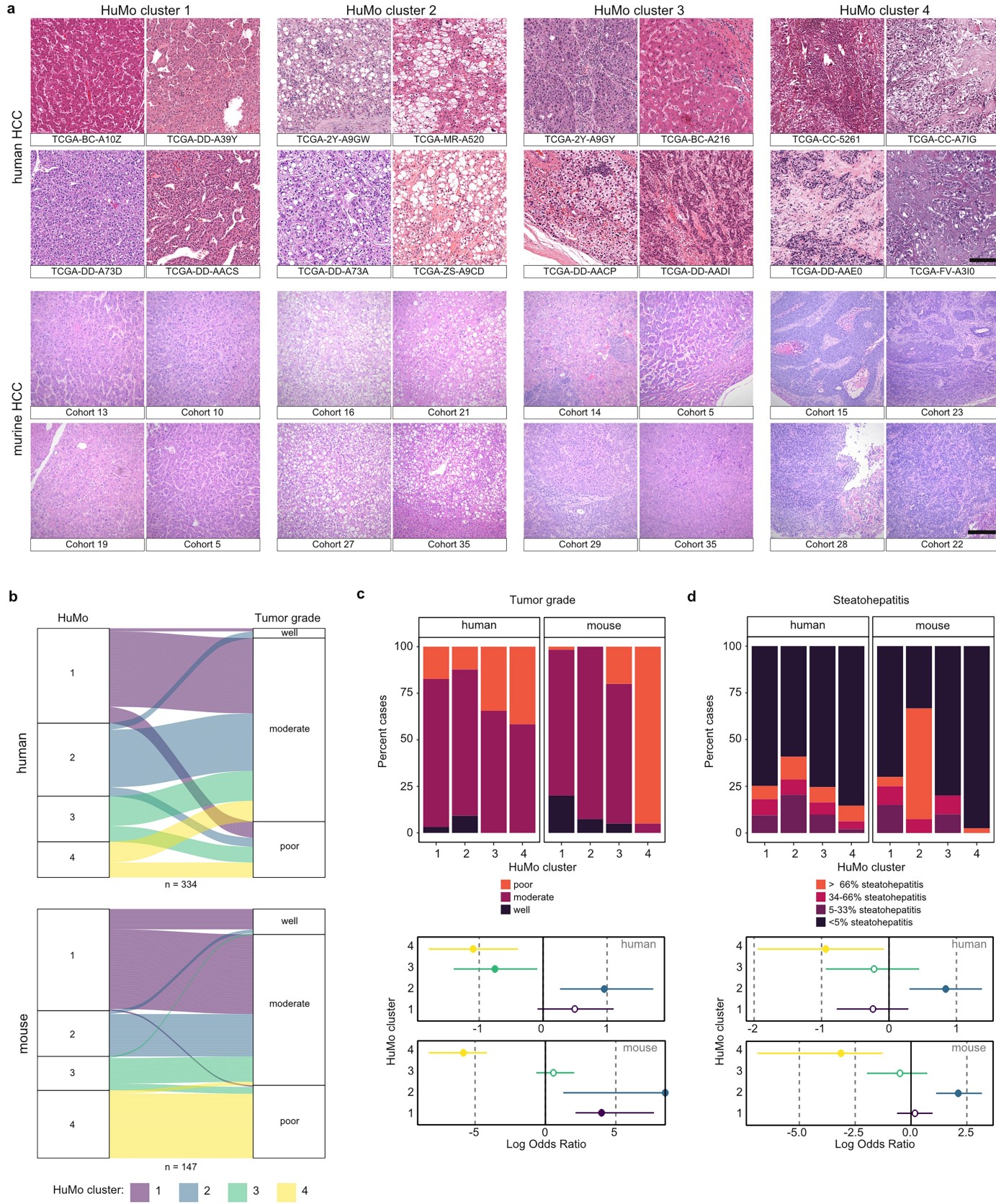

**Extended Data Fig. 5 | Histopathological features in mice and human HCC within the same cluster are comparable.** (**a**) Transcriptional subclasses translate to similar histopathologies between human and mouse liver samples from the same HuMo clusters as shown by H&E staining; representative images from n = 4 biological replicates in each HuMo shown. Scale bars equal 200 µm. (**b-d**) Analysis of tumour grade (**b** + **c**) and steatohepatitis (**d**) highlights the resemblance of human (n = 334) and mouse (n = 147) HCC within the same HuMo cluster. Dots/bars as log odds ratio/95% CIs respectively; Fisher test, >0.05 (open circles) and ≤0.05 (closed circles). Scanned whole liver lobes across biological replicates from each cohort are available via BioImage Archive (https://www.ebi.ac.uk/) via accession number S-BIAD1365.

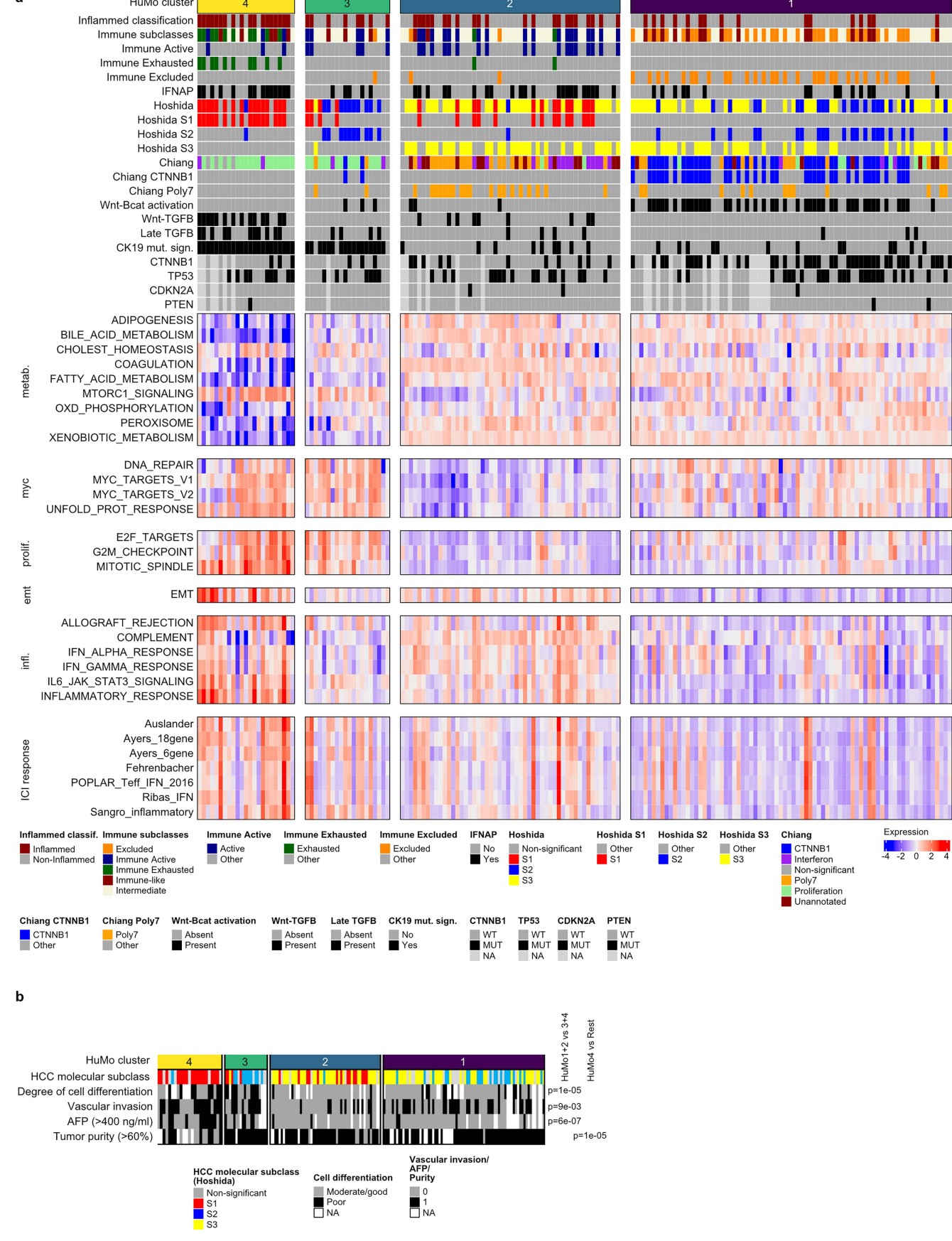

**Extended Data Fig. 6** | See next page for caption.

**Extended Data Fig. 6 | Independent human HCC dataset validates HuMo cluster classification.** (**a** + **b**) Validation of the HuMo cluster classification system with a previously published, independent human HCC dataset[25] shows similar distribution dynamics as for the TCGA data set. HuMo cluster 1 was enriched for immune-evasive signatures including the IFNAP signature[26], HuMo cluster 2 had higher inflammatory signalling signatures and was enriched for immune-active tumours, and HuMo clusters 3 and 4 featured a strong progenitor signature (CK19 mutation signature) consistent with the previously observed histological phenotype of these clusters. Only HuMo cluster 4 was significantly enriched for the inflamed HCC class with an immune-exhaustion signature and displays the lowest tumour purity, meaning higher fraction of stroma/immune infiltrate. Significant statistical comparisons between HuMos in a); Immune classification - 4 v.s rest, 3E-05; Immune exhausted – 4 vs. rest, 1E-08; Immune excluded – 1 vs rest 4E-10; IFNAP - 1 vs rest 3E-03; Hoshida[27] S1 - 4 vs rest 2E-09; Hoshida S2 - 3 vs rest 9E-04; Hoshida S3 1 + 2 vs rest 1E-10; Chiang CTNNB1 – 1 vs rest 5E-15; Wnt-βcat activation - 1 vs rest 1E-13, Wnt-TGFβ – 4 vs rest 3E-09; CTNNB1 - 1 vs rest 1E-07. Continuous variables were compared using Two-tailed student T-test (normally distributed, equal variance), Welch's T-test (normally distributed, unequal variance) or Wilcoxon-rank sum test (not normally distributed data). Categorical variables were compared using Fisher's exact test. Statistical comparisons related to Extended Data Fig. 7 are shown separately in Supplementary Table 2.

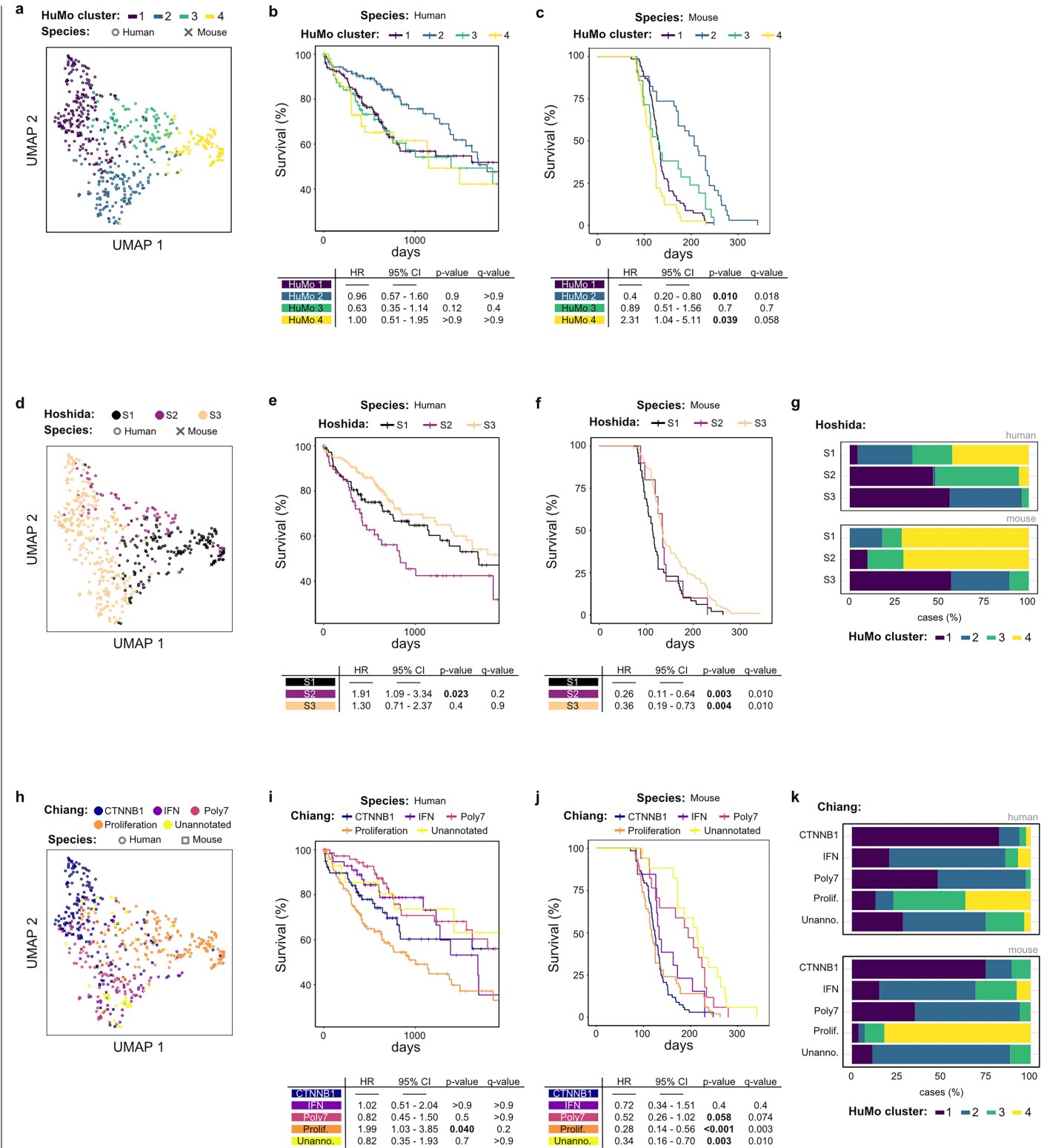

**Extended Data Fig. 7 | Conventional clinically used classifications of human hepatocellular carcinoma fail to cluster mouse and TCGA HCC data sets distinctly.** (**a**) UMAP visualization of data sets by HuMo cluster (**b** + **c**) Survival of HCC patients (n = 396) and mice (n = 165) by HuMo cluster. (**d**) UMAP visualization of data sets by Hoshida[27] subgroups. (**e** + **f**) Survival of HCC patients (n = 396) and mice (n = 165) by Hoshida subgroup. (**g**) Classification comparison (Hoshida vs HuMo cluster) by species. (**h**) UMAP visualization of

data sets by Chiang[28] subgroups. (**i** + **j**) Survival of HCC patients (n = 396) and mice (n = 165) by Chiang subgroup. (**k**) Classification comparison (Chiang vs HuMo cluster) by species. Survival analysis performed using univariate Cox Proportional-Hazards model reporting hazards ratio (HR), confidence intervals (95% CI), the p value was corrected for multiple testing (q value) using the false discovery rate method. UMAP in (**a**) is the same as in Fig. 2d and is shown here to allow direct comparison with data in this figure.

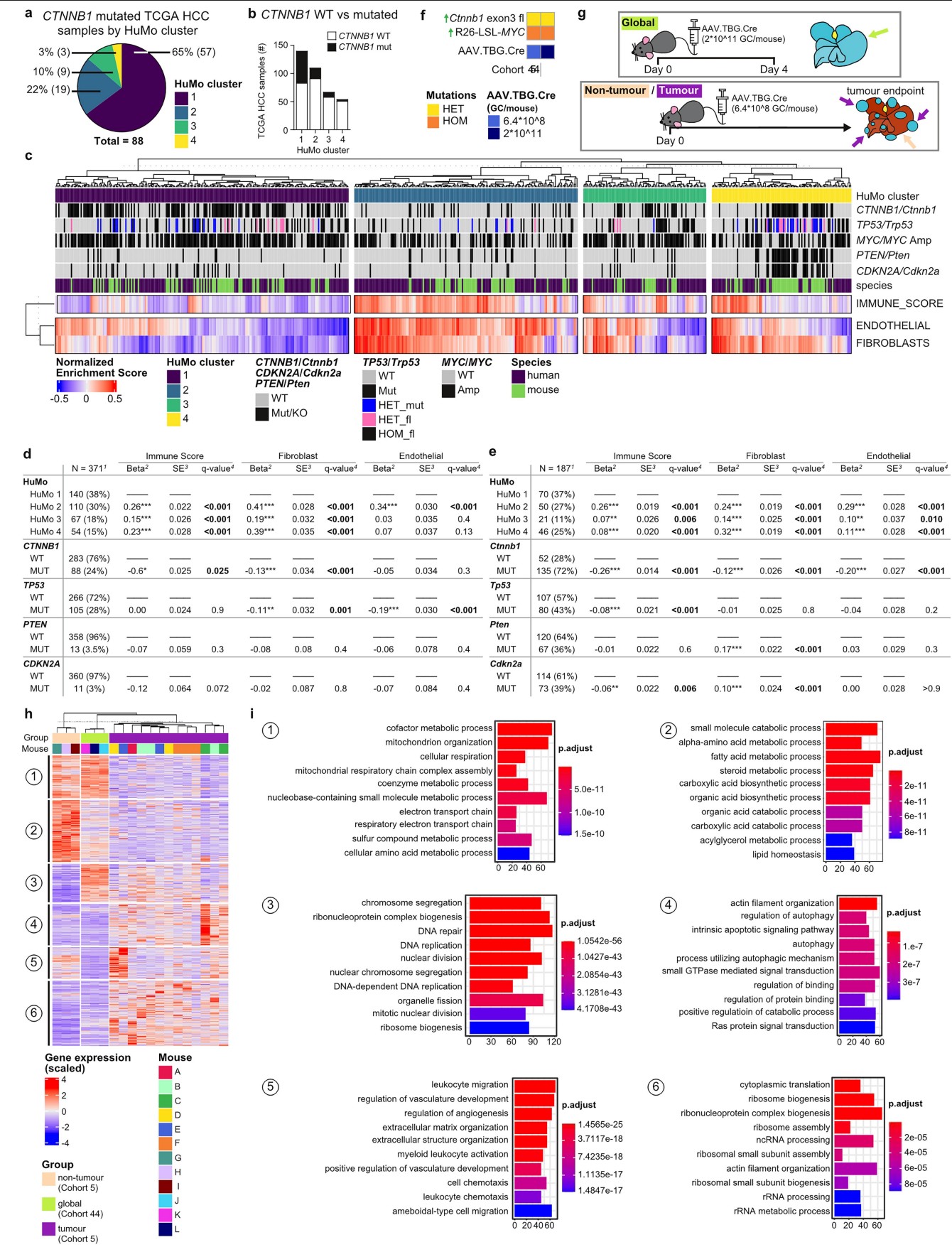

**Extended Data Fig. 8 |** See next page for caption.

**Extended Data Fig. 8 | HuMo cluster 1 tumours display immune-paucity whilst a representative HuMo cluster 1 model displays mild inter-tumoural heterogeneity.** (**a**) Percentage (total number) of human TCGA HCC samples with mutations in *CTNNB1* associated with each HuMo cluster. (**b**) Ratio of wild type and mutated *CTNNB1* in human TCGA HCC samples within each HuMo cluster. (**c**) Immune-pathway analysis shows a clear association of HuMo cluster 1 with immune paucity, whereas HuMo cluster 2 shows the highest association with immune-cell enrichment. (**d** + **e**). Correlation analysis of immune score, fibroblast, and endothelial signatures with HuMo clusters and mutations in human (d) and mouse (e) emphasizes the negative enrichment for immune score in HuMo cluster 1 and *CTNNB1* mutated samples in both species. (**f**) Summary of cohorts used in this figure; cohort 5 = BM. All mice used in this figure were male. (**g**) Experimental scheme for samples used in **h** + **i**. (**h**) Heatmap of differentially expressed genes between liver tissue from mice with global hepatocyte induction of altered genes and, non-tumour and tumour, tissue from mice with clonal hepatocyte induction of altered genes. Tumour tissue, despite induction of the same genetic alterations, differs greatly from the global induction group suggesting evolution of induced clones to develop tumours. n = 3 (global and non-tumour), 6 mice with up to 3 tumour samples per mouse (tumour). (**i**) Gene Ontology over-representation analysis shows upregulation of biological processes associated with oncogenesis in tumour tissue. One-sided Fisher's exact test; adjusted using the Benjamini-Hochberg procedure.

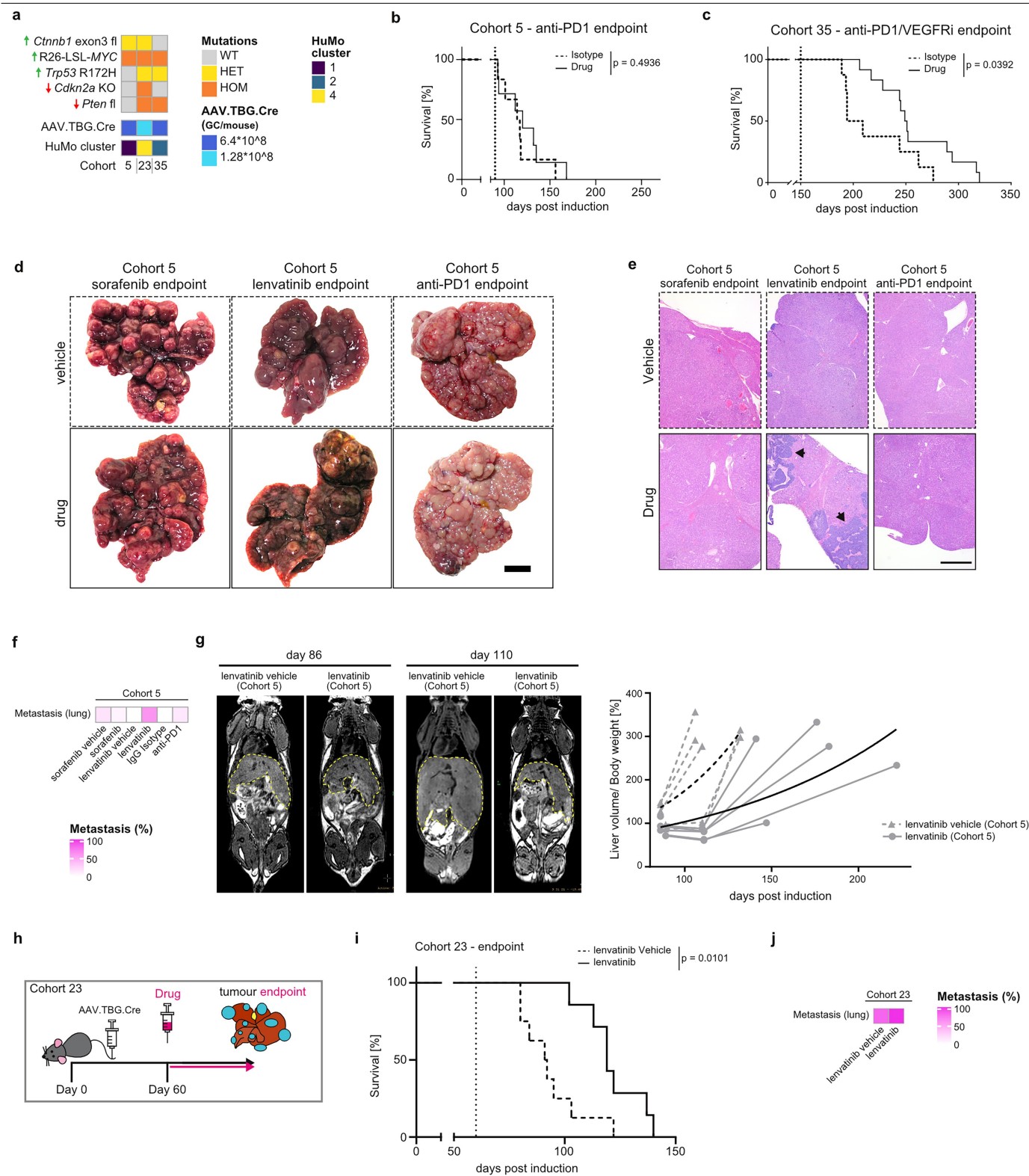

**Extended Data Fig. 9** | See next page for caption.

**Extended Data Fig. 9 | Treatment with the tyrosine kinase inhibitor lenvatinib leads to phenotypic changes and increased metastasis.**
(**a**) Summary of cohorts used in **b-j**. All mice used in this figure were male.
(**b**) Treatment with immune checkpoint inhibitor anti-PD1 (200 µg/mouse, ip) does not significantly improve survival in a mouse model representative of HuMo cluster 1. Dotted vertical line indicates treatment start. n = 6 (IgG Isotype), 7 (anti-PD1). Log rank test. (**c**) Treatment with immune checkpoint inhibitor anti-PD1 (200 µg/mouse, ip) and VEGFRi (3 mg/kg, oral) significantly improves survival in a mouse model representative of HuMo cluster 2. Dotted vertical line indicates treatment start. n = 8 (Vehicle/IgG Isotype), 12 (VEGFi/anti-PD1). Log rank test. (**d**) Macroscopic liver images of drug and vehicle treated Cohort 5 (BM) mice at endpoint. Scale bar equals 1 cm. (**e** + **f**) Treatment with lenvatinib, but not sorafenib or anti-PD1, results in a more aggressive tumour morphology (indicated by black arrows) and increased number of mice with detectable metastasis at endpoint in Cohort 5 (BM) mice. Scale bar equals 1 mm. n = 17 (sorafenib vehicle), 13 (sorafenib), 5 (lenvatinib vehicle + lenvatinib + IgG Isotype), 7 (anti-PD1) mice. (**g**) Non-invasive magnetic resonance imaging of Cohort 5 (BM) mice reveals delayed tumour growth in lenvatinib treated mice with liver volume as a proxy for tumour burden. n = 5 (lenvatinib Vehicle), 6 (lenvatinib). (**h**) Treatment scheme for **i-j** with drug given from d60 post induction to accommodate for faster model progression (**see** Extended Data Fig. 2a). (**i**) Lenvatinib treatment improves endpoint survival in a representative GEMM of HuMo cluster 4 (Cohort 23, BM + $Pten^{fl/fl}$ + $TrpS3^{R172H/wt}$ + $Cdkn2a^{KO/KO}$). Dotted vertical line indicates treatment start. n = 8 (lenvatinib vehicle), 7 (lenvatinib). Log rank test. (**j**) Cohort 23 (BM + $Pten^{fl/fl}$ + $TrpS3^{R172H/wt}$ + $Cdkn2a^{KO/KO}$) mice treated with lenvatinib have increased number of mice with detectable metastasis at endpoint. n = 6 (lenvatinib vehicle), 7 (lenvatinib).

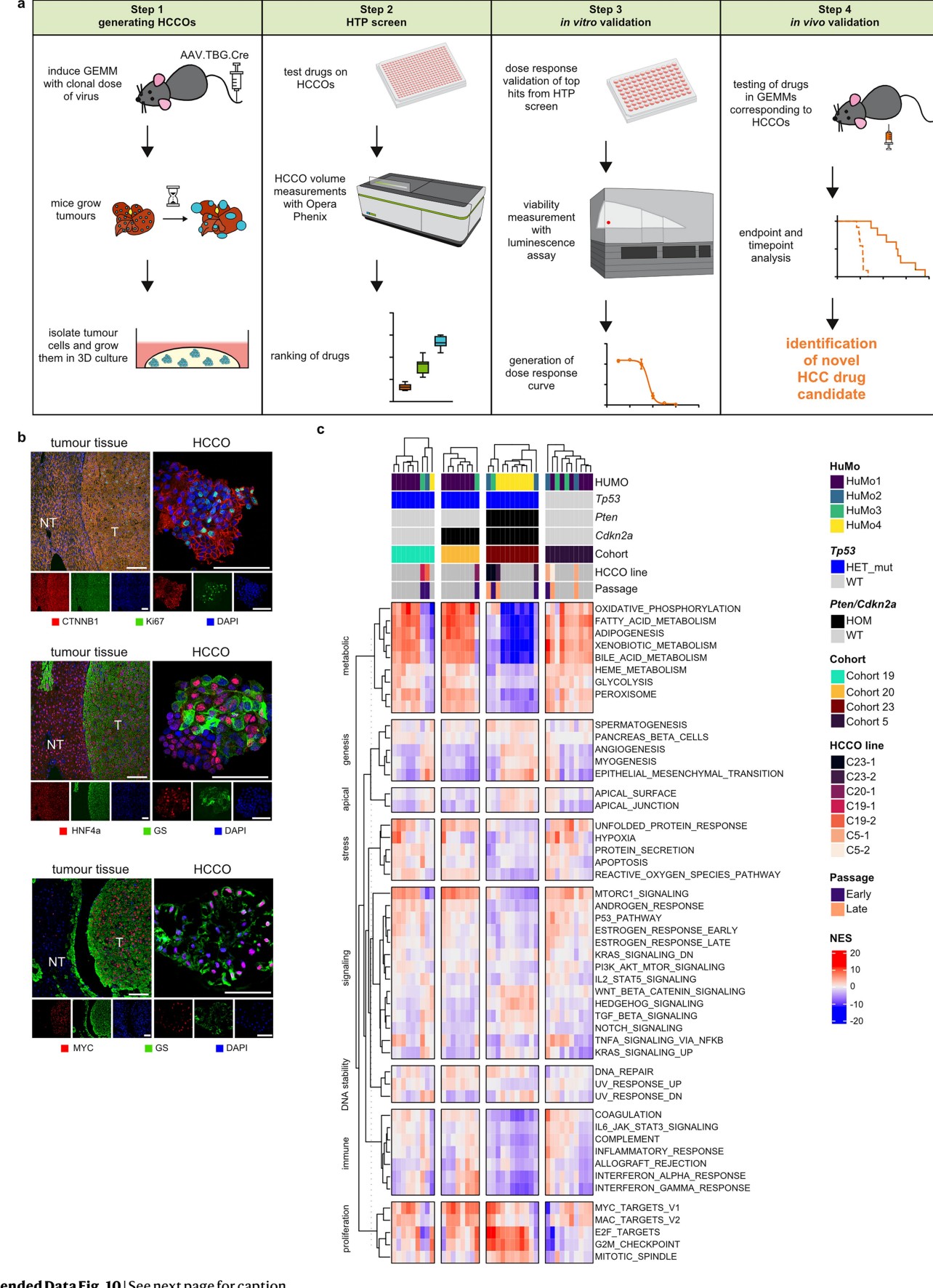

**Extended Data Fig. 10** | See next page for caption.

**Extended Data Fig. 10 | HCCOs mimic features of the primary tumour and are suited to investigate drug effects on tumour cells.** (**a**) Schematic of murine HCCO assay pipeline. HTP = high-throughput, GEMM = genetically-engineered mouse model, HCC = hepatocellular carcinoma, HCCOs = HCC organoids. (**b**) HCCOs keep characteristics of primary tumour tissue such as accumulation of beta-catenin (CTNNB1), proliferation marker Ki67, differentiation marker HNF4a, glutamine synthetase (GS) and MYC expression; representative images of single organoids from a bulk culture from a single organoid line and of single tumours from multiple autochthonous tumours within a single mouse. Scale bars equal 100 μm. NT = non-tumour, T = tumour. (**c**) The transcriptional phenotype of HCCOs differed from the original tumours of the same Cohort, likely due to the simplified nature of HCCOs as an epithelial-cell-only model as well adaptive response to the culture conditions. Note that all GEMMs and HCCOs are mutated/overexpressing of both CTNNB1/Myc respectively.

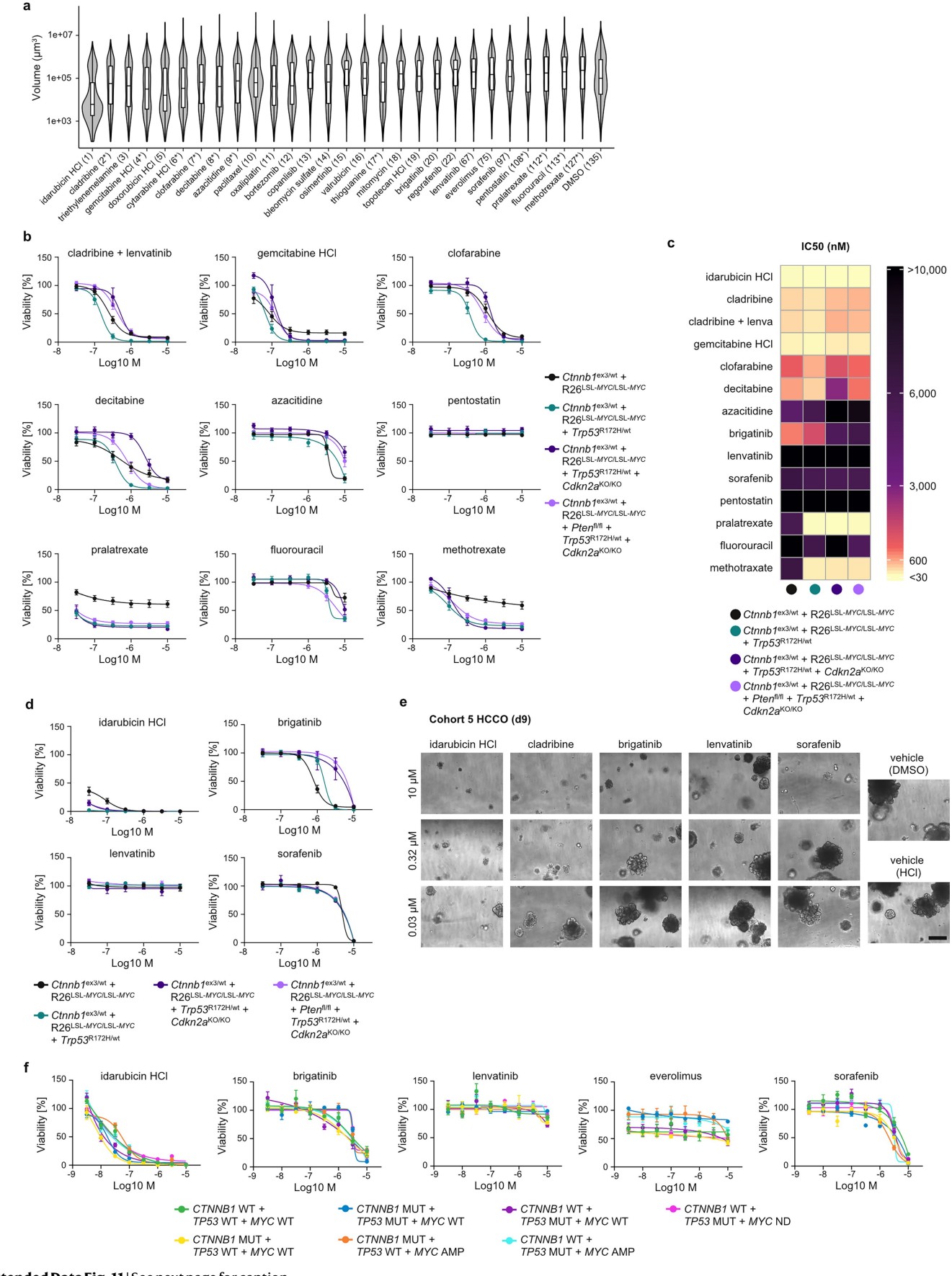

**Extended Data Fig. 11 |** See next page for caption.

**Extended Data Fig. 11 | A high-throughput tumouroid assay pipeline identifies anti-cancer drugs for repurposing as potential HCC therapy.** (**a**) Volumetric measurements of HCCOs after 9 d treatment with indicated drugs; merged data from 4 technical replicates in each of two plates per condition. Ranking position in parenthesis. Nucleobase/Nucleoside analogues indicated by asterisks. Box centre=median with box bounding 25–75th centiles, the upper whisker extend +/− 1.5*IQR from the hinge. N = 2421/2259/2748/ 1700/2172/2314/1555/2964/2383/3840/3192/1820/2876/4491/1561/1624/ 18/45/1608/2408/3564/1649/2066/2538/1911/2273/2008 organoids for Azacitidine/Bleomycin sulfate/Bortezomib/Brigatinib/Cladribine/ Clofarabine/Copanlisib/Cytarabine hydrochloride/Decitabine/DMSO/ Doxorubicin hydrochloride/Everolimus/Gemcitabine hydrochloride/ Idarubicin hydrochloride/Lenvatinib/Methotrexate/Mitomycin/Osimertinib/ Oxaliplatin/Paclitaxel/Regorafenib/Sorafenib internal/Thioguanine/ Topotecan hydrochloride/Triethylenemelamine/Valrubicin respectively.

(**b** + **c**) Testing a wide variety of antimetabolites demonstrates a drug-specific on-target effect for antimetabolites in the same subclass as cladribine. No synergy between cladribine and lenvatinib was observed in HCCOs. n = 3 (different passages from one to two HCCO lines per named mouse cohort, technical duplicates; black/green/blue/magenta = HCCOs originated from tumours of cohort 5/19/20/23 respectively). Data shown as mean ± s.e.m. (**d**) In vitro dose-dependency testing of drug efficacy in murine HCCOs validates results from screen. n = 3 (different passages from one to two HCCO lines per named mouse cohort, technical duplicates). Data shown as mean ± s.e.m. (**e**) Representative images of dose-dependent drug effects on murine HCCOs after 9 days of treatment. Scale bar equals 200 μm. (**f**) In vitro dose-dependency testing of drug efficacy in human HCCOs validates results from screen. n = 3 (different passages from one to five human HCCO lines per driver combination, see methods for details, technical duplicates). Data shown as mean ± s.e.m.

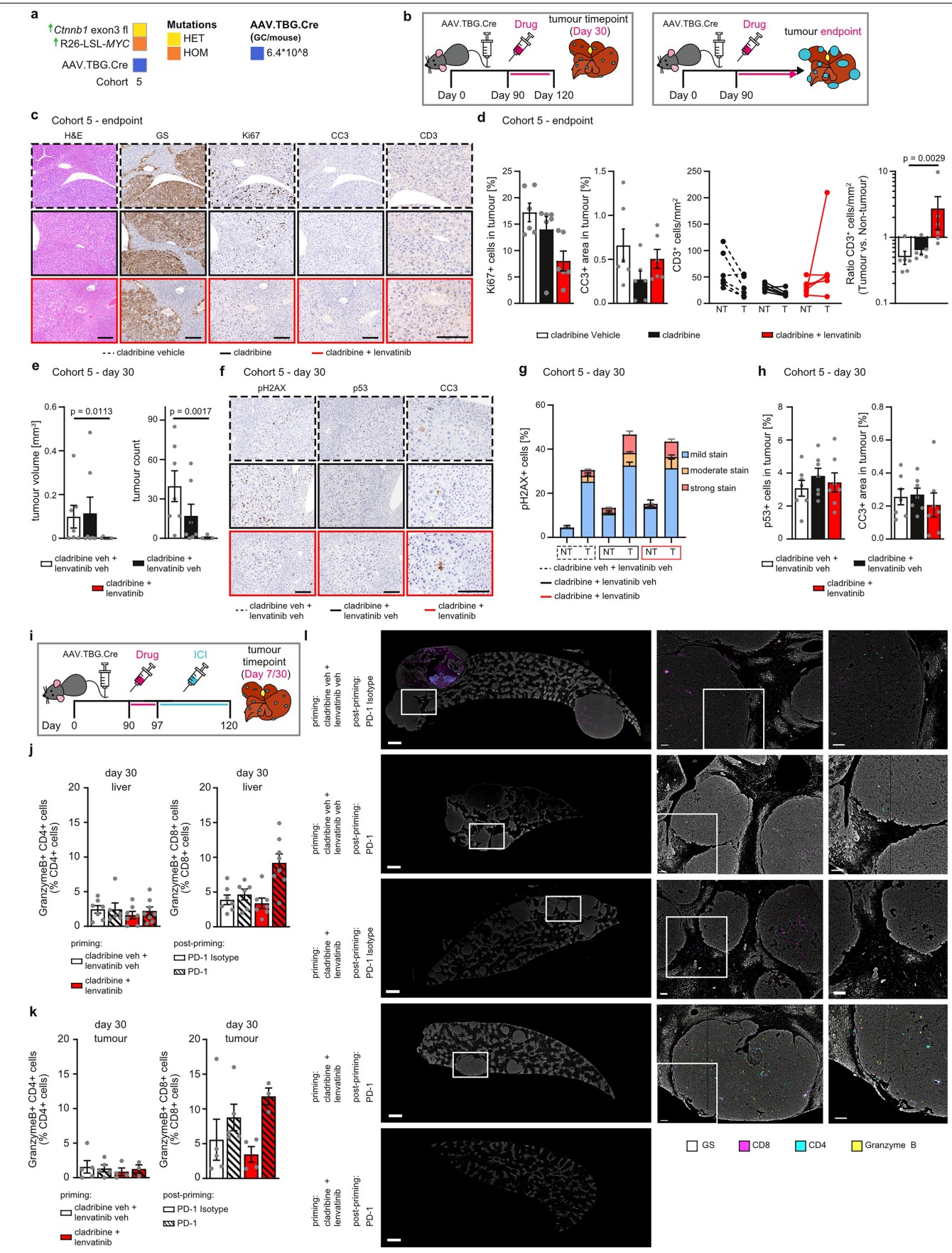

**Extended Data Fig. 12** | See next page for caption.

**Extended Data Fig. 12 | Cladribine decreases tumour burden associated with increasing immune cell infiltration and primes tumours for ICI therapy.** (**a** + **b**) Cohort summary and schematic of treatment regimens used in **c-h**. All mice used in this figure were male. (**c** + **d**) Cohort 5 (BM) mice treated with cladribine + lenvatinib have fewer proliferating cells in their tumours and more infiltration of CD3⁺ T-cells, but levels of cleaved Caspase 3 (CC3) as well as general morphology are unaltered when compared at endpoint. (**c**) Representative images. Scale bars equal 200 μm (H&E, GS, Ki67, CC3) or 100 μm (CD3). (**d**) Quantification of Ki67, CC3, and CD3 in matched non-tumour (NT) and tumour (T) tissue. n = 6 (cladribine vehicle, cladribine), 8 (cladribine + lenvatinib, two of which mice did not have microscopic tumours to quantify and therefore were excluded). Data shown as mean ± s.e.m. Kruskal-Wallis test with Dunn's correction (Ki67, CD3)/One-way ANOVA with Tukey correction (CC3). (**e**) After 30 days on treatment, Cohort 5 (BM) mice on cladribine + lenvatinib combination therapy have smaller and fewer tumours. n = 7 (all sample groups). Data shown as mean ± s.e.m. Kruskal-Wallis test with Dunn's correction. (**f-h**) Cladribine treatment for 30 days, either as monotherapy or combination therapy, induces DNA damage in matched tumour (T) and non-tumour (NT) tissue as determined by phosphorylation of Histone 2AX (pH2AX). This does not result in increased senescence, assessed by p53, or apoptosis, assessed by cleaved caspase 3 (CC3). (**f**) Representative immunohistochemistry images. Scale bars equal 200 μm (pH2AX, p53) or 100 μm (CC3). (**g**) Quantification of pH2AX in matched non-tumour (NT) and tumour (T) tissue. n = 7 (all sample groups). Data shown as mean + s.e.m. Two-way ANOVA with Tukey correction. (**h**) Quantification of p53 and CC3 expression in tumour tissue. n = 7 (all sample groups). Data shown as mean ± s.e.m. One-way ANOVA with Tukey correction. (**i**) Schematic of treatment regimens used in **j-l**. All mice used in this figure were male. (**j** + **k**) Increase of cytotoxic T-cells in liver and tumour at the d30 timepoint after priming + anti-PD-1. n = 7 (cladribine veh + lenvatinib veh + PD-1 Isotype liver, cladribine + lenvatinib + PD-1 Isotype liver), 6 (cladribine veh + lenvatinib veh + PD-1 liver), 8 (cladribine + lenvatinib + PD-1 liver), 5 (cladribine veh + lenvatinib veh + PD-1 Isotype tumour, cladribine veh + lenvatinib veh + PD-1 tumour), 4 (cladribine + lenvatinib + PD-1 Isotype tumour), 3 (cladribine + lenvatinib + PD-1 tumour). Data shown as mean ± s.e.m. Kruskal-Wallis test with Dunn's correction (CD4 liver, CD8 tumour), One-way ANOVA with Tukey correction (CD8 liver, CD4 tumour). (**l**) Representative whole lobe and magnified images used in Fig. 5k. Scale bars equal 1 mm (right panel) and 100 μm (middle and left panel).

# Reporting Summary

## Statistics

For all statistical analyses, confirm that the following items are present in the figure legend, table legend, main text, or Methods section.

| n/a | Confirmed | |
|---|---|---|
| ☐ | ☒ | The exact sample size (*n*) for each experimental group/condition, given as a discrete number and unit of measurement |
| ☐ | ☒ | A statement on whether measurements were taken from distinct samples or whether the same sample was measured repeatedly |
| ☐ | ☒ | The statistical test(s) used AND whether they are one- or two-sided<br>*Only common tests should be described solely by name; describe more complex techniques in the Methods section.* |
| ☒ | ☐ | A description of all covariates tested |
| ☐ | ☒ | A description of any assumptions or corrections, such as tests of normality and adjustment for multiple comparisons |
| ☐ | ☒ | A full description of the statistical parameters including central tendency (e.g. means) or other basic estimates (e.g. regression coefficient) AND variation (e.g. standard deviation) or associated estimates of uncertainty (e.g. confidence intervals) |
| ☐ | ☒ | For null hypothesis testing, the test statistic (e.g. *F*, *t*, *r*) with confidence intervals, effect sizes, degrees of freedom and *P* value noted<br>*Give P values as exact values whenever suitable.* |
| ☒ | ☐ | For Bayesian analysis, information on the choice of priors and Markov chain Monte Carlo settings |
| ☒ | ☐ | For hierarchical and complex designs, identification of the appropriate level for tests and full reporting of outcomes |
| ☐ | ☒ | Estimates of effect sizes (e.g. Cohen's *d*, Pearson's *r*), indicating how they were calculated |

*Our web collection on statistics for biologists contains articles on many of the points above.*

## Software and code

Policy information about availability of computer code

| Data collection | No software was used |
|---|---|
| Data analysis | HALO image analysis (v3.1.1076.363), Columbus Image analysis (v2.8.0.138890), VivoQuant (v4.0), Icy BioImage, v2.0.0.0, GraphPad Prism software (v9), R (v 4.0.2 and higher), FastQC (v0.11.9), FastP (v0.20.1), MultiQC (v1.9), FastQ Screen (v0.14.0), STAR (v2.7.8a and v2.5.1b), Subread (v2.0.1), GenomicDataCommons (v1.12.0), GenePattern v3.9, maftools (v2.4.2), DESeq2, v1.28.1 and v1.44.0, biomaRt version 2.56.1, uwot(v0.1.11), RANN (v2.6.1), igraph, versions 1.2.11 and 2.0.3, corto, 1.2.4, msigdbr (v7.4.1), ComplexHeatmap v2.4.3, ggplot2 versions 3.3.6 and 3.5.1, cowplot (v1.1.1), clusterProfiler version 3.16.1, DIVA (v8.0.1), FlowJo (v9.9.6), Visiopharm (v2024.06.0.19093 x64 and v2024.07.1.16745 x64), Scribus (v1.4.8), Gimp (v2.10.14), Inform (v2.6.0), featureCounts (v1.5.2).<br><br>Code availability : Scripts used for disease positioning is available at https://codeocean.com/capsule/9804119/tree/v1 |

For manuscripts utilizing custom algorithms or software that are central to the research but not yet described in published literature, software must be made available to editors and reviewers. We strongly encourage code deposition in a community repository (e.g. GitHub). See the Nature Portfolio guidelines for submitting code & software for further information.

## Data

Policy information about availability of data

All manuscripts must include a data availability statement. This statement should provide the following information, where applicable:

- Accession codes, unique identifiers, or web links for publicly available datasets
- A description of any restrictions on data availability
- For clinical datasets or third party data, please ensure that the statement adheres to our policy

Data files for transcriptomic analyses can be found on the Gene Expression Omnibus (GEO) repository; accession numbers: GSE275444 accessible via token 'ubefgeqkrxkjdep' on GSE273806 (Mouse models) and GSE275443 (Organoids). Our transcriptomic data are freely available to browse via a user-friendly interactive browsing online app enabling HuMo classification of external transcriptomic datasets (http://shinyapps.crukscotlandinstitute.ac.uk/humo_app/). Immunohistochemical and H&E staining of the GEMMs are publicly available via BioImage Archive (https://www.ebi.ac.uk/) via accession number S-BIAD1365. Montironi cohort data was provided upon request to the original authors (doi: 10.1136/gutjnl-2021-325918) and the TCGA data was accessed from publicly accessible databases (doi: 10.1016/j.cell.2017.05.046.). Mouse genome (GRCm39.103) was accessed from https://www.ensembl.org.
All data generated and/or analysed during the current study are also available from the corresponding authors on reasonable request.

# Field-specific reporting

Please select the one below that is the best fit for your research. If you are not sure, read the appropriate sections before making your selection.

☒ Life sciences ☐ Behavioural & social sciences ☐ Ecological, evolutionary & environmental sciences

For a reference copy of the document with all sections, see nature.com/documents/nr-reporting-summary-flat.pdf

# Life sciences study design

All studies must disclose on these points even when the disclosure is negative.

| | |
|---|---|
| Sample size | No statistical methods were used to pre-determine sample sizes but our sample sizes are similar to those reported in previous publications. For animal experiments biological replicate sizes were chosen taking into account the variability observed in pilot and prior studies in respective cohorts. Animal studies were also carried out respecting the limited use of animals in line with the 3R system: Replacement, Reduction, Refinement |
| Data exclusions | No data were excluded without having met prespecified QC limits. The following were excluded before analysis: One biological replicate failed QC post transcriptomic sequencing, all other biological replicates from this and other cohorts successfully passed QC and were included in downstream analysis; two drugs were excluded from the HCCO HTP screen due to microbiological contamination and drug precipitation in multiple replicates, respectively. One sample was excluded from the RFP expression analysis during analysis (total n=4 biological replicates): testing AAV-mediated recombination of RFP alleles in females (ED Figure 1b), one sample was a notable outlier (4.9% vs 25.7/25.1/25.8%) which upon re-review was caused by inconsistent RFP staining of the section - this outlier was removed from final analysis; details are provided in the figure legend also. Where tumour number could not be quantified due to tumour rupture no tumour number is reported (i.e. Fig 5d). |
| Replication | Individual animals of control and experimental cohorts are biologically unique - replicate data represents analysis of data/samples from independent replicate animals and is denoted by "n". Separate vehicle control treatment arms have been consistent with tumour penetrance and survival with the original untreated cohorts in all instances. Efficacy of cladribine and/or lenvatinib has been further replicated in a subsequent BM cohort. In vitro experiments were replicated at least once (HTP screen) but generally twice (dose validation assays) and results were replicable. |
| Randomization | To reduce the impact of confounding factors such as litter mates or induction dates for all experiments animals/sample assignment was matched for age-matched control and were assigned based upon randomly assigned mouse identification markings. No randomisation was performed during organoid screening or validation. Samples for transcriptomic analysis were prepared as a single batch sequentially without randomisation. |
| Blinding | The investigators were not blinded for the in vivo experiments. Technical staff administering therapy were blinded to the mouse genotypes. All subsequent tissue handling and analysis was blinded and/or performed using standardised automated analyses where possible. Quantitative image analysis was performed blinded to the genotype and treatment. |

# Reporting for specific materials, systems and methods

We require information from authors about some types of materials, experimental systems and methods used in many studies. Here, indicate whether each material, system or method listed is relevant to your study. If you are not sure if a list item applies to your research, read the appropriate section before selecting a response.

## Materials & experimental systems

| n/a | Involved in the study |
|---|---|
| ☐ | ☒ Antibodies |
| ☐ | ☒ Eukaryotic cell lines |
| ☒ | ☐ Palaeontology and archaeology |
| ☐ | ☒ Animals and other organisms |
| ☐ | ☒ Human research participants |
| ☒ | ☐ Clinical data |
| ☒ | ☐ Dual use research of concern |

## Methods

| n/a | Involved in the study |
|---|---|
| ☒ | ☐ ChIP-seq |
| ☒ | ☐ Flow cytometry |
| ☒ | ☐ MRI-based neuroimaging |

## Antibodies

| Antibodies used | All antibodies used are also described in extended data table 5 |
|---|---|
| | Antibody/Manufacturer /Cat_no /Clone number (monoclonals)/ Lot number |
| | GS Abcam ab49873 1/500 or 1/1000 NA GR3384613-5 |
| | GS Sigma-Aldrich HPA007316 1/1000 NA C81287 |
| | GS Sigma-Aldrich HPA007316 1/1000 NA C81287 |
| | Sox9 Milliore AB5535 1/500 NA 3249418 |
| | GFP Cell Signalling 2555 1/600 NA 6 |
| | HNF4a Santa Cruz SC6556 1/100 NA not available |
| | HNF4a Santa Cruz SC374229 1/100 H-1 G0116 |
| | HNF4a Biotechne PP-H1415-00 1/500 H1415 A-2 |
| | RFP Tebu-Bio 600-401-379 1/1000 NA 42872 |
| | F4/80 AbD SeroTec MCA497 1/150 A3-1 GR3279416-2 |
| | CD68 Dako M0876 1/400 PG-M1 not available |
| | CD4 eBioscience 14-9766-82 1/75 4SM95 2115647 |
| | CD4 Dako M7310 1/35 4B12 not available |
| | CD8a eBioscience 14-0808-82 1/75 4SM15 2127137 |
| | CD8a AbD SeroTec MCA1817T 1/30 4B11 6088619 |
| | Ki67 Abcam ab16667 1/200 SP6 GR216200-3/ GR3313195-42 |
| | Ki67 Cell Signalling 12202 1/1000 D3B5 6 |
| | cleaved Caspase 3 Cell Signalling 9661 1/500 NA 45 |
| | CD3 Abcam ab16669 1/100 SP7 GR3247742-11 |
| | γH2AX Cell Signalling 9718 1/120 20E3 17 |
| | p53 Leica NCL-L-p53CM5p 1/150 NA 6087005 |
| | CD34 Biolegend 119302 1/100 Mec14.7 not available |
| | Ctnnb1 BD BD610154 1 to 50 14 9315374 |
| | CD31 Abcam ab28364 1/75 NA 32447742-21 |
| | pAKT (473) Cell Signaling 4060 1/50 9DE 25 |
| | PTEN Cell Signaling 9559 1/70 138G6 19 |
| | Zeb1 Cell Signaling 70512 1/500 E2G6Y 2 |
| | c-Myc Roche Tissue Diagnostics 790-4628 RTU Y69 K05962 |
| | GS Sigma-Aldrich hpa007316 1/500 NA 16878 |
| | CD8 Thermo Fisher Scientific 14-0808-82 1 to 75 4SM15 2720194 |
| | CD4 Thermo Fisher Scientific 14-9766-82 1 to 25 4SM95 2526300 |
| | GranzymeB Novus Biologicals nb100-684 1/100 NA R705 |
| | GS Sigma-Aldrich hpa007316 1/500 NA 16878 |
| | CD45 Abcam ab10558 1/300 NA 1041690-2 |
| | Ctnnb1 Cell signalling Technologies 8480 1/25 D10A8 9 |
| | CD3e-BV650 (clone 17A2) BioLegend 100229 1/100 17A2 B394769 |
| | TCRβ-BV510 (clone H57-597) BioLegend 109234 1/100 H57-597 B367672 |
| | TCRd-FITC (clone GL3) eBioscience 11-5711-85 1/200 GL3 1935313 |
| | CD4-BV605 (clone GK1.5) BioLegend 100421 1/100 GK1.5 B386408 |
| | CD8a- BUV395 (clone 53-6.7) BD 563786 1/100 53-6.7 3003153 |
| | Granzyme B-AF647 (clone GB11) BioLegend 515406 1/50 GB11 B367007 |
| | PD1 BioLegend 114102 200ug RMP1-14 B411441 |
| | IgG isotype BioLegend 400502 200ug RTK2758 B409040 |

| Validation | All antibody validation are also described in extended data table 5. All antibodies were used according to the manufacturer's intended application. Antibodies were also routinely validated within the lab and/or histology department on control tissue, with appropriate cellular/tissue localisation confirmed and where appropriate in cells morphologically consistent with the target and "No primary antibody (NPA)" controls were included for all stainings shown in this manuscript. |
|---|---|
| | Antibody /det_species /Manufacturer /Cat_no /Application /Dilution /Notes /Clone number (monoclonals) /Lot number /Validation method/species/application |
| | GS mouse Abcam ab49873 IF 1/500 or 1/1000 NA GR3384613-5 synthetic peptide by manufacturer, zonal specificity in liver by researchers/mouse/IHC |
| | GS mouse Sigma-Aldrich HPA007316 IHC 1/1000 NA C81287 zonal specifity in liver by researchers/mouse/IHC |

GS human Sigma-Aldrich HPA007316 IHC 1/1000  NA C81287 zonal specificity in liver by researchers/human/IHC

Sox9 mouse Milliore AB5535 IHC 1/500  NA 3249418 IHC and WB by manufacturer/mouse/IHC

GFP NA Cell Signalling 2555 IF 1/600  NA 6 transfected cells by manufacturer, specific expression in positive control tissue by researchers/NA/WB and IHC respectively

HNF4a mouse Santa Cruz SC6556 IF 1/100 discontinued - used Biotechne antibody subsequently NA not available hepatocyte specific nuclear expression by researchers/mouse/IHC

HNF4a mouse Santa Cruz SC374229 IF  1/100  H-1 G0116 hepatocyte specific nuclear expression by researchers/mouse/IHC

HNF4a mouse Biotechne PP-H1415-00 IF 1/500  H1415 A-2 isoform recognition by manufacturer, hepatocyte specific nuclear expression by researchers/human and mouse respectively/IHC

RFP NA Tebu-Bio 600-401-379 IF 1/1000  NA 42872 specific to purified RFP with no cross reactivity to human or mouse serum by manufacturer, specific expression in positive control tissue by researchers/NA/immunoelectrophoresis and IHC respectively

F4/80 mouse AbD SeroTec MCA497 IHC 1/150  A3-1 GR3279416-2 murine F4.80 antigen by manufacturer/mouse/IHC

CD68 human Dako M0876 IHC 1/400  PG-M1 not available clustered as anti-CD68 at Fifth International Workshop and Conference on Human Leucocyte Differentiation Antigens held in Boston in 1993/human/IHC

CD4 mouse eBioscience 14-9766-82 IHC 1/75  4SM95 2115647 cell treatment by manufacturer/mouse/IHC

CD4 human Dako M7310 IHC 1/35  4B12 not available none available

CD8a mouse eBioscience 14-0808-82 IHC 1/75  4SM15 2127137 relative expression by manufacturer/mouse/not available

CD8a human AbD SeroTec MCA1817T IHC 1/30  4B11 6088619 synthetic peptide by manufacturer/human/not available

Ki67 mouse Abcam ab16667 IF 1/200  SP6 GR216200-3/ GR3313195-42 knockout validated/mouse/IHC

Ki67 mouse Cell Signalling 12202 IHC 1/1000  D3B5 6 none available

cleaved Caspase 3 mouse Cell Signalling 9661 IHC 1/500  NA 45 blocking peptide by manufacturer/mouse/IHC

CD3 mouse Abcam ab16669 IHC 1/100  SP7 GR3247742-11 advanced validation by manufacturer/mouse/IHC

yH2AX mouse Cell Signalling 9718 IHC 1/120  20E3 17 DNA damage and lambda phosphatase +ve/-ve controls by manufacturer/human/IHC

p53 mouse Leica NCL-L-p53CM5p IHC 1/150  NA 6087005 knockout by other researchers/mouse/WB

CD34 mouse Biolegend 119302 IHC 1/100  Mec14.7 not available none available

Ctnnb1 mouse BD BD610154 IF 1 to 50  14 9315374 murine immunogen and QC testing by manufacturer/human/WB

CD31 mouse Abcam ab28364 IHC 1/75  NA 32447742-21 endothelial localisation by manufacturer and researcher/human and mouse/IHC

pAKT (473) mouse Cell Signaling 4060 IHC 1/50  9DE 25 insulin treatment +ve and lambda phosphatase/human/IHC

PTEN mouse Cell Signaling 9559 IHC 1/70  138G6 19 knockout and control peptide by manufacturer/human/IHC

Zeb1 mouse Cell Signaling 70512 IHC 1/500  E2G6Y 2 recombinant protein/Human/NA

c-Myc  mouse Roche Tissue Diagnostics 790-4628 IHC (duplex) RTU stain: anti-Rabbit HQ & anti-HQ HRP (Roche Tissue Diagnostics, 760-4815, 07017936001), Opal 570 (Akoya Biosciences, FP1488001KT), 1:50, 8 min Y69 K05962 CE-IVD antibody approved, a positive control (Human tonsil)was used to validate the assay and Ms liver staining was assessed  by a pathologist; specificity of the secondary antibody was proved via negative control/Mouse and human/IHC

GS mouse Sigma-Aldrich hpa007316 IHC (duplex) 1/500 stain: OmniMap-antiRb HRP (Roche Tissue Diagnostics, 760-4311) 12 min, Opal 520 (Akoya Biosciences, FP1487001KT), 1:300, 8 min NA 16878 Antibody tested in WB and IHC by supplier, zonal specifity in liver by researchers,  specificity of the secondary antibody was proved via negative control/mouse/IHC

CD8 mouse Thermo Fisher Scientific 14-0808-82 IHC (multiplex) 1 to 75 stain: OmniMap-antiRt HRP, 20 min (Roche Tissue Diagnostics, 760-4457), Opal 690 (Akoya Biosceinces, FP1497001KT), 1:100, 8 min 4SM15 2720194 relative expression by manufacturer, immune cells in ms liver were assessed by a pathologist,specificity of the secondary antibody was proved via negative control /mouse/IHC

CD4 mouse Thermo Fisher Scientific 14-9766-82 IHC (multiplex) 1 to 25 stain: Impress anti-Rat, (Vector Laboratories, ZJ0512) 48 min, Opal 620 (Akoya Biosciences, FP1495001KT), 1:50, 8 min 4SM95 2526300 cell treatment by manufacturer, immune cells in ms liver were assessed by a pathologist,specificity of the secondary antibody was proved via negative control /mouse/IHC

GranzymeB mouse Novus Biologicals nb100-684 IHC (multiplex) 1/100 stain: OmniMap-antiRb HRP (Roche Tissue Diagnostics, 760-4311) 12 min, Opal 650 (Akoya Biosciences, FP1496001KT), 1:300, 8 min NA R705 Positive control (mouse spleen) was used to validate the assay and double positivity with CD8 was assessed by a pathologist in Ms liver,specificity of the secondary antibody was proved via negative control/mouse/IHC

GS  mouse Sigma-Aldrich hpa007316 IHC (multiplex) 1/500 stain: OmniMap-antiRb HRP (Roche Tissue Diagnostics, 760-4311) 12 min, Opal 520 (Akoya Biosciences, FP1487001KT), 1:300, 8 min NA 16878 Antibody tested in WB and IHC by supplier, zonal specifity in liver by researchers,  specificity of the secondary antibody was proved via negative control/mouse/IHC

CD45 mouse Abcam ab10558 IHC (multiplex) 1/300 stain: OmniMap-antiRb HRP (Roche Tissue Diagnostics, 760-4311), 12 min, Opal 480 (Akoya Biosciences, FP1500001KT), 1:25, 8 min NA 1041690-2 Antibody tested in WB and IHC by supplier, immune cells were assessed by a pathologist,specificity of the secondary antibody was proved via negative control /mouse/IHC

Ctnnb1 mouse Cell signalling Technologies 8480 IHC (multiplex) 1/25 stain: Ultramap-antiRb HRP (Roche Tissue Diagnostics, 760-4315) 12 min, TSA-DIG (Akoya Biosciences, FP1502001KT), 1:100, 12 min and Opal 780 (Akoya Biosciences, FP1501001KT), 1:10, 1h.  D10A8 9 Antibody tested by supplier with HeLa Cells and tissue, zonal specifity in liver assessed by a pathologist,  specificity of the secondary antibody was proved via negative control/human +mouse/WB and IHC

CD3e-BV650 (clone 17A2) mouse BioLegend 100229 FACS 1/100  17A2 B394769 manufacturer/mouse/FACS

 TCRβ-BV510 (clone H57-597)  mouse BioLegend 109234 FACS 1/100  H57-597 B367672 manufacturer/mouse/FACS

TCRd-FITC (clone GL3)  mouse eBioscience 11-5711-85 FACS 1/200  GL3 1935313 none available

CD4-BV605 (clone GK1.5)  mouse BioLegend 100421 FACS 1/100  GK1.5 B386408 manufacturer/mouse/FACS

CD8a- BUV395 (clone 53-6.7)  mouse BD 563786 FACS 1/100  53-6.7 3003153 manufacturer/mouse/FACS

Granzyme B-AF647 (clone GB11) mouse BioLegend 515406 FACS 1/50  GB11 B367007 manufacturer/human+mouse/FACS

PD1 mouse BioLegend 114102 Therapy 200ug  RMP1-14 B411441 The RMP1-14 antibody has been reported to block the binding of PD-1 to its ligands (B7-H1 and B7-DC) and to inhibit T cell proliferation and cytokine production costimulated by macrophages (but not by dendritic cells and B cells)

IgG isotype mouse BioLegend 400502 Therapy control 200ug  RTK2758 B409040 screening of a variety of resting, activated, live and fixed mouse tissues by manufacturer/mouse/IHC

# Eukaryotic cell lines

Policy information about cell lines

| | |
|---|---|
| Cell line source(s) | Human organoid cell lines were previously described (Nuciforo et al., Cell Reports, 2018). Murine organoid lines were derived at the Cancer Research UK Beatson Institute.<br>The Hep-53.4 (RRID:CVCL_5765) was purchased from Cytion – LOT-L230232R |
| Authentication | None of the murine cell lines were authenticated as they were mouse-derived organoid lines. Comparison, both histological, protein expression and bulk transcriptome is provided in comparison to originating tumour models in the manuscript.<br>The same Hep53.4 line was used in house for this study as was used previously (doi: 10.1136/gutjnl-2021-326259) including genomic and transcriptomic characterisation at that time. |
| Mycoplasma contamination | lines were routinely tested by PCR for mycoplasma contamination and the lines used tested negative. |
| Commonly misidentified lines<br>(See ICLAC register) | no commonly misidentified lines were used in this study |

# Animals and other organisms

Policy information about studies involving animals; ARRIVE guidelines recommended for reporting animal research

| | |
|---|---|
| Laboratory animals | Details for all animals involved in this study can be found in the methods section and are as follows:<br>Unless otherwise specified male mice on a mixed background were used. The following transgenic mice strains were used: Gt(ROSA)26Sortm14(CAG-tdTomato)Hze (R26LSL-Tom), Ctnnb1tm1Mmt (Ctnnb1ex3), Gt(ROSA)26Sortm1(MYC)Djmy (R26LSL-MYC), Trp53tm1Brn (Trp53fl), Trp53tm2Tyj (Trp53R172H), Cdkn2atm1.1Brn (Cdkn2aKO), Ptentm2Mak (Ptenfl), Gt(ROSA)26Sortm1(Notch1)Dam (R26LSL-NICD), Krastm4Tyj (KrasG12D), Cdkn1atm1Led (Cdkn1aKO), Axin1 (Axin1fl) and Bap1tm2c(EUCOMM)Hmgu. Mice were induced between 8 and 12 weeks of age, unless otherwise indicated.<br>For the GEMM+MWD model 6-week old mice were kept on a modified western diet (Envigo -TD.120528) plus sugar water (23.1 g/L fructose and 18.9 g/L glucose) in combination with repeated CCl4 injections (ip, 0.2 µl/g of body weight, Veh: Cornoil) as referenced and were induced with AAV-TBG-Cre at 10 weeks of age.<br>For the DEN/ALIOS model, C57BL/6 WT mice, were injected with a single dose of DEN (80 mg/kg by i.p. injection) at 14 days of age. Mice were fed ALIOS diet (Envigo, TD.110201) and sugar water (23.1 g/L fructose and 18.9 g/L glucose) from 60 days of age. Mice were harvested at day 284.<br>For MWD+CCl4 model mice were kept on a modified western diet (Envigo -TD.120528) plus sugar water (23.1 g/L fructose and 18.9 g/L glucose) in combination with repeated CCl4 injections (ip, 0.2 µl/g of body weight, Veh: Cornoil) as previously described51.<br>For the streptozotocin (STZ) model, male and female C57BL/6J WT mice were injected with a single dose of STZ (200µg in 0.1M citrate buffer, pH 4.0) subcutaneously at 2 days of age. Mice were fed high-fat diet (TestDiet 58R3, cat.no. 1810835) from 30 days of age.<br>For the orthotopic model, Hep-53.4 cells (female C57BL/6J hepatoma cell line) were injected intrahepatic into the left lobe of male C57BL/6J mice.<br>Mice were housed under controlled conditions (specific pathogen free, 12hr light-dark cycle, 19-22 °C, 45-65% humidity) with access to food and water ad libitum. We added environmental enrichments, in the form of gnawing sticks, plastic tunnels, and nesting material to all cages. |
| Wild animals | no wild animals were used in this study |
| Field-collected samples | no field-collected samples were used in this study |
| Ethics oversight | All animal experiments were performed in accordance with UK Home Office licences (70/8891, PP0604995, 70/8646, 70/8468, and PP8854860) and in accordance with the UK Animal (Scientific Procedures) Act 1986 and EU direction 2010. They were subject to review by the animal welfare and ethical review board of the University of Glasgow and the University of Newcastle upon Tyne |

Note that full information on the approval of the study protocol must also be provided in the manuscript.

# Human research participants

Policy information about studies involving human research participants

| | |
|---|---|
| Population characteristics | All human TCGA data used in this study is publically available (doi: 10.1016/j.cell.2017.05.046). HuMo clusters were validated with the bulk RNAseq data of an independent cohort of 171 HCC samples from patients undergoing resection collected in the setting of the HCC Genomic Consortium (European Genome-Phenome Archive code EGAS00001005364). |
| Recruitment | No patients were recruited for this study. |
| Ethics oversight | For the representative human HCC sample (no longer included post revision): the use of consenting patients' tissues surplus to diagnostic requirements for research purposes was approved by the Newcastle and North Tyneside Regional ethics committee, the Newcastle Academic Health Partners Bioresource (NAHPB) and the Newcastle upon Tyne NHS Foundation Trust Research and Development (R&D) department, in accordance with Health Research Authority guidelines. (References 10/H0906/41; NAHPB Project 48; REC 12/NE/0395; R&D 6579; Human Tissue Act license 12534).<br>Ethics oversight for the previous cohorts described in the TCGA and Montironi et al. cohorts have been described previously. |

Note that full information on the approval of the study protocol must also be provided in the manuscript.

