## [Peer Review file · Nature]

Human-correlated genetic models identify precision therapy for liver cancer

Corresponding Author: Professor Thomas Bird

Version 0:

Reviewer comments:

Referee #1

(Remarks to the Author)

Müller et al. present a roust and interesting manuscript centered around a plethora of GEMMs representing the majority of common human HCC driver genes. These GEMMs serve as a platform to identify novel HCC treatment strategies. Despite the tremendous efforts to comprehensively model human HCC in mice and to make use of the autochthonous HCC models to identify a combination therapy potentially surpassing the standard-of-care therapeutic lenvatinib, various issues related to the presented data need to be addressed to produce a compelling manuscript. Details are delineated below.

1. The authors cite Ally et al. (Comprehensive and Integrative Genomic Characterization of Hepatocellular Carcinoma, Cell, 2017; reference 4) as resource to identify HCC driver genes and potentially relevant combinations thereof. The authors should elaborate on why several top driver gene alterations (e.g., ALB or BAP1 LoF mutations) were not considered for modeling? Importantly, the authors should also provide statistical evaluations supporting the selected driver gene combinations, i.e., by using odds ratios and Fishers exact testing to evaluate co-occurrence / mutual exclusivity among co-drivers. Based on the TCGA oncoplot displayed in Figure 1 in Ally et al., it is unclear whether for instance TP53 and CTNNB1 mutations co-occur or are rather mutually exclusive. Such types of co-occurrence analyses using TCGA or other HCC oncogenomic datasets would strengthen the rationale for selection of modeled driver genes and especially their combinations.
2. In Figure 1b, the authors present a tremendous effort to model driver gene combinations potentially relevant for HCC initiation/progression in mice. However, except for Ctnnb1 and Myc (Extended Data Fig. 1i, l), the single driver gene cohorts are missing. In a revised version of this manuscript, these relevant controls should be provided and should also be included into the histological and molecular analyses in Figures 2-3.
3. The authors should present metastasis frequencies (resolved by metastatic site) for each cohort in the main Figure 1, rather than just in Extended Data Table 1.
4. It remains unclear whether the emerging HCCs in each cohort are actually driven by the targeted driver genes. The authors should use IHC or other measures to evaluate overexpression or loss, respectively, of targeted driver genes in each cohort.
5. The RNA-seq based analyses in Figure 3 nicely identify 4 distinct HCC clusters using mouse and human TCGA data. Do the 4 clusters identified here correspond to the 4 clusters previously described in Ally et al. (Figure 2A)? The authors should elaborate on this and cross-compare their identified clusters to the previously described TCGA-HCC clusters. Cross-species analyses in Figure 3a should also be carried out in a driver-gene (combination)-specific manner. For instance, show both mouse and human HCCs harboring MYC amplification/overexpression similar transcriptional landscapes (e.g., enrichment of MYC_targets and metabolism hallmarks)?

6.
The authors claim that mutation status does not necessarily correspond to (downstream) signaling status (lines 138-139). Nevertheless, the authors should examine whether in their 4 clusters specific modeled drivers or driver combos were enriched. Moreover, it should be evaluated whether the corresponding human driver genes are likewise enriched in the human samples within the according clusters.
7.
Distinct morphological features across the 4 clusters described in the text (lines 160-166) and based on the HEs displayed in Figure 3b should be quantitatively assessed using appropriately sized cohorts for each cluster. These should include quantification of the histopathological features including pathological grade, differentiation, ECM deposition, steatosis, as well as immune infiltration. Moreover, in support of the morphological features observed in the HEs, authors could consider IHC stainings for markers representative of the described features (e.g., CD45 or more specific markers for immune cell subsets to characterize immune infiltration). Finally, to strengthen distinction of the 4 clusters, IHC markers representative of gene ontology hallmarks disclosed in Figure 3a should be used. For instance, EMT markers such as ZEB1 should preferentially stain cluster 4 tumors.
8.
Statistical evaluations of survival probability differences in Extended Data Figure 5c-e are missing.
9.
Enrichments of immune scores as displayed in oncoplot Extended Data Figure 6c should be quantified in both human and mouse cluster 1-4 and also subdivided into CTNNB1 WT vs mutant. Moreover, in the UMAP plot in Extended Data Figure 6d it is unclear where the 4 clusters locate to. The Figure legend does also not help understanding this panel. As currently displayed, it does not support the sentence in the manuscript text in lines 199-200.
10.
IHCs in Extended Data Figure 6e have to be quantified across cohorts of the respective genotypes and should be expanded by IHCs on samples representing the other human and mouse clusters, in support of Extended Data Figure 6c.
11.
Besides clusters 1 and 2 highlighted in Extended Data Figure 7c and analyzed in 7d, the authors should also perform gene ontology analyses on the remaining gene clusters. In particular, the top and bottom clusters showing 'high-high-low' and low-low-high' gene expression patterns appear intriguing, as these indicate transcriptional changes in tumor progression from adenoma to adenocarcinoma (which, to some extent, is also the case for cluster 1).
12.
Although TKIs, such as sorafenib and lenvatinib, are first-line therapies for HCC, it appears unclear as to why in particular the cohort 5 / BM model should be sensitive to these inhibitors? Are other HCC mouse models equally sensitive to these inhibitors? Moreover, the authors should leverage the RNA-seq analyses in Extended Data Figure 7c (and possibly of other cohorts than cohort 5) to demonstrate upregulation of RTKs targeted by sorafenib / lenvatinib (specifically) in the cohort 5 / BM model, serving as a rationale for these targeted interventions.
13.
Anti-PD therapy showed no efficacy in cohort 5 tumors (Figure 4g). These data should either be moved to supplementary or further analyses including immunoprofiling would be needed to explain this observation. Are any of the other modeled driver gene combinations giving rise to tumors responding to ICIs?
14.
In Extended Data Figure 9b, authors should also demonstrate maintained (over)-expression of MYC.
15.
In Extended Data Figure 9c, authors should also show 'growth volumes' of control organoids. Moreover, the mTOR inhibitor everolimus is incorrectly named 'iverolimus'.
16.
Figure 5b shows that Trp53, Pten, and/or Cdkn2a LoF renders cohort 5 / BM organoids insensitive to cladribine. Examination of organoids from cohorts 11 (BM Trp53), 12 (BM Cdkn2a), and others would allow pinpointing the driver gene required for cladribine insensitivity. Furthermore, is cladribine sensitivity specific for the Ctnnb1 / Myc driver gene combo or are other cohorts that do not harbor Trp53, Pten, and/or Cdkn2a LoF also sensitive to cladribine?
17.
It would be nice if the authors were using in Figure 5c a human organoid line that would also harbor a MYC amplification. Based on the human organoid lines used, it appears that TP53 mutation status does not affect cladribine sensitivity, which, however, should be put to the test as delineated in comment 16.
18.
In the text (lines 280-282) related to the combination drug interventions presented in Figure 6c, authors refer to 20% weight loss observed in up to 60% of the mice treated with drug combo. Thus, in the Kaplan-meier curve of the combination regime,

many of the mice are censored during combination therapy due to clinical symptoms (weight loss). Therefore, the survival proportions as presented are misleading (mice succumbed to side-effects cannot be considered 'cured'). Instead of displaying disease-specific survival, authors should also plot overall survival. For the animals sacrificed during treatment, it remains unknown whether eventual tumor relapse would occur, comparable to the cladribine single treatment arm. As currently presented, the data are no convincing case for the combination therapy. The authors might want to optimize cladribine and/or lenvatinib regimes especially for the combination treatment optimally minimizing side effects while maintaining potency. Importantly, the single treatment arm for lenvatinib is missing in panel 6c as well as in subsequent analyses in Figure 6e-i (and corresponding Extended Data panels).

19. The observed CD3+ T-cell influx into cladribine / lenvatinib combo-treated tumors 30 days post treatment start (and apparently before mouse loss due to treatment side effects) is interesting. Authors could set-up an intervention experiment where cohort 6 mice would be primed for 30 (or less) days with cladribine / lenvatinib combo and then subsequently enrolled for ICI therapy.

20. It is intriguing in Figure 6c, i, j that cladribine / lenvatinib combination appears to be specifically efficient in models driven by *Cttnb1* and *Myc*. The authors could also apply the cladribine / lenvatinib combination to human organoids carrying altered *CTNNB1* (and *MYC*, if available).

Referee #2

(Remarks to the Author)

The manuscript "Human-correlated genetic HCC models identify combination therapy for precision 2 medicine" by Muller et al describes the generation of a variety of genetically-driven HCC models in mice in vivo and in cell lines in vitro. From these mouse models the investigators generated and then evaluated transcriptomic data and referenced them with available human HCC data to correlate the mouse tumors with human HCC data. Mouse subtypes that correlated with the human subtypes were similar in that mutational status was not linked to signaling (as similar to seen in prior human HCC).

had distinct histopathologic findings. From the cell lines that were generated a drug was identified (cladribine) that improved HCC subtype X outcome that had not been implicated in HCC treatment previously. Overall this manuscript is a tour-de-force in the large scale of animal models and cell lines that were generated. And the tumor->cell line->drug screen->target->in vitro and in vivo studies represents a novel tool and platform for future studies and for the community. However, the authors ignore or do not mention a large body of literature that argues that HCC is dissimilar from other cancers and is not simply driven from tumor drivers but is due to the complex interaction of the microenvironmental changes that occur during cirrhosis, the ongoing regenerative stimulus, and also results from nodule and tumor heterogeneity. Given this and other concerns diminishes the impact of the manuscript and in its current form should be rejected with hope for resubmission.

Comments for the author:

1. The authors state "Human HCC is thought to evolve from a hepatocytic clonal origin" and reference Brunner et al. While this has been the historical viewpoint on HCC development, Brunner et al and other more recent studies have shown and suggested that this is not the case. This represents a misinterpretation of this manuscript and needs to be corrected. Other reports for example (Zhu et al PMID: 30955891) identify several somatic mutations that are recurrent during chronic liver injury/liver disease promote regeneration without necessarily promoting hepatocarcinogenesis so the idea put forward as the premise underlying this manuscript that "driver" mutations identified in TCGA could be used to model human HCC somewhat ignore this body of literature.
2. Also not mentioned is a major conclusion of the Brunner et al. paper that "the connective tissue laid down during cycles of damage and regeneration sequesters clones from early stages of the disease process" which is not being replicated in any of the generated mouse models. Even more interestingly in the Brunner et al they found that the majority of hepatocytes in the nodules did not have any driver mutations and then suggest that the clonal expansions that were observed were likely due to the inherent regenerative capacity of human hepatocytes.
3. Human HCC is marked by tumor heterogeneity and has been demonstrated by a multitude of reports. The authors do not address or examine this in any of the murine models. Do any of the models generate intra-tumor heterogeneity? How does tumor heterogeneity impact tumor growth, mouse survival, and (in later experiments) drug response.
4. The authors acknowledge that several disease states impact HCC development such as NAFLD/NASH and then use the GEMM+MWD/CCL4 treatment model. HCC development in NASH can occur without the development of cirrhosis but is far more common and likely after cirrhosis development. However only 4 weeks of pretreatment with this diet was initiated prior to AAV administration. More extended times to generate a NASH-like state would be needed for these claims.
5. The authors delineate the different mouse models into different clusters when compared to human data from TCGA. The 4 cohorts that are identified as being more similar phenotypically. Deeper cellular characterization of the tumors including the tumor microenvironment would further strengthen that disparate mutations result in similar phenotypical and cellular compositions.
6. The authors show in Figure 4 that anti-PD1 therapy has no impact on survival in the Cohort 5 mice however a control showing benefit in another cohort group would show that this effect is specific to the *CTNNB1* mutant cohort rather than a lack of response in general in the animal models as performed.
7. The tumoroid platform in itself has been generated previously by several other groups however the generation of

tumoroids from this broad cohort of GEMM-derived HCCs and mutations is novel. However it is not clear how this platform will capture the tumor heterogeneity seen in human HCC. It is not clear how the tumoroids drift over time and differ from the initial mouse or human tumor. Further studies evaluating the drift genomically, transcriptomically, and in cellular composition would add greater strength to this model platform.

8. The tumoroid screen identified cladribine as a potent target which was validated in multiple of the mouse HCC models. More detailed impacts of all of the antimetabolites would add further detail on these related compounds.

Minor comments.

1) Histopathology noted in Figure 3B is somewhat hard to compare to each other due to figure size. Mouse images seem somewhat drowned out and not sure if this was due to figure uploading. Higher resolution images would greatly aid in its interpretation and should be included in the supplement.

2) Supp Figure 6E- immune staining for the different components should be from adjoining sections to better demonstrate the immune infiltration that is or is not present. The CD8 staining present seems to have a significant background staining.

Referee #3

(Remarks to the Author)

In their current manuscript Bird and colleagues report on the development and comprehensive characterization of twenty-seven genetically engineered autochthonous immunocompetent HCC mouse models. Adult mice harbouring endogenous floxed alleles of tumor suppressor genes or activatable oncogenes (particular focus on WNT pathway, cell cycle and RTK/RAS/PI3K pathway) were injected with an adenoviral vector encoding Cre recombinase under control of the hepatocyte specific thyroxine-binding Globulin (TBG) promoter to initiate tumorigenesis. Time to tumor onset, metastasis and survival were captured for all models. Furthermore, all models were subjected to a thorough histopathological analysis, which revealed a wide range of histological phenotypes, including well-differentiated HCC, undifferentiated HCC, pseudoglandular HCC and steatotic HCC, however, a clear genotype-phenotype correlation could apparently not be concluded from the presented data. Many models seem to develop lung and bone metastases which seems to be enriched in models driven by mutated CTNNB1 and Myc.

All models were subjected to transcriptional profiling and obtained data was integratively analyzed with data from chemically induced mouse HCC as well as with human HCC transcriptomic data. Using the Louvain method the authors compared the human and mouse transcriptome data based on functionally and mechanistically relevant pathway enrichment. These analyses yielded four major human/mouse (HuMo) clusters. Genetic mouse models were represented in all four clusters with varying heterogeneity within cohorts, whereas the purely carcinogen-induced models were only found representative of HuMo cluster 2. HuMo cluster 1 was enriched for pathways linked to metabolism and differentiation, but had negative enrichment for proliferation and inflammatory pathways. HuMo cluster 2 was related to cluster 1 but was distinct particularly through a higher enrichment in pro-inflammatory pathways. HuMo clusters 3 and 4 were both poorly differentiated and highly proliferative, with cluster 4 showing enrichment in epithelial-to-mesenchymal transition.

The presented cross species clustering approach seems more precise than previously established and clinically used classifications of human HCC, such as the Hoshida classification. For example, the herein presented clustering was able to distinguish two patient populations within Hoshida subclass S3, namely HuMo clusters 1 and 2. The reported distinction resulted in differences in patient survival, with patients associated with HuMo cluster 2 showing an improved survival probability relative to patients associated with the other HuMo clusters.

In line with recent reports, Muller et al could show that immune checkpoint blockade showed no therapeutic efficacy in CTNNB1-driven murine HCC, while some efficacy of sorafenib or lenvatinib was observed in such models.

HCC organoids were generated from CTNNB1/Myc HCC and used for a medium throughput drug screen with a compound library comprising 147 FDA approved drugs. The most efficacious drugs were a group of antimetabolites, nucleobase analogues that interfere with DNA synthesis. Cladribine, the most effective antimetabolite, was validated in murine and human organoids. In vivo cladribine treatment showed marked therapeutic efficacy in CTNNB1/Myc HCC (even in the presence of other lesions such as PTEN loss) but not in Kras driven liver carcinomas.

Overall the data presented by Muller et al. represents a useful resource for the HCC research community. A particular strength of the manuscript is the suggested new HCC classification, which seems superior to previously established and clinically used classifications of human HCC, such as the Hoshida classification. Nevertheless, while it is undisputed that the presented manuscript has great potential and represents a tour de force from a technical point of view, in its current form several parts of the study lack depth and rigorousness.

The following points could help to improve the manuscript:

1) The new HCC classification is compelling and harbours great translational potential. However, the presented analyses must be seen as a training set and it is mandatory to validate the new classification on further independent human cohorts.

2) Owing to the chosen figure format, it is very difficult for the reader to draw conclusions regarding correlation of a particular genotype and histopathological subtypes and tumor biological features, e.g. metastasis. A better format for presenting the data is needed. In the text it should be summarized which genotypes result in which histopathological subtypes and what the key features of these tumors are in terms of their tumor biology.

3) The authors confirm previous data that CTNNB1-driven HCC do not respond well to immune checkpoint blockade. However, as atezolizumab in combination with antiangiogenic therapy (bevacizumab) is effective in many human HCCs, the

herein presented platform should be used to pinpoint those genotypes that respond best to immune checkpoint blockade. Likewise, the potential of the presented platform to predict differences towards TKI therapy is not yet fully borne out. Is the platform suited to predict significant differences in the therapy responses towards sorafenib or lenvatinib ?

4) The data on the nucleoside analogon cladribine as a new therapeutic option for HCC treatment is interesting but raises many questions. Cladribine is a cytotoxic drug used for the treatment of hairy cell leukemia. Cladribine is a prodrug and the pharmacologically active metabolite CdATP accumulates in cells with high dCK activity and low desoxynucleotidase activity such as lymphocytes and other cells of the hematopoietic system. A high efficacy in HCC cells is rather unexpected and more data is needed to clarify whether the shown effects in HCC cells are on-target or whether off-target activities are involved. Also, the suggested genotype-specific effects of cladribine as a potential new HCC treatment remain unclear. Based on the presented data, only Kras driven tumors seem to respond less to cladribine. Have the authors confirmed that Kras-driven liver carcinomas really represent HCC and not cholangiocarcinomas ? Otherwise the comparison of treatment responses would be comparing apples to oranges.

5) The authors emphasize that one key finding of their study is that "MYC overexpression + Trp53 alteration, which induced HCC in previously published models", had very low to no tumour penetrance in their model. This statement is not correct, as it was also shown in transposon based models that low tumor penetrance is observed in a TrP53 deficient background (e.g. Dauch et al, Nature Medicine, 2016).

Version 1:

Reviewer comments:

Referee #2

(Remarks to the Author)

The manuscript "Human-correlated genetic HCC models identify combination therapy for precision medicine" by Muller et al is a substantially revised manuscript which describes the generation of a variety of genetically-driven HCC models in mice in vivo and in cell lines in vitro. Similar to the original manuscript the mouse models the investigators generated were then evaluated for transcriptomic data and then referenced with available human HCC data to correlate the mouse tumors with human HCC data. Overall this manuscript is a tour-de-force in the large scale of animal models and cell lines that were generated. And the tumor->cell line->drug screen->target->in vitro and in vivo studies represents a novel tool and platform for future studies and for the community. Substantial concerns with missing controls and experiments in the initial submission have been largely addressed in this revision. Overall with these changes and the substantial revision the manuscript should be accepted pending minor revision.

Comments for the author:

1. I appreciate the investigator response to the differences between the hepatic microenvironment in the varied mouse models and human clinical disease but I do believe that additional discussion is required. The investigator states that "the addition of background fibrotic disease having little transcriptomic influence" but I would argue that just points out the profound difference between the mouse models and the human cirrhosis and HCC development. HCC rarely forms outside of the human cirrhotic state (e.g. HBV, hemochromatosis, MASH) and cirrhosis is not a state that has been reliably generated in the mouse. This is not a criticism of the overall work but I just don't want the reader to miss the point that the lack of fibrosis/cirrhosis likely alters HCC tumorigenesis and development and this point should be more broadly discussed as a limitation of these studies.

(Remarks on code availability)

I reviewed the figures generated by the code but not the code itself.

Referee #3

(Remarks to the Author)

The authors have addressed all points in an appropriate manner. I suggest one minor change: the authors state "Lenvatinib and sorafenib, which are understood to act principally upon the tumour microenvironment, showed...". I do not agree that lenvatinib and sorafenib primarily exert their therapeutic effects by influencing the tumor microenvironment. The corresponding sentence should be changed.

(Remarks on code availability)

Referee #1 (Remarks to the Author):

Müller et al. present a robust and interesting manuscript centered around a plethora of GEMMs representing the majority of common human HCC driver genes. These GEMMs serve as a platform to identify novel HCC treatment strategies. Despite the tremendous efforts to comprehensively model human HCC in mice and to make use of the autochthonous HCC models to identify a combination therapy potentially surpassing the standard-of-care therapeutic lenvatinib, various issues related to the presented data need to be addressed to produce a compelling manuscript. Details are delineated below.

We are grateful for the positive assessment of the original manuscript and the model platform we have developed and interrogated for translational therapy in liver cancer.

1.1.

The authors cite Ally et al. (Comprehensive and Integrative Genomic Characterization of Hepatocellular Carcinoma, Cell, 2017; reference 4) as resource to identify HCC driver genes and potentially relevant combinations thereof. The authors should elaborate on why several top driver gene alterations (e.g., ALB or BAP1 LoF mutations) were not considered for modeling? Importantly, the authors should also provide statistical evaluations supporting the selected driver gene combinations, i.e., by using odds ratios and Fishers exact testing to evaluate co-occurrence / mutual exclusivity among co-drivers. Based on the TCGA oncoplot displayed in Figure 1 in Ally et al., it is unclear whether for instance TP53 and CTNNB1 mutations co-occur or are rather mutually exclusive. Such types of co-occurrence analyses using TCGA or other HCC oncogenomic datasets would strengthen the rationale for selection of modeled driver genes and especially their combinations.

Our original aim was to take account of both driver genes but also the pathway activation referred to in Figure 6 of Ally *et al.*¹. We commenced the modelling with β -catenin as it is believed to be the most difficult to treat, common, mutational driver in HCC. From here we expanded to other common HCC drivers and combinations thereof. As the reviewers suggested we have included additional models. This includes Bap1 in both male and female mice as BAP1 mutations are specifically enriched in female HCC patients; these data are now included in the revised figures (Figure 1b and Extended data Fig4a – shown below). We have not modelled ALB mutations currently as despite mutation sites it is not believed to be a driver mutation in HCC^{2,3}. There are similar instances of other cancers where tumour-type-specific hotspot targeting occurs in highly expressed genes from the epithelial origin³⁻⁵. Our data from analysis of the non-malignant liver suggests that ALB mutations are dominated by Indels, with putative causation from transcription-coupled mutagenesis^{6,7}.

We agree with the reviewer that in the instance of TP53 and CTNNB1 the two mutations preferentially desegregated in the TCGA data (and other datasets using resected HCC specimens). However, it is reported now that TP53 mutations become increasingly common in late-stage disease⁸; these are the patients typically treated with systemic chemotherapy for whom there is a particular unmet need. In response to the reviewer's point, we have performed further analysis on mutual exclusive and co-occurrent mutations using the TCGA dataset and have additionally performed analysis on co-occurrence/mutual exclusivity of β -catenin and TP53 in different stages of HCC; data shown in revised figures (Extended data Fig 2a+b, and below). While we see mutual exclusivity of β -catenin and TP53 mutations in early-stage disease, this is no longer apparent in late-stage disease. We, therefore, believe these models to have clinical relevance particularly to the development of non-surgical therapies for later stage HCC and therefore have included these combinations in our genetically engineered mouse models (GEMMs).

Fig1B – see manuscript for legends

Extended data Figure 4A (selection) – see manuscript for legends

Extended data Figure 2 – see manuscript for legends

1.2.

In Figure 1b, the authors present a tremendous effort to model driver gene combinations potentially relevant for HCC initiation/progression in mice. However, except for *Ctnnb1* and *Myc* (Extended Data Fig. 1i, l), the single driver gene cohorts are missing. In a revised version of this manuscript, these relevant controls should be provided and should also be included into the histological and molecular analyses in Figures 2-3.

We agree with this comment regarding single driver genes and have performed this in the relevant instances and included these data (Extended data Fig 1 and other data not shown in the original manuscript and included below). Due to the survival to pre-specified endpoint and the lack of tumours

in the majority of mice in these single driver cohorts, they were not further analysed for histology and molecular biology.

Specifically, to address this concern however, in addition to those include single driver controls for CTNNB1, MYC, in the original manuscript (Extended data Fig. 1i-l) we now include data for solitary induction of driver mutations e.g. BAP1 (see **Point 1.1** and data therein) and provide data for Notch, KRAS and Axin1 for the reviewers below. We show in the manuscript that, even with MYC, TP53 does not induce liver tumours in the model, therefore solitary TP53 is unlikely to be informative (Fig. 1B). For the constitutive knockouts (CDKN1A and CDKN2A) these models are well characterised systemically^{9,10}; leading typically to lymphoma and/or sarcomas within 12 months. No liver specific carcinogenesis is observed in these models. Our aim, from inception, has been to develop a model framework that can be expanded in the future to include increasingly infrequent tumour drivers as required; as we have with BAP1. We believe we have provided a thorough characterisation of the GEMMs and their tumours and demonstrate the utility of the model platform for translational science through a number of proof of principle studies in this manuscript.

Data for the reviewers' attention not included in the revised manuscript.

Survival of individual cohorts of male mice induced with AAV.TBG.Cre 6.4×10^8 GC/mouse for monoallelic presence of alleles described in the manuscript. Biological replicates shown for each

1.3.

The authors should present metastasis frequencies (resolved by metastatic site) for each cohort in the main Figure 1, rather than just in Extended Data Table 1.

We fully accept this point and for the clarity have adjusted the data visualisation and text to make the point clearer as we focused on and quantified the lung metastatic rate in each mouse model (Figs. 1a and d and Extended data Figs 12d and h).

1.4.

It remains unclear whether the emerging HCCs in each cohort are actually driven by the targeted driver genes. The authors should use IHC or other measures to evaluate overexpression or loss, respectively, of targeted driver genes in each cohort.

We have performed whole exome sequencing of a broad range of the tumour samples (those included in Figure 2), however, reliably validating the data has been exceptionally challenging due to difficulties in aligning to the reference standard mouse genome. We are working more broadly on ways to address this challenge. However, currently we are not confident enough in the data to include it in the revision. Nonetheless, to address this question specifically we have performed genetic analysis for the recombination of the target genes in question, again on the same samples used for the transcriptomic analysis (e.g. samples shown in Figure 2) to confirm recombination of the target alleles in the tumour tissue. These data are now shown in the revision in Extended data Figure 3 (and highlighted below). In summary, we show very high levels of predicted recombination, even in multi-allelic models and

even with the reduced dose of AAV.TBG.Cre (1.28×10^8 GC/mouse) in cohorts 14, 15, 23 and 24. We have updated the text and figures to reflect these data. For the attention of the reviewers, we believe that a single sample without recombination of both genotyped alleles (cohort 22) may have not been an HCC as the mouse was believed to have concurrent lymphoma, we have however included these data. There was some variability between alleles with all other instances of failure to recombine as predicted (n=3) being observed in the β -catenin allele and in association with loss of P16 (CDKN2A).

Extended data Fig 3 – see manuscript for legends

We would also like to highlight for the reviewers that immunohistochemistry for GS is a specific downstream marker of β -catenin pathway activity in the liver (shown in Extended data Figure S4) and we further highlight IHC validation of PTEN loss in PTEN deleted tumours in additional data for their attention below. We are making these whole scanned slides available to the community via the BiImage Archive, including a variety of stains (H&E and IHC), enabling linkage between the multimodal analyses described in the paper. Relevant accession numbers and an app portal web address are now provided in the revised manuscript. We have, therefore, not included the stains shown below in the revised manuscript itself.

Data for the reviewers' attention not included in the revised manuscript.

Additional IHC analysis of endpoint tumours from representative GEMM tumours across the 4 HuMos. Genetics of the cohorts is highlighted (grey=WT, yellow=Het, red =Hom). We highlight the absence of PTEN staining in the HuMo 2/4 samples where this gene is targeted for homozygous deletion, together with epithelial pAKT activation in these tumours and in addition the relative abundance of ZEB1 in the HuMo4 tumours consistent with the transcriptional signature as discussed in the manuscript.

1.5.

The RNA-seq based analyses in Figure 3 nicely identify 4 distinct HCC clusters using mouse and human TCGA data. Do the 4 clusters identified here correspond to the 4 clusters previously described in Ally et al. (Figure 2A)? The authors should elaborate on this and cross-compare their identified clusters to the previously described TCGA-HCC clusters. Cross-species analyses in Figure 3a should also be carried out in a driver-gene (combination)-specific manner. For instance, show both mouse and human HCCs harboring MYC amplification/overexpression similar transcriptional landscapes (e.g., enrichment of MYC_targets and metabolism hallmarks)?

The iCluster classification of the human TCGA-HCC dataset (Ally *et al.* ¹) consists of an integrated analysis of transcriptomics, microRNA expression, DNA copy number, reverse phase protein array, and DNA methylation signatures, therefore it will not be easily relatable to Chiang¹¹, Lee¹² or our HuMo clustering which are based upon transcriptomic signatures only. The iCluster categorisation of the human TCGA-HCC samples allows us to only unidirectionally compare these samples to our HuMo clusters. We provide these data below but have not included them for the reasons stated above.

Data for the reviewers' attention not included in the revised manuscript.

Classification comparison between the TCGA patients assigned by iCluster and their original iCluster status, presented as a proportion of the total HuMos cluster. Additional comparison between the iCluster and HuMo cluster status was not possible from transcriptomic data alone.

We focused on the Hoshida classification in the original manuscript but now include cross comparison of our HuMo clusters with additional established HCC subtypes. Specifically, we additionally cross compared the Chiang *et al.*¹¹ classification with our HuMo clusters, as gene set enrichment signatures for these samples are available. We also performed statistical analysis on the comparison of survivals in both Hoshida and Chiang subclasses and the contribution of each species to each in our dataset (Extended data Figure 9) as recommended in reviewer comment 1.8.

Extended data Fig 9 – see manuscript for legends

Additionally, we have analysed the data for cross species comparison for specific driver genes (Extended data Figure 10d, and below). Similarly, we show in a revised Fig 3a data on transcriptional targets of MYC activation in both species. Additionally, for the reviewers we provide analysis of MYC transcriptional targets in both species by MYC amplification status (see data for reviewers' attention below). This data demonstrates that in both species MYC amplification status is associated with equivalent elevations in the transcriptional signature described with MYC activation.

Figure 3 – see manuscript for legends

Extended data Figure 10d – see manuscript for legends

d

	N = 371 ¹	Immune Score			Fibroblast			Endothelial		
		Beta ²	SE ³	q-value ⁴	Beta ²	SE ³	q-value ⁴	Beta ²	SE ³	q-value ⁴
HuMo										
HuMo 1	140 (38%)	—	—	—	—	—	—	—	—	—
HuMo 2	110 (30%)	0.26***	0.022	<0.001	0.41***	0.028	<0.001	0.34***	0.030	<0.001
HuMo 3	67 (18%)	0.15***	0.026	<0.001	0.19***	0.032	<0.001	0.03	0.035	0.4
HuMo 4	54 (15%)	0.23***	0.028	<0.001	0.39***	0.035	<0.001	0.07	0.037	0.13
CTNNB1										
WT	283 (76%)	—	—	—	—	—	—	—	—	—
MUT	88 (24%)	-0.6*	0.025	0.025	-0.13***	0.034	<0.001	-0.05	0.034	0.3
TP53										
WT	266 (72%)	—	—	—	—	—	—	—	—	—
MUT	105 (28%)	0.00	0.024	0.9	-0.11**	0.032	0.001	-0.19***	0.030	<0.001
PTEN										
WT	358 (96%)	—	—	—	—	—	—	—	—	—
MUT	13 (3.5%)	-0.07	0.059	0.3	-0.08	0.08	0.4	-0.06	0.078	0.4
CDKN2A										
WT	360 (97%)	—	—	—	—	—	—	—	—	—
MUT	11 (3%)	-0.12	0.064	0.072	-0.02	0.087	0.8	-0.07	0.084	0.4

Data for the reviewers' attention not included in the revised manuscript.

Normalised Enrichment Score (NES) for MYC/Myc target genes in human and mouse respectively in MYC amplified versus WT in HCC tissue/models in human/mouse respectively.

1.6.

The authors claim that mutation status does not necessarily correspond to (downstream) signaling status (lines 138-139). Nevertheless, the authors should examine whether in their 4 clusters specific modeled drivers or driver combos were enriched. Moreover, it should be evaluated whether the corresponding human driver genes are likewise enriched in the human samples within the according clusters.

To formally test this, we have performed additional analysis. We provide this analysis below examining by mutation status the contribution to each HuMo subgroup in human HCC material and provide these data for the reviewers. Reassuringly to us, we observe CTNNB1 mutations enriched within HuMo 1 which corresponds to other subtypes enriched for this activation of this pathway in other molecular subtypes (e.g. Chiang and Hoshida; as discussed above in **Point 1.5**), but also in GEMMs driven by CTNNB1 and furthermore in our validation data analysis of human samples as described above and presented in Extended data Figure 8. We provide additional discussion of this point in the revised manuscript.

Data for the reviewers' attention

Odds ratios of HuMo cluster assignment based upon mutational status in HCC patients.

1.7.

Distinct morphological features across the 4 clusters described in the text (lines 160-166) and based on the HEs displayed in Figure 3b should be quantitatively assessed using appropriately sized cohorts for each cluster. These should include quantification of the histopathological features including pathological grade, differentiation, ECM deposition, steatosis, as well as immune infiltration. Moreover, in support of the morphological features observed in the HEs, authors could consider IHC stainings for markers representative of the described features (e.g., CD45 or more specific markers for immune cell subsets to characterize immune infiltration). Finally, to strengthen distinction of the 4 clusters, IHC markers representative of gene ontology hallmarks disclosed in Figure 3a should be used. For instance, EMT markers such as ZEB1 should preferentially stain cluster 4 tumours.

We have addressed this point by assessment by an expert histopathologist (T.J.K.) of a variety of quantitative histopathological features as recommended, i.e., inflammation (Fig. 3b), steatosis (Fig. 3c), extracellular matrix (Fig. 3d), tumour grade (Extended data Fig. 7b-c), and steatohepatitis (Extended data Fig. 7d). We have added quantification data including odds ratios as shown also comparing between the species which supports our conclusion for a correlation in histological features within the subtypes between the species. We would highlight also the correlation between inflammation and EMT seen as the transcriptional level as demonstrated between the GEMMs and TCGA data and additionally in an independent validation cohort of 171 human HCCs¹³ in Figure 3a and Extended data Figure 8b respectively. This data was consistent with the histological and clinicopathological patient's reports from the validation cohort. Tumours from the HuMo 3 and 4 clusters were less differentiated than tumours from the HuMo 1 and 2 clusters (55% vs 15%; $p=9E-06$) and displayed higher vascular invasion (63% vs 37%; $p=9E-03$). These patients also presented higher AFP levels (≥ 400 ng/ml) (45% vs 7%; $p=6E-07$). All these features further support the fact that HuMo 3 and 4 tumours are more aggressive than HuMo 1 and 2 tumours. HuMo 4 cluster presented significantly higher level of stromal/immune infiltration and consequently lower tumour purity (pathologically reviewed) compared to the other HuMo clusters (53% vs 7%; $p=2E-05$). Tumour purity was dichotomized into high tumour purity defined as per tumours displaying $\geq 60\%$ of tumour cells (and $\leq 40\%$ of stroma/immune cells) and low tumour purity with $\geq 40\%$ of stroma/immune cells and $\leq 60\%$ of tumour cells. This data paralleled the enrichment in inflamed class tumours observed in HuMo 4 tumours as is now shown in the Extended data Figure 1. These additional data support correlation between degree of differentiation that we observed histologically between the HuMo subtypes.

Figure 3 – see manuscript for legends

Extended data Figure 7 – see manuscript for legends

Extended data Figure 8 – see manuscript for legends

1.8.

Statistical evaluations of survival probability differences in Extended Data Figure 5c-e are missing.

We have provided these additional data in the revised figures as discussed previously in **Point 1.5** (Data is shown in Extended data Figures 9b-c [HuMo], f-g [Hoshida], j-k [Chiang]) and below.

Extended data Fig 9 - see manuscript for legends

1.9.

Enrichments of immune scores as displayed in oncoplot Extended Data Figure 6c should be quantified in both human and mouse cluster 1-4 and also subdivided into CTNNB1 WT vs mutant. Moreover, in the UMAP plot in Extended Data Figure 6d it is unclear where the 4 clusters locate to. The Figure legend does also not help understanding this panel. As currently displayed, it does not support the sentence in the manuscript text in lines 199-200.

We have updated the figure to provide data and description to address this point. This includes removal of original Figure 6d, and the additional of new data analysis including stats (Extended data Figures 10c-e and below).

Extended data Fig 10 - see manuscript for legends

d

	N = 371 [†]	Immune Score			Fibroblast			Endothelial		
		Beta ²	SE ²	q-value [‡]	Beta ²	SE ²	q-value [‡]	Beta ²	SE ²	q-value [‡]
HuMo										
HuMo 1	140 (38%)	—	—	—	—	—	—	—	—	—
HuMo 2	110 (30%)	0,26***	0,022	<0,001	0,41***	0,028	<0,001	0,34***	0,030	<0,001
HuMo 3	67 (18%)	0,15***	0,026	<0,001	0,19***	0,032	<0,001	0,03	0,035	0,4
HuMo 4	54 (15%)	0,23***	0,028	<0,001	0,39***	0,035	<0,001	0,07	0,037	0,13
CTNNB1										
WT	283 (76%)	—	—	—	—	—	—	—	—	—
MUT	88 (24%)	-0,6*	0,025	0,025	-0,13***	0,034	<0,001	-0,05	0,034	0,3
TP53										
WT	266 (72%)	—	—	—	—	—	—	—	—	—
MUT	105 (28%)	0,00	0,024	0,9	-0,11**	0,032	0,001	-0,19***	0,030	<0,001
PTEN										
WT	358 (96%)	—	—	—	—	—	—	—	—	—
MUT	13 (3,5%)	-0,07	0,059	0,3	-0,08	0,08	0,4	-0,06	0,078	0,4
CDKN2A										
WT	360 (97%)	—	—	—	—	—	—	—	—	—
MUT	11 (3%)	-0,12	0,064	0,072	-0,02	0,087	0,8	-0,07	0,084	0,4

e

	N = 187 [†]	Immune Score			Fibroblast			Endothelial		
		Beta ²	SE ²	q-value [‡]	Beta ²	SE ²	q-value [‡]	Beta ²	SE ²	q-value [‡]
HuMo										
HuMo 1	70 (37%)	—	—	—	—	—	—	—	—	—
HuMo 2	50 (27%)	0,26***	0,019	<0,001	0,24***	0,019	<0,001	0,29***	0,028	<0,001
HuMo 3	21 (11%)	0,07**	0,026	0,006	0,14***	0,025	<0,001	0,10**	0,037	0,010
HuMo 4	46 (25%)	0,08***	0,020	<0,001	0,32***	0,019	<0,001	0,11***	0,028	<0,001
Ctnnb1										
WT	52 (28%)	—	—	—	—	—	—	—	—	—
MUT	135 (72%)	-0,26***	0,014	<0,001	-0,12***	0,026	<0,001	-0,20***	0,027	<0,001
TP53										
WT	107 (57%)	—	—	—	—	—	—	—	—	—
MUT	80 (43%)	-0,08***	0,021	<0,001	-0,01	0,025	0,8	-0,04	0,028	0,2
Pten										
WT	120 (64%)	—	—	—	—	—	—	—	—	—
MUT	67 (36%)	-0,01	0,022	0,6	0,17***	0,022	<0,001	0,03	0,029	0,3
Cdkn2a										
WT	114 (61%)	—	—	—	—	—	—	—	—	—
MUT	73 (39%)	-0,06**	0,022	0,006	0,10***	0,024	<0,001	0,00	0,028	>0,9

1.10.

IHCs in Extended Data Figure 6e have to be quantified across cohorts of the respective genotypes and should be expanded by IHCs on samples representing the other human and mouse clusters, in support of Extended Data Figure 6c.

We relate our response to this point to that of response to **Point 1.7**. We have addressed this point by semi-quantitative histopathological assessment by an expert histopathologist (Fig. 3b). We have

removed the original Figure in Extended data figure 6e as this is now superfluous. This association with immune paucity is further highlighted in the transcriptomic data for immune signatures discussed previously in relating to Figure 3a cross the species and then further validated in independent patient cohort now presented in Extended data Figure 8, furthermore we focus in on the non-epithelial transcriptomic features in a revised Extended data Figure 10c (replacing the original Extended data Figure 6c).

The original Extended Data Figure 6e displayed IHC on glutamine synthetase (GS), CD48 (F4/80), CD4 and CD8 from CTNNB1 mutated samples (human and mouse). The reviewer asks to quantify the IHC across cohorts and include samples from other human and mouse clusters. The final objective is to support the original Extended Data Figure 6c. The original Extended Data Figure 6c showed that HuMo cluster 1 has low stroma and immune infiltration. As described in **Point 1.7**, HuMo clusters 1, 2 and 3 present all high levels of pathologically assessed tumour purity and, therefore, comparatively low percentage contribution of stroma and/or immune infiltration (see revised Extended data Figure 8b) relative to the HuMo 4 cluster, in line with what was shown in the original Extended Data Figure 6c. This HuMo 4 cluster is also enriched in CTNNB1 mutations. We show that, in the context of the validation cohort (Extended data Figure 8; discussed also in **Point 3.1**), CTNNB1 mutations are also enriched in the HuMo 1 cluster. Moreover, this cluster is enriched in the CTNNB1-driven HCC subclass¹⁴ and we have shown in the past that these tumours were positive for nuclear and cytoplasmic β -catenin staining and cytoplasmic GS (see figures below) much like their respective GEMM. We also describe in the revised manuscript that the HuMo cluster 4 is enriched in the Wnt-TGFB/proliferation HCC subclass and therefore, as we showed previously¹⁵, the β -catenin IHC of these tumours corresponds to a membrane staining (shown below).

Data for the reviewers' attention not included in the revised manuscript.

Equivalent GS and nuclear β -catenin in the cohort 5 model representative of HuMo1; red-CMYC, Green-GS, White β -catenin, magenta-CD45. Scale bars – 100um.

Published data for the reviewers' attention

Two distinct Wnt classes and their specific Wnt-related mRNA expression profiles in HCC. **A**) Prediction of Chiang's¹¹ and Hoshida's¹⁶ CTNNB1- and Wnt-TGFβ subclass using the NTP module from Gene Pattern (FDR<0.05). CTNNB1 class was significantly enriched for combined nuclear-cytoplasmic β-catenin (p<0.001) and cytoplasmic glutamine synthetase (GS, p<0.05) immunohistochemistry (IHC). Positive samples for signatures capturing CTNNB1-mutation and CTNNB1-Wnt signaling were significantly correlated with the CTNNB1 class (p<0.001), whereas samples that were positive for the Wnt-TGFβ gene signature significantly overlapped with the Wnt-TGFβ class (p=0.005). **B–D**) IHC of β-catenin and GS: percentage of positive samples in the HCC molecular subclasses and representative positive immunostaining for nuclear and cytoplasmic β-catenin, membrane β-catenin, and GS (Figures extracted from Lachenmayer *et al.*¹⁵).

On the other hand, we previously reported that tumours of the proliferation class (here corresponding to HuMo 3 and HuMo 4) have activation of AKT/mTOR signalling^{11,16}, and are stained for phospho-S6, a kinase that is downstream of mTORC1 and regulates protein synthesis and allows progression from the G1 to the S phase. In the past we have shown that these tumours also present positivity for pEGFR and pIGFR¹⁷ (see below). This is additionally broadly in agreement with our staining in the GEMM subtypes as shown above in **Point 1.4**.

Published data for the reviewers' attention

A Table 2. Analysis of Clinicopathologic and Molecular Variables Associated With p-RPS6 Staining in Human HCC (n = 82)

Clinicopathologic variables	P
BCLC staging B/C	.003
AFP plasmatic levels	.01
Platelet count	.03
Moderately/poorly differentiated tumor	.04
Vascular invasion	.17
Multinodularity	.34
Bilirubin level	.45
Albumin level	.6
Tumor size	.9
Molecular variables	
High mRNA levels of EGF	.01
p-EGFR staining	.02
p-IGFR1 staining	.03
p-mTOR staining	.2
p-AKT staining	.6

- A) Table displaying the clinicopathologic and immunohistochemistry data associated with pS6 and mTOR activation. Phospho-EGFR and pIGFR1 were significantly associated with pS6 (p=0.02 and p=0.03, respectively). B) Representative images for the immunostaining of phospho-EGFR, phospho-mTOR, and phospho-RPS6 in human HCC samples and cirrhotic tissue (as control). Amplified images represent the following: (i) Membranous localization of p-mTOR in cirrhotic tissue and loss of membranous localization of p-mTOR in HCC. (ii) Prominent positive staining for p-S6 in endothelial cells in HCC (absent in cirrhosis). Figure extracted from Vilanueva *et al.*¹⁷

We provide below a summary all these data:

1.11.

Besides clusters 1 and 2 highlighted in Extended Data and analyzed in 7d, the authors should also perform gene ontology analyses on the remaining gene clusters. In particular, the top and bottom clusters showing 'high-high-low' and low-low-high' gene expression patterns appear intriguing, as

these indicate transcriptional changes in tumor progression from adenoma to adenocarcinoma (which, to some extent, is also the case for cluster 1).

We have added the missing gene ontology analysis as requested (Extended data Figure 10d).

Extended data Figure 11d. - see manuscript for legends

1.12.

Although TKIs, such as sorafenib and lenvatinib, are first-line therapies for HCC, it appears unclear as to why in particular the cohort 5 / BM model should be sensitive to these inhibitors? Are other HCC mouse models equally sensitive to these inhibitors? Moreover, the authors should leverage the RNA-seq analyses in Extended Data Figure 7c (and possibly of other cohorts than cohort 5) to demonstrate upregulation of RTKs targeted by sorafenib / lenvatinib (specifically) in the cohort 5 / BM model, serving as a rationale for these targeted interventions.

We agree with the reviewer that there is no evidence of specificity in TKI response in either the models we describe nor in human patients. We have trialed TKIs in multiple models from multiple HuMo clusters and show in the manuscript that each shows response of a related magnitude, improving survival transiently. We, therefore, do not feel that inclusion of further data analysis from the bulk RNAseq will be helpful for identifying a transcriptomic rationale for the multicellular efficacy of TKIs in HCC. We apologise for our lack of clarity in the original manuscript. We did not intend to claim this but have ensured that the wording is clearer now.

1.13.

Anti-PD therapy showed no efficacy in cohort 5 tumors (Figure 4g). These data should either be moved to supplementary or further analyses including immunoprofiling would be needed to explain this

observation. Are any of the other modeled driver gene combinations giving rise to tumors responding to ICIs?

In response to this reviewer's comment, we have moved this to supplementary data (Extended data Figure 12). We would envisage that, using the system we describe, a large body of future work will explore immunotherapy responses across the model further. Having now validated our HuMo clusters in an independent cohort of patients and shown relevance to immune subclass we have also trialled the combination of VEGF pathway inhibition and ICI and these data are also included in a revised Extended data Figure 12 and are discussed in more detail in response to **Point 3.1**

1.14.

In Extended Data Figure 9b, authors should also demonstrate maintained (over)-expression of MYC.

We are happy to provide this data in a revised Extended data Figure 13b using immunohistochemical detection of Myc in the organoids compared to the GEMM tumours from which they are derived in cohort 5. This is also broadly consistent with the transcriptomic data now presented in Extended data Figure 13c but due to the NAS scores across the groups, without baseline control, this is not clearly visualised in this data representation.

Extended data Figure 13b lower panel. - see manuscript for legends

1.15.

In Extended Data Figure 9c, authors should also show 'growth volumes' of control organoids. Moreover, the mTOR inhibitor everolimus is incorrectly named 'iverolimus'.

We have addressed these comments in a revised Extended data Figure 13a and show these changes below.

Extended data Figure 13a. - see manuscript for legends

1.16.

Figure 5b shows that Trp53, Pten, and/or Cdkn2a LoF renders cohort 5 / BM organoids insensitive to cladribine. Examination of organoids from cohorts 11 (BM Trp53), 12 (BM Cdkn2a), and others would allow pinpointing the driver gene required for cladribine insensitivity. Furthermore, is cladribine sensitivity specific for the Ctnnb1 / Myc driver gene combo or are other cohorts that do not harbor Trp53, Pten, and/or Cdkn2a LoF also sensitive to cladribine?

We have now tested cladribine sensitivity in a broader range of lines as suggested, i.e., β -catenin/cMyc (BM) + Trp53, BM + Trp53 + Cdkn2a, BM + Trp53 + Cdkn2a + Pten. Specifically adding p53, p16 and Pten perturbation sequentially does not significantly affect the sensitivity to cladribine. This might also be affected by the culture conditions. We have included the additional data in the revised manuscript (Fig. 5b + Extended data Figure 14c; both shown below). We were only able to grow Pten/Myc/Trp53mut organoids in the presence of Wnt agonists (Rspodin1) in the culture media. Testing cladribine in these organoids and did not see any difference to the BM (cohort 5) tumoroids which we assume to be due to the culture conditions activating the Wnt/Bcatenin pathway.

Figure 5b. - see manuscript for legends

Extended data Figure 14b and c. - see manuscript for legends

1.17.

It would be nice if the authors were using in Figure 5c a human organoid line that would also harbor a MYC amplification. Based on the human organoid lines used, it appears that TP53 mutation status does not affect cladribine sensitivity, which, however, should be put to the test as delineated in comment 16.

We have added data on the Myc mutational status of the human organoid lines used and can confirm that a variety of non-amplified and Myc-amplified lines are present. (Fig. 5c and Extended data Figure 14f). As the reviewer comments, p53 mutation status does not affect cladribine sensitivity neither in human nor in mouse tumoroids, see previous **Point 1.16**.

1.18.

In the text (lines 280-282) related to the combination drug interventions presented in Figure 6c, authors refer to 20% weight loss observed in up to 60% of the mice treated with drug combo. Thus, in the Kaplan-meier curve of the combination regime, many of the mice are censored during combination therapy due to clinical symptoms (weight loss). Therefore, the survival proportions as presented are misleading (mice succumbed to side-effects cannot be considered 'cured'). Instead of displaying disease-specific survival, authors should also plot overall survival. For the animals sacrificed during treatment, it remains unknown whether eventual tumor relapse would occur, comparable to the cladribine single treatment arm. As currently presented, the data are no convincing case for the combination therapy. The authors might want to optimize cladribine and/or lenvatinib regimes especially for the combination treatment optimally minimizing side effects while maintaining potency.

Importantly, the single treatment arm for lenvatinib is missing in panel 6c as well as in subsequent analyses in Figure 6e-i (and corresponding Extended Data panels).

We have modified the visualisation of our data and have now changed all survival graphs using only uncensored data as suggested by the reviewer. We should point out that, as stipulated by the UK Home Office animal licensing conditions, these are not genuine "survival" studies but instead take

account of prespecified humane endpoints rather than death. We believe that there are benefits/caveats of each way of presenting the data (e.g. censoring removes other non-tumoural related endpoints). We now however include uncensored survival analyses, even with lower tumour burden at sampling, which we trust satisfies this concern.

Regarding the comments relating to relapse and dose optimisation we would highlight the data in Fig. 6d. These data (shown below) provide a longitudinal visualisation of tumour count and volume following this combination therapy. From these data and their cross relationship to the survival curves it is apparent that the long-term survival of multiple animals in the combination arm of the study is accompanied by long term tumour control and that this long-term tumour control is only observed in the combination arm and occurs soon after the institute of combination therapy (shown in red). Whilst some responses are observed in the cladribine only arm of the study (black), all animals progressing past median survival in this arm had significant tumour masses at endpoint. In comparison, both animals in the combination study sampled at the conclusion of the prolonged time course had minimal disease. Overall, these data would suggest that there is long-term resistance-free survival with combination therapy.

Figure 6d. - see manuscript for legends

Weight loss related early endpoints were seen in both the cladribine monotherapy and combination arms and weight loss itself did not predict strong tumour suppression in the cladribine arm. Therefore, dose limiting side effects of cladribine, did not predict tumour response. This, therefore, argues strongly for a synergy between the compounds (cladribine and lenvatinib). We highlight the impressive tumour control observed in long term therapy as shown in Fig. 6e.

We do not see that further long-term endpoint studies using this drug combination at the previous doses will benefit the study. We have performed a study with reduced dosing, and this maintained treatment efficacy but again with instances of off-target weight loss were observed (n=3 mice). In the interests of animal welfare, we instead prioritised other longer term therapy studies in our revision as suggested by the reviewers in total.

1.19.

The observed CD3+ T-cell influx into cladribine / lenvatinib combo-treated tumors 30 days post treatment start (and apparently before mouse loss due to treatment side effects) is interesting.

Authors could set-up an intervention experiment where cohort 6 mice would be primed for 30 (or less) days with cladribine / lenvatinib combo and then subsequently enrolled for ICI therapy.

This is an interesting and insightful suggestion; we have now performed these experiments (using Cohort 5 mice - BM) where we primed the mice with cladribine/lenvatinib followed by ICI. We now providing these additional data in revised Fig. 6i-k and Extended data Figure 16.

Based on the data shown in the figure above we chose a 7-day lead in with a priming combination of cladribine/lenvatinib which was ceased prior to ICI initiation. We see some evidence of antitumour activity after 7 days of cladribine/lenvatinib prior to ICI in this arm but insufficient to reduce tumour number at timepoint.

We have analysed these treatment arms with flow cytometry and IHC and see evidence of long-term immune modulation by using the priming combination of cladribine/lenvatinib prior to ICI. These effects include an increase in functional cytotoxic T lymphocytes (Granzyme B⁺/CD8⁺ T lymphocytes) both in the residual tumours and in the surrounding non-tumoural liver. There were subtle effects with the treatment with ICI monotherapy but no effects on overall tumour response. However, in the combination arms the altered immune environment was associated with a marked reduction in overall tumour count with this therapy (4 of 6 animals had no macroscopic nor microscopic tumours on whole lobe sections). We are, therefore, very grateful for this suggestion and delighted to include these additional data in the revision.

Figure 6. - see manuscript for legends

Extended data Figure 16. - see manuscript for legends

1.20.

It is intriguing in Figure 6c, i, j that cladribine / lenvatinib combination appears to be specifically efficient in models driven by *Cttnb1* and *Myc*. The authors could also apply the cladribine / lenvatinib combination to human organoids carrying altered *CTNNB1* (and *MYC*, if available).

We have addressed this with combination therapy in the murine HCC organoids (Extended data Figure 14b+c). We observed no difference between cladribine/lenvatinib combination therapy and cladribine

monotherapy. We are grateful for this suggestion but would propose that these data are not surprising as Lenvatinib acts principally upon the tumour microenvironment and not epithelial cells. This is consistent with the lack of effect of Lenvatinib we observed across a range of HCC derived organoids both from the murine GEMMs and from patient samples. As we did not observe an effect in mouse tumoroids, we did not perform the combination in human tumoroids as they also showed insensitivity to lenvatinib monotherapy (Extended data Figure 14d).

Extended data Figure 14b. - see manuscript for legends

Extended data Figure 14d. - see manuscript for legends

We believe these data highlights the importance of an integrated model system as we utilise here which can move from epithelial targeting therapies and combine with additional therapies both targeting the tumour microenvironment and tumour immune microenvironment in human relevant immune competent models as we describe herein.

Referee #2 (Remarks to the Author):

The manuscript "Human-correlated genetic HCC models identify combination therapy for precision 2 medicine" by Muller et al describes the generation of a variety of genetically-driven HCC models in mice in vivo and in cell lines in vitro. From these mouse models the investigators generated and then evaluated transcriptomic data and referenced them with available human HCC data to correlate the mouse tumors with human HCC data. Mouse subtypes that correlated with the human subtypes were similar in that mutational status was not linked to signaling (as similar to seen in prior human HCC). had distinct histopathologic findings. From the cell lines that were generated a drug was identified (caladribine) that improved HCC subtype X outcome that had not been implicated in HCC treatment previously. Overall this manuscript is a tour-de-force in the large scale of animal models and cell lines that were generated. And the tumor->cell line->drug screen->target->in vitro and in vivo studies represents a novel tool and platform for future studies and for the community. However, the authors ignore or do not mention a large body of literature that argues that HCC is dissimilar from other cancers and is not simply driven from tumor drivers but is due to the complex interaction of the microenvironmental changes that occur during cirrhosis, the ongoing regenerative stimulus, and also results from nodule and tumor heterogeneity. Given this and other concerns diminishes the impact of the manuscript and in its current form should be rejected with hope for resubmission.

Comments for the author:

2.1.

The authors state "Human HCC is thought to evolve from a hepatocytic clonal origin" and reference Brunner et al. While this has been the historical viewpoint on HCC development, Brunner et al and other more recent studies have shown and suggested that this is not the case. This represents a misinterpretation of this manuscript and needs to be corrected. Other reports for example (Zhu et al PMID: 30955891) identify several somatic mutations that are recurrent during chronic liver injury/liver disease promote regeneration without necessarily promoting hepatocarcinogenesis so the idea put forward as the premise underlying this manuscript that "driver" mutations identified in TCGA could be used to model human HCC somewhat ignore this body of literature.

We appreciate the reviewers concern here. Are original sentence was aiming to convey the importance of hepatocellular origin or HCC but we have modified the text accordingly and added additional references including that which is recommended.

The revised text now reads.

'Human HCC is thought to evolve from a hepatocytic clonal origin under specific conditions promoting carcinogenesis, in contrast to recently described non-malignant clonal expansion^{3,13-16}.'

Referencing

3. Brunner, S. F. et al. Somatic mutations and clonal dynamics in healthy and cirrhotic human liver. *Nature* 574, 538–542 (2019).
13. Guo, L. et al. Single-Cell DNA Sequencing Reveals Punctuated and Gradual Clonal Evolution in Hepatocellular Carcinoma. *Gastroenterology* 162, 238–252 (2022).
14. Zhu, M. et al. Somatic Mutations Increase Hepatic Clonal Fitness and Regeneration in Chronic Liver Disease. *Cell* 177, 608-621.e12 (2019).
15. Ng, S. W. K. et al. Convergent somatic mutations in metabolism genes in chronic liver disease. *Nature* 598, 473–478 (2021).

16. Verweij, N. et al. Germline Mutations in CIDEA and Protection against Liver Disease. *N. Engl. J. Med.* 387, 332–344 (2022).

2.2.

Also not mentioned is a major conclusion of the Brunner et al. paper that “the connective tissue laid down during cycles of damage and regeneration sequesters clones from early stages of the disease process” which is not being replicated in any of the generated mouse models. Even more interestingly in the Brunner et al they found that the majority of hepatocytes in the nodules did not have any driver mutations and then suggest that the clonal expansions that were observed were likely due to the inherent regenerative capacity of human hepatocytes.

We agree with this interpretation and additionally that the lack of modelling cirrhosis in murine models of liver disease is a significant deficiency for the field. The original manuscript did briefly discuss this aspect, but we have expanded it to highlight this in more detail.

The revised text now reads:

‘Preliminary data from our transcriptomic analyses indicated that genetics dominate cluster association, with the addition of background fibrotic disease having little transcriptomic influence (Cohort 5 vs 37). However, future research incorporating multifaceted environmental factors is needed to better understand HCC biology in human patients who usually present with chronic liver disease and hepatic impairments, which likely influences the course of disease establishment³ and impacts the treatment strategy available to patients^{2,20.}’

2.3.

Human HCC is marked by tumor heterogeneity and has been demonstrated by a multitude of reports. The authors do not address or examine this in any of the murine models. Do any of the models generate intra-tumor heterogeneity? How does tumor heterogeneity impact tumor growth, mouse survival, and (in later experiments) drug response.

We certainly observe significant inter- and intra-tumoral heterogeneity within the tumours histologically, more so in some models than others (Extended data Figure 5 and examples below). We observed also tumour heterogeneity in response to drug treatment (Extended data Figure 12e and example below). We also highlight the additional data highlighting heterogeneity in our response to **Point 1.4**. Due to the nature of the models, tumours are typically multifocal. Therefore, associating any specific tumour’s intratumoural heterogeneity to mouse survival is challenging. Relating tumour heterogeneity to clonal expansion and drug response requires longitudinal tracking of individual clones which is out width of the current study, which develops, validates, and performs translational therapeutic studies in this novel suite of genetic engineered mouse models. This is something we plan to address in future studies. We would highlight that the data shown below is just some examples of markers with associated heterogeneity, but we have also seen significant heterogeneity with cell state markers e.g. p21, Ki67 both within tumours between them and between cohorts.

Extended data Figure 5 (Cohorts 24 to 34 data). - see manuscript for legends

Extended data Figure 12e. - see manuscript for legends

2.4.

[REDACTED]

[REDACTED]

2.5.

The authors delineate the different mouse models into different clusters when compared to human data from TCGA. The 4 cohorts that are identified as being more similar phenotypically. Deeper cellular characterization of the tumors including the tumor microenvironment would further strengthen that disparate mutations result in similar phenotypical and cellular compositions.

We have performed further characterisation of these tumours. This includes additional histopathological characterisation including of the tumour microenvironment. We have related these data to human data as outlined in response to **Point 1.7**. We would like to highlight that the transcriptomic data from whole tumour tissue (including the TME) presented in the original manuscript demonstrates that various non-epithelial tumour hallmarks e.g. EMT and inflammation are different between the tumour subtypes and conserved between man and murine models within subtypes. (see Fig. 3b-d and Extended data Figure 7b-d). Please see also **Point 1.10** for additional discussion also and our additional IHC shown in **Point 1.4**.

2.6.

The authors show in Figure 4 that anti-PD1 therapy has no impact on survival in the Cohort 5 mice however a control showing benefit in another cohort group would show that this effect is specific to the CTNNB1 mutant cohort rather than a lack of response in general in the animal models as performed.

This point is related to **Point 1.13**. We have now added models of the standard of care Atezolizumab/Bevacizumab first line human HCC systemic therapy (combining AZ2717 with aPD1). We do see differences in response between cohorts (specifically cohorts 5 and 35). Importantly these are major representative of distinct subtypes of human disease associated not only with separate immune subclass states but also with signatures of ICI response in patients; see **Point 1.7** and data in revised Extended data Figure 8. We have also shown, as addressed in **Point 1.19**, that cladribine/lenvatinib priming sensitizes to ICI, showing that Cohort 5 mice can respond to ICI under the specific therapeutic modulation.

2.7.

The tumoroid platform in itself has been generated previously by several other groups however the generation of tumoroids from this broad cohort of GEMM-derived HCCs and mutations is novel. However it is not clear how this platform will capture the tumor heterogeneity seen in human HCC. It is not clear how the tumoroids drift over time and differ from the initial mouse or human tumor. Further studies evaluating the drift genomically, transcriptomically, and in cellular composition would add greater strength to this model platform.

To address the question of drift over time in the tumouroids we have perform RNA sequencing from HCC tumouroids over time and provide this data in a revision (Extended data Figure 13c; shown below). This demonstrates some evidence of stability over time of these murine HCC derived organoids; however, organoids with more mutations seem more unstable (as might be expected with *cdkn2a* loss for example). We would highlight the equivalent stability of human HCC tumouroids that was noted by Meri Huch's group in their original publication¹⁸.

We have tested different lines from different mice from the same cohorts and find great similarity in their response to drugs. Additionally, as discussed previously in **Point 1.16** with respect to sensitivity to cladribine in the different organoid lines, we see some different drug sensitivities between different organoid lines from different models. Therefore, we do find evidence of comparability of organoids of the same cohort which differs between cohorts. We find some evidence of them lacking in

heterogeneity, especially compared to the autochthonous mouse models, but they are particularly suited to test for responses relevant to HuMo1 and lend themselves to high throughput screening which we see as a major strength of their utility.

We therefore believe that they serve as an integrated addition to the platform but that *in vivo* validation of therapeutic response is crucial as organoids are a very simplified model lacking other components of the tumour microenvironment and are dependent upon and influenced by the culture condition.

Extended data Figure 13c. - see manuscript for legends

2.8.

The tumoroid screen identified cladribine as a potent target which was validated in multiple of the mouse HCC models. More detailed impacts of all of the antimetabolites would add further detail on these related compounds.

We have addressed this point by additional drug dose response studies using further antimetabolites in the organoids (revised Extended data Figure 14b+c). We observe a similar, but less sensitive response for clofarabine (a second-generation version of cladribine) as for cladribine itself. However, other antimetabolites, such as azacitidine, did not elicit a drug response in the tumoroids, indicating that what we see is not a general response to all antimetabolites but is class specific.

Extended data Figure 14 (selected data)c. - see manuscript for legends

2.9

Minor comments.

- 1) Histopathology noted in Figure 3B is somewhat hard to compare to each other due to figure size. Mouse images seem somewhat drowned out and not sure if this was due to figure uploading. Higher resolution images would greatly aid in its interpretation and should be included in the supplement.
- 2) Supp Figure 6E- immune staining for the different components should be from adjoining sections to better demonstrate the immune infiltration that is or is not present. The CD8 staining present seems to have a significant background staining.

As suggested by other reviewers we have quantified the histology. We have removed the previous data from Extended data Figure 6E as we find the quantification more representative of the whole Cohorts rather than showing one representative example. We provide figures at the resolution requested by the journal and are happy to provide raw data. We apologise if the resolution in the manuscript available to reviewers is of insufficient resolution.

Referee #3 (Remarks to the Author):

In their current manuscript Bird and colleagues report on the development and comprehensive characterization of twenty-seven genetically engineered autochthonous immunocompetent HCC mouse models. Adult mice harbouring endogenous floxed alleles of tumor suppressor genes or activatable oncogenes (particular focus on WNT pathway, cell cycle and RTK/RAS/PI3K pathway) were injected with an adenoviral vector encoding Cre recombinase under control of the hepatocyte specific thyroxine-binding Globulin (TBG) promoter to initiate tumorigenesis. Time to tumor onset, metastasis and survival were captured for all models. Furthermore, all models were subjected to a thorough histopathological analysis, which revealed a wide range of histological phenotypes, including well-differentiated HCC, undifferentiated HCC, pseudoglandular HCC and steatotic HCC, however, a clear genotype-phenotype correlation could apparently not be concluded from the presented data. Many models seem to develop lung and bone metastases which seems to be enriched in models driven by mutated CTNNB1 and Myc. All models were subjected to transcriptional profiling and obtained data was integratively analyzed with data from chemically induced mouse HCC as well as with human HCC transcriptomic data. Using the Louvain method the authors compared the human and mouse transcriptome data based on functionally and mechanistically relevant pathway enrichment. These analyses yielded four major human/mouse (HuMo) clusters. Genetic mouse models were represented in all four clusters with varying heterogeneity within cohorts, whereas the purely carcinogen-induced models were only found representative of HuMo cluster 2. HuMo cluster 1 was enriched for pathways linked to metabolism and differentiation but had negative enrichment for proliferation and inflammatory pathways. HuMo cluster 2 was related to cluster 1 but was distinct particularly through a higher enrichment in pro-inflammatory pathways. HuMo clusters 3 and 4 were both poorly differentiated and highly proliferative, with cluster 4 showing enrichment in epithelial-to-mesenchymal transition.

The presented cross species clustering approach seems more precise than previously established and clinically used classifications of human HCC, such as the Hoshida classification. For example, the herein presented clustering was able to distinguish two patient populations within Hoshida subclass S3, namely HuMo clusters 1 and 2. The reported distinction resulted in differences in patient survival, with patients associated with HuMo cluster 2 showing an improved survival probability relative to patients associated with the other HuMo clusters.

In line with recent reports, Muller et al could show that immune checkpoint blockade showed no therapeutic efficacy in CTNNB1-driven murine HCC, while some efficacy of sorafenib or lenvatinib was observed in such models.

HCC organoids were generated from CTNNB1/Myc HCC and used for a medium throughput drug screen with a compound library comprising 147 FDA approved drugs. The most efficacious drugs were a group of antimetabolites, nucleobase analogues that interfere with DNA synthesis. Cladribine, the most effective antimetabolite, was validated in murine and human organoids. In vivo cladribine treatment showed marked therapeutic efficacy in CTNNB1/Myc HCC (even in the presence of other lesions such as PTEN loss) but not in Kras driven liver carcinomas.

Overall the data presented by Muller et al. represents a useful resource for the HCC research community. A particular strength of the manuscript is the suggested new HCC classification, which seems superior to previously established and clinically used classifications of human HCC, such as the Hoshida classification. Nevertheless, while it is undisputed that the presented manuscript has great potential and represents a tour de force from a technical point of view, in its current form several parts of the study lack depth and rigorousness.

The following points could help to improve the manuscript:

3.1

The new HCC classification is compelling and harbours great translational potential. However, the presented analyses must be seen as a training set and it is mandatory to validate the new classification on further independent human cohorts.

We have validated our classification with an independent human data set. This was performed in an independent cohort of 171 HCC tumours samples that had been previously profiled at the immune, transcriptomic and mutational level in Montironi *et al.*¹³. For additional details on the analyses done with this cohort, refer to **Points 1.7** and **1.10**. These data are shown now in Extended data Figure 8 and below and pertain to discussion in **Points 1.7, 1.10** and **2.6** and can confirm our classification. Proportions of patient HCCs are similar between each HuMo cluster to that observed in the TCGA data. We also included the most recent transcriptomic predictors of response to ICI. We discuss the makeup and cross subtypes comparisons of the subclasses in more detail in **Point 1.10**. Furthermore, we have used these subclasses to design our testing of ICI therapy in distinct subclasses of the disease (notably HuMo 1 vs 2; representative immune evasive and immune active respectively). These differential ICI responses are discussed in **Point 2.6** but demonstrate lack of response in the immune evasive subtype unlike the significant response in the immune active subtype.

Extended data Figure 8a. - see manuscript for legends

3.2

Owing to the chosen figure format, it is very difficult for the reader to draw conclusions regarding correlation of a particular genotype and histopathological subtypes and tumor biological features, e.g. metastasis. A better format for presenting the data is needed. In the text it should be summarized which genotypes result in which histopathological subtypes and what the key features of these tumors are in terms of their tumor biology.

We have made significant changes to the presentation of the data in the revision. We are unclear as to how to address this point fully. Textual description of each genotype summary from the 37 models would be very challenging for both the format and the reader in our opinion. We will be happy to undertake a further revision of the figures under editorial guidance to address this point.

The strength of our model platform is to provide a wide range of models from which we can identify subtypes of the disease validated across the species. Furthermore, individual cohorts show faithful recapitulation of the intended genotype of the tumour (revised Extended data Figure 2). We would also like to highlight, that we see different histopathological subtypes in mice with the same genotype, therefore a 1:1 prediction of histopathological subtype and tumour biology is not possible based on genotype. We are also unable to link metastasis/tumour biological features to an individual tumour-of-origin in the autochthonous models which by their nature are multifocal. In theory the induction dose could be further lowered to produce monocentric and less penetrant disease for specific biological questions, but for practical reason we have focused on multifocal disease for testing therapy in the models described here.

The revised manuscript provides a detailed description over many genetic distinct models with detailed molecular characterization in over 200 animals. For obvious reasons providing level of granularity required to link individual features in the multimodal comparison in the main figures is challenging. However, we are happy to make the raw data publicly available and trust that with the revised figures we provide satisfactory evidence for our conclusions in the manuscript. I am afraid we have been unable, despite extensive discussion internally, to formulate a better way of distilling this multimodal data for presentation in a figure. We have discussed this also with the Journal Editor who has similarly struggled to find a solution, if any of the reviewers has any specific solutions we be delighted to consider an alternative presentation of the data.

3.3

The authors confirm previous data that CTNNB1-driven HCC do not respond well to immune checkpoint blockade. However, as atezolizumab in combination with antiangiogenic therapy (bevacizumab) is effective in many human HCCs, the herein presented platform should be used to pinpoint those genotypes that respond best to immune checkpoint blockade. Likewise, the potential of the presented platform to predict differences towards TKI therapy is not yet fully borne out. Is the platform suited to predict significant differences in the therapy responses towards sorafenib or lenvatinib?

We now include a number of additional therapeutic studies to address this point. We have modelled Atezolizumab/Bevacizumab (A/B) treatment in Cohort 5 and see non-significant trend towards improved survival. Additionally, correlated to human immune class, we demonstrate response to A/B in Cohort 35 as discussed previously in **Point 3.1** and **2.6**.

With respect to TKI responses, this is an interesting question but problematic with the differential timing between induction and late-stage disease relating to the aggression of disease in the individual GEMMs. Therefore, it would be challenging to compare improvements (measured in days) in survival on treatment between the models when the point of treatment initiation is likely to be different in each of the models. However, in all models tested we see temporising improvement in survival with

TKIs which is broadly equivalent, and this is consistent with clinical understanding of these therapies in patients where no predictors of TKI response exist for clinical utility currently to our knowledge.

3.4

The data on the nucleoside analogon cladribine as a new therapeutic option for HCC treatment is interesting but raises many questions. Cladribine is a cytotoxic drug used for the treatment of hairy cell leukemia. Cladribine is a prodrug and the pharmacologically active metabolite CdATP accumulates in cells with high dCK activity and low desoxynucleotidase activity such as lymphocytes and other cells of the hematopoietic system. A high efficacy in HCC cells is rather unexpected and more data is needed to clarify whether the shown effects in HCC cells are on-target or whether off-target activities are involved. Also, the suggested genotype-specific effects of cladribine as a potential new HCC treatment remain unclear. Based on the presented data, only Kras driven tumors seem to respond less to cladribine. Have the authors confirmed that Kras-driven liver carcinomas really represent HCC and not cholangiocarcinomas? Otherwise the comparison of treatment responses would be comparing apples to oranges.

We are unclear regarding the statement relating to on and off target effects of cladribine. We have shown direct on target effects through the tumour epithelium with our organoid screening and validation. We understand this comment to be related to mechanism of action e.g. through DNA versus other mechanism(s). In response to this we tested a range of different antimetabolite classes to explore further distinct mechanisms of sensitivity, where we see that clofarabine the second-generation compound of cladribine works similarly whereas other antimetabolites do not. This is similarly discussed in **Point 2.8**.

In response to the concern regarding the tumour type in the kRAS model. Firstly, we are confident from the specificity of the AAV system that the tumours have arisen from a hepatocellular origin. We take on board the concerns regarding phenotype of the resultant tumours and as per the classification of the tumours (Figure 1) the kRAS Myc model is mostly trabecular in pattern and lacks cholangiocarcinoma features including both Sox9 expression and any significant desmoplastic reaction. Histologically, therefore, the kRAS Myc driven GEMMs are consistent with HCC and this corresponds with their transcriptional alignment with HCC and distinction from cholangiocarcinoma. Furthermore this (and all other) models were reviewed by an expert histopathologist (T.J.K.) who is in agreement with their appearances being consistent with HCC and not cholangiocarcinoma.

3.5

The authors emphasize that one key finding of their study is that "MYC overexpression + Trp53 alteration, which induced HCC in previously published models", had very low to no tumour penetrance in their model. This statement is not correct, as it was also shown in transposon based models that low tumor penetrance is observed in a TrP53 deficient background (e.g. Dauch et al, Nature Medicine, 2016).

We have amended the text to include this point and reference the study quoted.

References:

- 1 Comprehensive and Integrative Genomic Characterization of Hepatocellular Carcinoma. *Cell* **169**, 1327-1341.e1323, doi:10.1016/j.cell.2017.05.046 (2017).
- 2 Letouze, E. *et al.* Mutational signatures reveal the dynamic interplay of risk factors and cellular processes during liver tumorigenesis. *Nat Commun* **8**, 1315, doi:10.1038/s41467-017-01358-x (2017).
- 3 Imielinski, M., Guo, G. & Meyerson, M. Insertions and Deletions Target Lineage-Defining Genes in Human Cancers. *Cell* **168**, 460-472.e414, doi:10.1016/j.cell.2016.12.025 (2017).
- 4 Kim, N. & Jinks-Robertson, S. Transcription as a source of genome instability. *Nature Reviews Genetics* **13**, 204-214, doi:10.1038/nrg3152 (2012).
- 5 Tarpey, P. S. *et al.* Frequent mutation of the major cartilage collagen gene COL2A1 in chondrosarcoma. *Nature genetics* **45**, 923-926, doi:10.1038/ng.2668 (2013).
- 6 Brunner, S. F. *et al.* Somatic mutations and clonal dynamics in healthy and cirrhotic human liver. *Nature* **574**, 538-542, doi:10.1038/s41586-019-1670-9 (2019).
- 7 Ng, S. W. K. *et al.* Convergent somatic mutations in metabolism genes in chronic liver disease. *Nature* **598**, 473-478, doi:10.1038/s41586-021-03974-6 (2021).
- 8 Nault, J. C. *et al.* Clinical Impact of Genomic Diversity From Early to Advanced Hepatocellular Carcinoma. *Hepatology (Baltimore, Md.)* **71**, 164-182, doi:10.1002/hep.30811 (2020).
- 9 Deng, C., Zhang, P., Harper, J. W., Elledge, S. J. & Leder, P. Mice lacking p21CIP1/WAF1 undergo normal development, but are defective in G1 checkpoint control. *Cell* **82**, 675-684, doi:10.1016/0092-8674(95)90039-x (1995).
- 10 Krimpenfort, P., Quon, K. C., Mooi, W. J., Loonstra, A. & Berns, A. Loss of p16Ink4a confers susceptibility to metastatic melanoma in mice. *Nature* **413**, 83-86, doi:10.1038/35092584 (2001).
- 11 Chiang, D. Y. *et al.* Focal gains of VEGFA and molecular classification of hepatocellular carcinoma. *Cancer research* **68**, 6779-6788, doi:10.1158/0008-5472.Can-08-0742 (2008).
- 12 Lee, J. S. *et al.* Application of comparative functional genomics to identify best-fit mouse models to study human cancer. *Nature genetics* **36**, 1306-1311, doi:10.1038/ng1481 (2004).
- 13 Montironi, C. *et al.* Inflamed and non-inflamed classes of HCC: a revised immunogenomic classification. *Gut* **72**, 129-140, doi:10.1136/gutjnl-2021-325918 (2023).
- 14 Llovet, J. M. *et al.* Hepatocellular carcinoma. *Nature Reviews Disease Primers* **7**, 6, doi:10.1038/s41572-020-00240-3 (2021).
- 15 Lachenmayer, A. *et al.* Wnt-pathway activation in two molecular classes of hepatocellular carcinoma and experimental modulation by sorafenib. *Clinical cancer research : an official journal of the American Association for Cancer Research* **18**, 4997-5007, doi:10.1158/1078-0432.Ccr-11-2322 (2012).
- 16 Hoshida, Y. *et al.* Integrative transcriptome analysis reveals common molecular subclasses of human hepatocellular carcinoma. *Cancer research* **69**, 7385-7392, doi:10.1158/0008-5472.can-09-1089 (2009).
- 17 Villanueva, A. *et al.* Pivotal role of mTOR signaling in hepatocellular carcinoma. *Gastroenterology* **135**, 1972-1983, 1983.e1971-1911, doi:10.1053/j.gastro.2008.08.008 (2008).
- 18 Broutier, L. *et al.* Human primary liver cancer-derived organoid cultures for disease modeling and drug screening. *Nature medicine* **23**, 1424-1435, doi:10.1038/nm.4438 (2017).

Referee #2 (Remarks to the Author):

The manuscript “Human-correlated genetic HCC models identify combination therapy for precision medicine” by Muller et al is a substantially revised manuscript which describes the generation of a variety of genetically-driven HCC models in mice in vivo and in cell lines in vitro. Similar to the original manuscript the mouse models the investigators generated were then evaluated for transcriptomic data and then referenced with available human HCC data to correlate the mouse tumors with human HCC data. Overall this manuscript is a tour-de-force in the large scale of animal models and cell lines that were generated. And the tumor->cell line->drug screen->target->in vitro and in vivo studies represents a novel tool and platform for future studies and for the community. Substantial concerns with missing controls and experiments in the initial submission have been largely addressed in this revision. Overall with these changes and the substantial revision the manuscript should be accepted pending minor revision.

We are grateful for the positive assessment of our revised manuscript.

Comments for the author:

I appreciate the investigator response to the differences between the hepatic microenvironment in the varied mouse models and human clinical disease but I do believe that additional discussion is required. The investigator states that “the addition of background fibrotic disease having little transcriptomic influence” but I would argue that just points out the profound difference between the mouse models and the human cirrhosis and HCC development. HCC rarely forms outside of the human cirrhotic state (e.g. HBV, hemochromatosis, MASH) and cirrhosis is not a state that has been reliably generated in the mouse. This is not a criticism of the overall work but I just don’t want the reader to miss the point that the lack of fibrosis/cirrhosis likely alters HCC tumorigenesis and development and this point should be more broadly discussed as a limitation of these studies.

We accept that this is a point which should be clarified further, and we have modified the text to reflect this.

Previous text in discussion:

Beyond the investigation of further genetic alterations, our models can also be easily combined with environmental liver disease models, such as high-fat diets. Preliminary data from our transcriptomic analyses indicated that genetics dominate cluster association, with the addition of background fibrotic disease having little transcriptomic influence (Cohort 5 vs 37). However, future research incorporating multifaceted environmental factors is needed to better understand HCC biology in human patients who usually present with chronic liver disease and

hepatic impairments, which likely influences the course of disease establishment³ and impacts the treatment strategy available to patients.

Revised text in discussion:

In contrast to the GEMMs, human patients usually present with cirrhosis, which likely influences the course of disease establishment and progression and impacts treatment options. Future research incorporating multifaceted environmental factors in preclinical models, including advanced fibrosis, is needed to better understand HCC biology and potential differences between species. Our models can also be easily combined with environmental liver disease models, such as high-fat diets. Preliminary data from our transcriptomic analyses indicated that genetics dominate cluster association, with the addition of background fibrotic disease having little transcriptomic influence in mice (Cohort 5 vs 37).

Referee #3 (Remarks to the Author):

The authors have addressed all points in an appropriate manner. I suggest one minor change: the authors state “Lenvatinib and sorafenib, which are understood to act principally upon the tumour microenvironment, showed...”. I do not agree that lenvatinib and sorafenib primarily exert their therapeutic effects by influencing the tumor microenvironment. The corresponding sentence should be changed.

We agree are grateful for the positive assessment of our revised manuscript. We do not entirely agree with the reviewer on one detail in this point. Our data and some other data would suggest that, at least in epithelial organoid systems these tyrosine kinase inhibitors are remarkably ineffective upon the tumour epithelium. However, we accept that this is a point for debate and further research. We have modified the text to take account of this.

Previous text (Line 322 approx).

Lenvatinib and sorafenib, which are understood to act principally upon the tumour microenvironment, showed little tumour-epithelial efficacy in both the screen and separate validation, including in combination with cladribine (Extended Data Fig. 14b-f).

Revised text:

Lenvatinib and sorafenib showed little tumour-epithelial efficacy in both the screen and separate validation, including in combination with cladribine (Extended Data Fig. 13b-f).